# Control of voluntary and optogenetically perturbed locomotion by spike rate and timing of neurons of the mouse cerebellar nuclei

Rashmi Sarnaik[†], Indira M Raman*

Department of Neurobiology, Northwestern University, Evanston, United States

**Abstract** Neurons of the cerebellar nuclei (CbN), which generate cerebellar output, are inhibited by Purkinje cells. With extracellular recordings during voluntary locomotion in head-fixed mice, we tested how the rate and coherence of inhibition influence CbN cell firing and well-practiced movements. Firing rates of Purkinje and CbN cells were modulated systematically through the stride cycle (~200–300 ms). Optogenetically stimulating ChR2-expressing Purkinje cells with light steps or trains evoked either asynchronous or synchronous inhibition of CbN cells. Steps slowed CbN firing. Trains suppressed CbN cell firing less effectively, but consistently altered millisecond-scale spike timing. Steps or trains that perturbed stride-related modulation of CbN cell firing rates correlated well with irregularities of movement, suggesting that ongoing locomotion is sensitive to alterations in modulated CbN cell firing. Unperturbed locomotion continued more often during trains than steps, however, suggesting that stride-related modulation of CbN spiking is less readily disrupted by synchronous than asynchronous inhibition.
DOI: https://doi.org/10.7554/eLife.29546.001

**\*For correspondence:**
i-raman@northwestern.edu

**Present address:** [†]Boston Children's Hospital, Boston, United States

## Introduction

The activity of neurons of the cerebellar nuclei (CbN) facilitates motor learning, error correction, and coordination of movements (*Manto, 2008*; *McCormick and Thompson, 1984*; *Medina et al., 2002*; *Morton and Bastian, 2004*; *Raymond et al., 1996*; *Thach, 1968*; *Thach et al., 1992*). CbN neurons, which form the sole output of the non-vestibular cerebellum, receive excitation from mossy fibers from multiple brain regions and convergent inhibitory input from Purkinje cells of the cerebellar cortex. Learning-related reductions of Purkinje cell activity facilitate novel motor behaviors, presumably by disinhibiting CbN cells (*Albus, 1971*; *Gilbert and Thach, 1977*; *Ito et al., 1964*; *Jirenhed et al., 2007*; *Marr, 1969*; *Medina and Mauk, 1999*). Indeed, movements can be triggered by optogenetically reducing firing by Purkinje cells, thereby elevating CbN cell activity (*Heiney et al., 2014*; *Lee et al., 2015*; *Proville et al., 2014*; *Witter et al., 2013*; *Ten Brinke et al., 2017*). Conversely, in behaving animals, pharmacologically inactivating regions of the CbN prolongs reaction times, implicating the CbN in controlling even simple movements (*Goodkin and Thach, 2003*; *Mason et al., 1998*). Moreover, motor control is severely disrupted by pushing CbN cell output to either extreme, that is, by silencing cerebellar output through complete ablation or by complete disinhibition through Purkinje cell loss (*Glickstein, 1994*; *Mark et al., 2015*; *Vinueza Veloz et al., 2015*; *Yu et al., 2015*). Movement disorders can also be generated by mutations that make CbN cells fire irregularly (*Fremont et al., 2017*; *LeDoux et al., 1998*; *Walter et al., 2006*). Together, these results suggest that both the rate and timing of CbN cell activity must be precisely regulated.

Nevertheless, how the firing patterns of CbN neurons shape ongoing, well-practiced movements such as locomotion is less well defined. The firing rates of Purkinje as well as CbN neurons increase

and decrease systematically during walking (*Armstrong and Edgley, 1984a*, *Armstrong and Edgley, 1984b*) and other volitional movements (*Armstrong and Rawson, 1979*; *Arshavsky et al., 1980*; *Harvey et al., 1979*; *Thach, 1968*), consistent with the idea that modulated CbN cell firing contributes to motor behaviors; however, passive motion can induce similar modulation (*Casabona et al., 2010*; *Cody et al., 1981*). The question is complicated further by the responses of CbN cells to Purkinje cell input. Despite extensive convergent inhibition and powerful inhibitory synaptic contacts (*Ito et al., 1964*; *Palkovits et al., 1977*; *Person and Raman, 2012a*), the firing rates of Purkinje and CbN cells are not always anticorrelated (*Armstrong and Edgley, 1984b*; *McDevitt et al., 1987*; *Thach, 1968*). Predicting CbN output from Purkinje cell rates may be nontrivial in part because both cell types are excited directly (CbN) or indirectly (Purkinje, via granule cells) by mossy fibers, whose activity is also modulated during locomotion (*Powell et al., 2015*). In addition, CbN cells in vitro are sensitive to the temporal correlation among convergent inhibitory inputs (*De Zeeuw et al., 2011*; *Gauck and Jaeger, 2000*; *Person and Raman, 2012a*; *Wu and Raman, 2017*).

Such observations raise the questions of how CbN cells respond to modulated inhibition from Purkinje cells and how these responses relate to behavior. To explore these questions, we recorded extracellularly from stride-modulated Purkinje or CbN cells, while monitoring hind paw movement in awake, head-fixed mice running voluntarily on a cylindrical treadmill. The rate and temporal pattern of inhibition of CbN cells were briefly altered by optogenetically stimulating ChR2-expressing Purkinje cells. Perturbations of CbN cell firing rate modulation correlated well with irregularities ('slips') in the step cycle, and the slip probability varied with whether the light stimulus evoked asynchronous or relatively synchronous Purkinje cell firing.

## Results

### Targeting Purkinje and CbN cells with running-related activity

To investigate the activity of principal neurons of the cerebellum during voluntary locomotion, we recorded from Purkinje neurons and CbN cells in awake, head-fixed mice running on a freely rotating (non-motorized) cylindrical treadmill, while simultaneously tracking locomotion by video-monitoring the ipsilateral hind paw (*Figure 1A*). The mice were the offspring of Ai27D x *Pcp2*-cre crosses and expressed channelrhodopsin (ChR2) in Purkinje cells. Loose cell-attached recordings were made from either Purkinje neurons in the lobulus simplex or CbN cells in the interpositus nucleus, which are known to be modulated during walking in cats (*Armstrong and Edgley, 1984a*). Pilot studies identified corresponding regions of the mouse interpositus in which firing rates of CbN cells were modulated during strides (Materials and methods). In separate experiments, Purkinje cells projecting to regions of the CbN with stride-modulated cells were identified by retrograde tracing with fluorescent markers (Materials and methods). Labeled Purkinje somata were consistently found in the ventral third of lobulus simplex, always including the bottom of the primary fissure (*Figure 1B*), and often extending about 500 μm more posteriorly. These regions of lobulus simplex were stereotaxically targeted for Purkinje cell recordings. The angle of approach of the electrode was adjusted so that Purkinje and CbN cells in likely connected regions could be obtained in the same penetration. In 35/41 recorded Purkinje cells and 39/46 recorded CbN cells, at least one condition of stimulation via the optical fiber in the patch electrode increased firing rates of Purkinje cells or decreased firing rates of CbN cells, confirming that the dominant effect of ChR2 activation was to excite Purkinje cells, and also altered movements of the ipsilateral hindlimb (analyzed further below). Ipsilateral forelimb or trunk movements could occasionally be evoked as well, indicating that the Purkinje and CbN cells were not exclusively associated with regulation of hindlimb movement. Nevertheless, only the hindlimb was monitored, since the goal was to track continuity or discontinuity of the stride, rather than to correlate neuronal activity with actions of specific muscles.

After recordings, Purkinje and CbN cell locations were labeled with injections of fluorescent dye. All Purkinje cells in the study were in the lobulus simplex (VIa) and spanned about 200–300 μm in the anterior-posterior axis (*Figure 1C and D*). CbN cell locations were usually in the posterior interpositus (*Figure 1C and D*), but sometimes extended to the anterior and dorsolateral divisions.

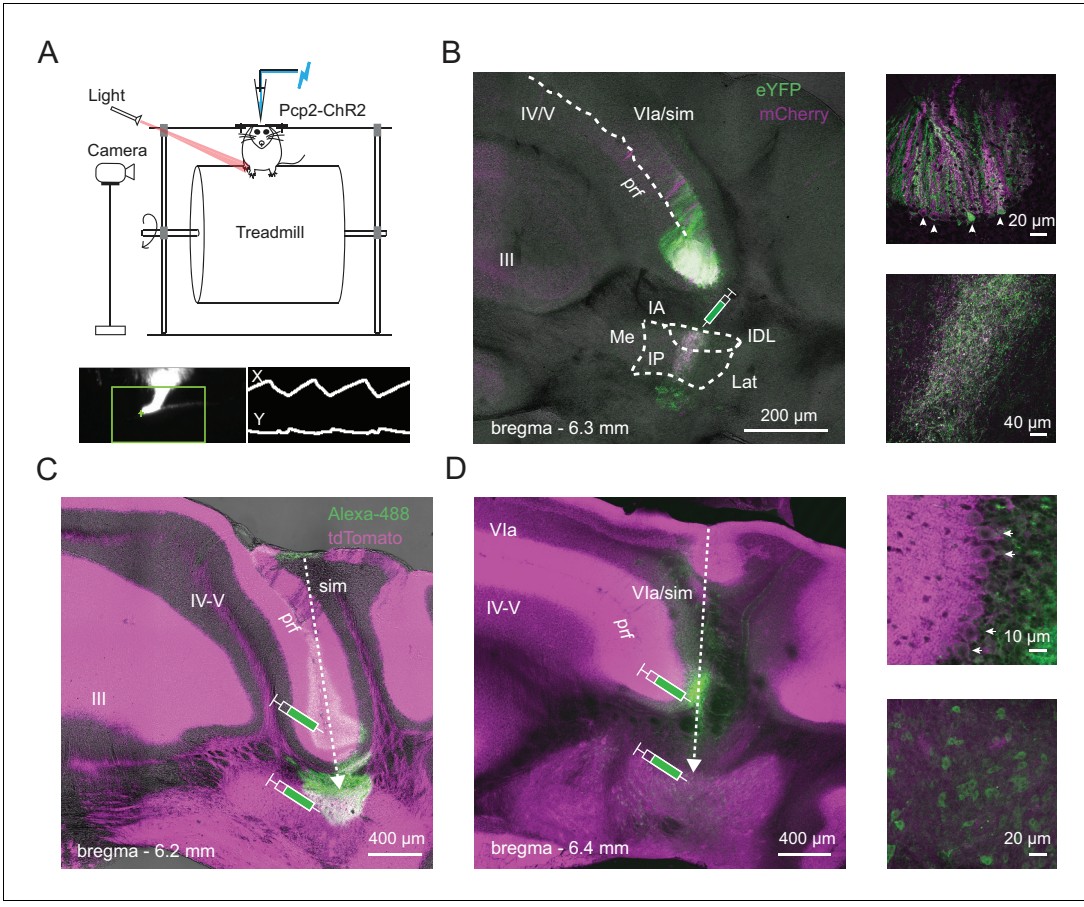

**Figure 1.** Identification of Purkinje and CbN cells involved in hindlimb movement during locomotion. (**A**) *Top,* Schematic of the setup including a head-fixed mouse running on the cylindrical treadmill, with paw movement recorded with an infrared camera. The patch pipette contained an electrode wire and an optical fiber. *Bottom.* Side-view of the ipsilateral hindlimb (*left*). A cursor (*green cross*) tracked the X and Y positions of tip of the paw within a marked ROI (*green box*). The paw record (*right*) was captured at ~ 240 frames/s. (**B**) *Left,* AAV2/9-mediated retrograde labeling of Purkinje inputs following injection of a viral mixture (ChR2-mCherry and ChR2-eYFP) in the interpositus nucleus of *Pcp2*-cre mice. Sample image of a coronal section showing an overlap (*white*) of mCherry (*magenta*) and eYFP (*green*) label in Purkinje cells of lobulus simplex. *sim*, simplex, *prf*, primary fissure, *IA*, interpositus anterior, *IP*, interpositus posterior, *IDL*, interpositus dorsolateral hump, *Lat*, lateral nucleus, *Me*, medial nucleus. *Upper right,* higher magnification image of labeled Purkinje somata (*arrows*) at the bottom of lobulus simplex surrounding the primary fissure. *Lower right,* higher magnification image of the site of injection in the nucleus interpositus posterior, showing labeled terminals of Purkinje afferents, but not CbN cell bodies. (**C**) and (**D**), *Left,* Sample images of coronal sections from two different mice. *Green,* Alexa 488-dextran amine injected at two sites along the recording track in mice expressing ChR2 and tdTomato (*magenta*) in all Purkinje cells (Ai27D x *Pcp2*-cre). *Dotted arrows,* injection pipette track. (**D**), *right.* Higher magnification images of the labeled Purkinje somata (*top; arrows*) and CbN cell somata (*bottom*) along the track in D.
DOI: https://doi.org/10.7554/eLife.29546.002

## Firing rates during rest and run

To obtain baseline information for subsequent optogenetic manipulation as well as for comparison to previous work, we began by recording action potentials from either Purkinje or CbN cells while mice ran on a non-motorized treadmill. Firing rates were analyzed in the subset of mice that both rested and ran during the recording period. Traces from a sample Purkinje and CbN cell, along with instantaneous firing rate and paw position, are shown in *Figure 2A* for rest (*upper panels*) and run (*lower panels*). At rest, Purkinje cells fired 93 ± 6 spikes/s (N = 35) and CbN cells fired 85 ± 6 spikes/s (N = 33; *Figure 2B*). Both these values are somewhat higher than spontaneous firing rates recorded in juvenile to weanling rodent cerebellar slices with all synaptic transmission blocked (Purkinje cells ~50 Hz,

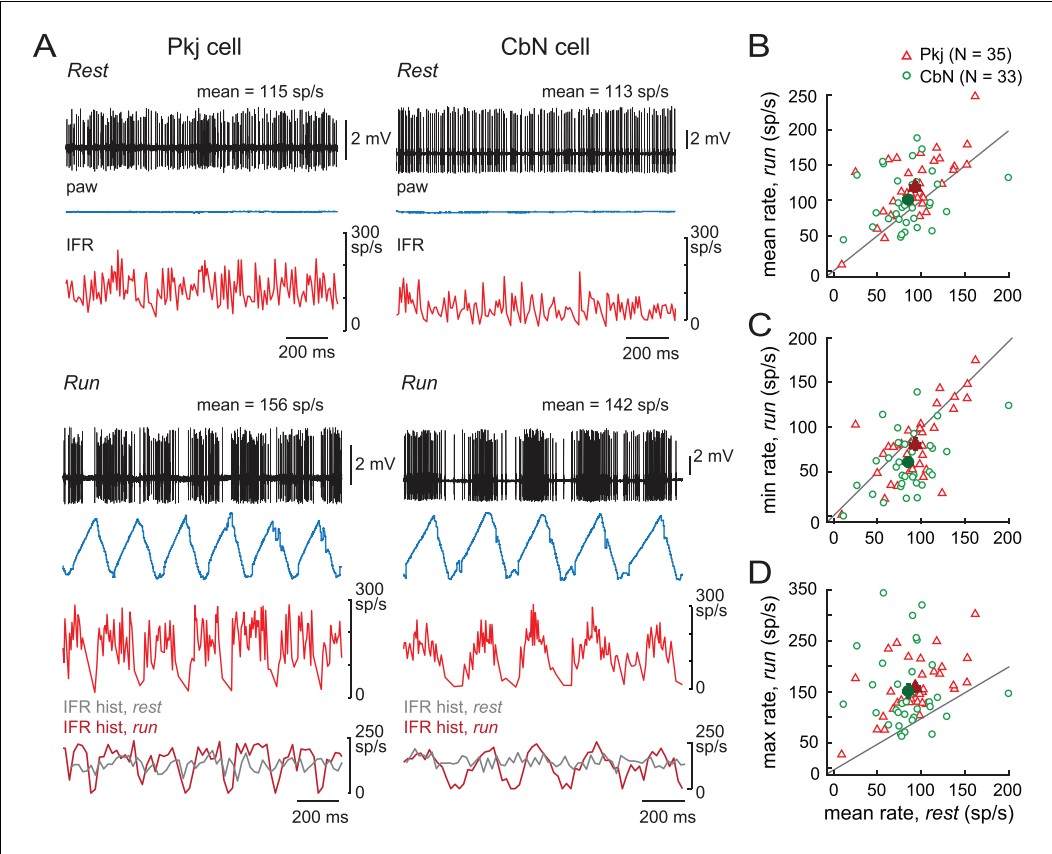

**Figure 2.** Running related changes in Purkinje and CbN cell firing rates. (**A**) Sample traces of loose cell-attached recordings from a Purkinje (Pkj) cell (*left*) or a CbN cell (*right*) in awake, head-fixed mice on a non-motorized treadmill during periods of rest (*top*) or run (*bottom*). *Paw,* ipsilateral hind paw position in the x-domain. *IFR,* instantaneous firing rate; *IFR hist,* histogram of the IFR in 20 ms bins, overlaid for rest and run. (**B, C, D**) Mean, minimal, or maximal firing rates during run periods across multiple strides *vs.* mean firing rate during rest (≥2 s) for all cells. *Open symbols,* individual cells; *filled symbols,* means; *grey lines,* unity.

DOI: https://doi.org/10.7554/eLife.29546.003

*Khaliq et al., 2003*; *Monsivais et al., 2005*; for CbN cells of male mice, ~70 Hz, *Mercer et al., 2016*). During running, mean firing rates for both Purkinje and CbN cells both increased equivalently compared to rest (*Figure 2B*; mean run rate Pkj, 119 ± 8 spikes/s; CbN, 101 ± 7 spikes/s, rest *vs.* run, Pkj, p<0.001; CbN, p=0.06, *paired t-test*; Pkj *vs.* CbN run-rest difference, Pkj, 26 ± 6 spikes/s; CbN, 16 ± 8 spikes/s, p=0.3, *Wilcoxon rank sum test*), suggesting that both classes of cells undergo a net increase in excitation during locomotion.

## Firing rate modulation with stride

In both cell types, firing rates were modulated during the stride (*Figure 2A*, *lower panels*), consistent with previous work in cats on motorized treadmills moving at a fixed rate and for Purkinje cells in freely moving rats (*Armstrong and Edgley, 1984a*, *Armstrong and Edgley, 1984b*; *Sauerbrei et al., 2015*). Superimposing histograms of the firing rate during running and rest (20 ms bins) revealed that firing was modulated both above and below the mean rest firing rates for the same cell (*Figure 2A*, *bottommost plots*). Most Purkinje and CbN cells showed bidirectional modulation of the instantaneous firing rates, with minima below rest (Pkj, 81 ± 7 spikes/s, p=0.05, CbN, 61 ± 6 spikes/s, p=0.0014, *Wilcoxon rank sum test, one-sided, Figure 2C*) and maxima above rest (Pkj, 159 ± 9 spikes/s, p<0.001; CbN, 152 ± 13 spikes/s, p<0.001, *Wilcoxon rank sum test, one-sided, Figure 2D*).

Next, we quantified the extent of firing rate modulation and its relative phase to the stride. Since locomotion was voluntary, the stride duration of the mouse on the treadmill varied with running speed, step length, or other alterations of gait, but left-right limb alternation was preserved as in freely walking mice (*Bellardita and Kiehn, 2015*; *Machado et al., 2015*). The stride duration ranged from 140 ms for the briefest strides (i.e., fastest locomotion) to 419 ms for the longest strides (mean, 222 ms; S.D., 71 ms, N = 9091 strides). Stride durations, along with the associated firing rates of the recorded neuron, are illustrated for one Purkinje cell (*Figure 3A and C*) and one CbN cell (). The modulation of firing rates of Purkinje and CbN cells during different stride durations was examined by separating each stride into its four components: the *lift* of the paw, the *swing* of the paw forward, the *plant* of the paw, and the *stance* as the paw moves backward on the treadmill. Aligning strides to the lift revealed that, despite variations in stride duration, firing rates tended to rise and fall at consistent phases of the stride for both Purkinje and CbN cells (*Figure 3C, 3DFigure 3C and D*),

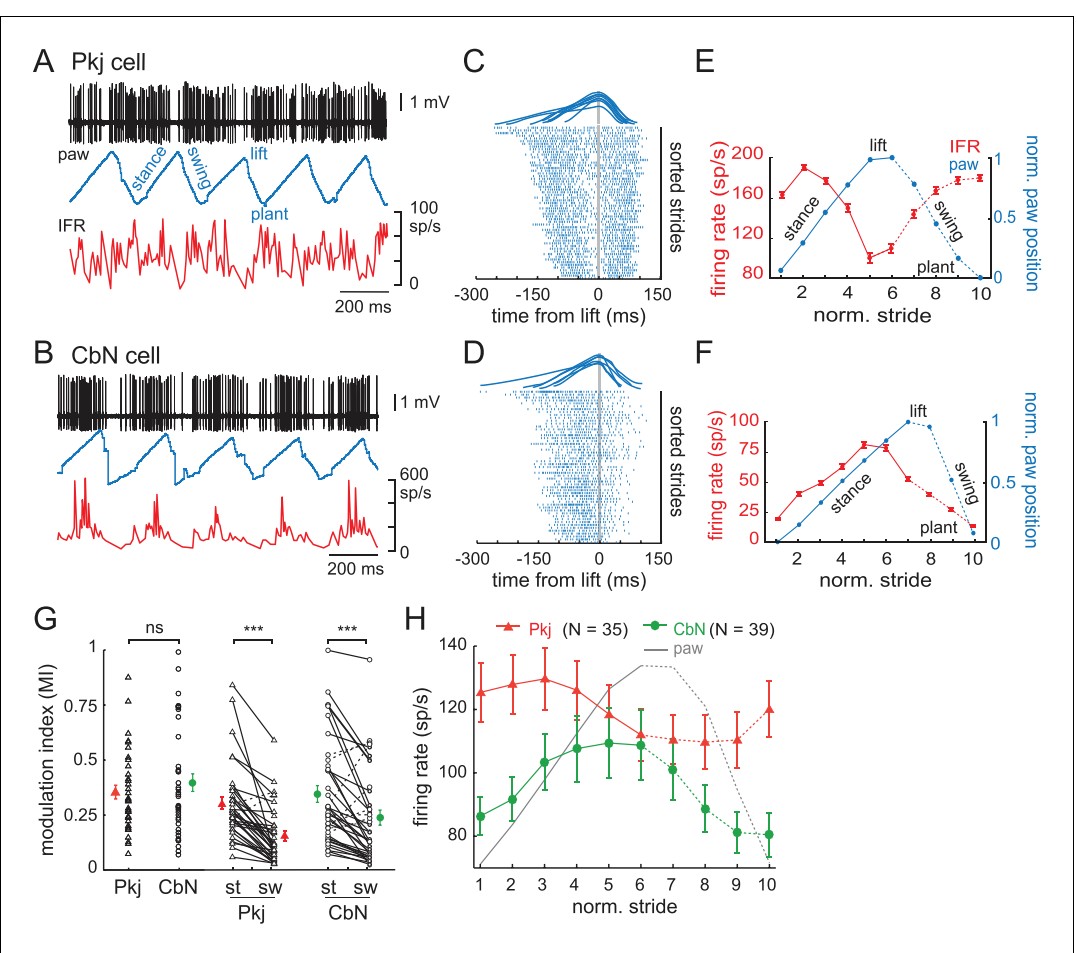

**Figure 3.** Modulation of firing rates relative to the phase of strides. (A) Sample records from a running mouse of Purkinje cell spikes, paw position, and instantaneous firing rate, illustrating *stance, swing, lift, plant*, component phases of a stride. (B) As in (A), for a CbN cell. (C) *Top,* sample strides aligned to lift. Raster plots of firing by the Purkinje cell in (A) during strides sorted by duration and aligned to the lift phase. Every third stride of 171 strides is plotted. (D) As in (C), for the CbN cell in (B). Rasters during every third stride of 159 strides. (E, F) *Red,* binned instantaneous firing rates averaged across all lift-aligned strides vs. normalized stride bin. *Blue,* paw position averaged across all normalized strides. Data from the Purkinje cell in (A, C) and the CbN cell in (B, D). (G) Modulation indices (MI) for all cells calculated from the mean IFR curves as in (E, F). *Open symbols,* individual cells, *filled symbols,* means; *st,* stance phase; *sw,* swing phase. (H) Population average of all IFR tuning curves across all Purkinje and CbN cells, superimposed on paw position. *Solid line,* stance; *dotted line,* swing. On all panels, all figures, *ns,* non-significant, *p<0.05, **p<0.01, ***p<0.001.
DOI: https://doi.org/10.7554/eLife.29546.004

indicating that the phase relationship between firing and the stride did not greatly change with speed. Therefore, to analyze the changes in firing rate over the course of the step cycle, we normalized the duration of strides aligned to the lift by dividing the stride into a total of ten bins before (stance) and after (swing) the lift (); eliminating the longest or shortest strides did not alter these plots, justifying collapsing the data across stride durations. The mean instantaneous firing rate per bin was calculated for each stride and averaged across all strides. These firing rates were plotted, along with normalized paw position, against normalized stride time (; Materials and methods). We refer to this change in instantaneous firing rate on the time scale of the stride (usually 200–300 ms) as 'stride-related modulation'.

The depth of stride-related modulation on a cell-by-cell basis was estimated as a modulation index (MI), defined as the ratio of maximal change in firing rate to the sum of the maximal and minimal firing rates, (max − min)/(max + min). An MI of 0 thus indicates no modulation and an MI of 1 indicates maximal modulation. Purkinje and CbN cells showed similar degrees of modulation for the whole stride (*Figure 3G*, *left*; Pkj, 0.4 ± 0.03, N = 35; CbN, 0.4 ± 0.04, N = 39; p=0.7, *Wilcoxon rank sum test*). As an alternative test of modulation, we analyzed the individual spike times during the step cycle for all strides in a given recording with Kuiper's tests. All cells demonstrated a significant deviation from a uniform distribution of spike times throughout the step cycle, confirming stride-related modulation of firing rates (Pkj, N = 35; CbN, N = 39; p<0.001, all cells). Because the extent of modulation often appeared different in the stance and the swing, as in previous studies (*Armstrong and Edgley, 1984a*, *Armstrong and Edgley, 1984b*; *Orlovsky, 1972a*), we compared the MI for the stance alone ($St_{MI}$) to that for the swing alone ($Sw_{MI}$). Most Purkinje and CbN cells were modulated to a greater extent in the (hindlimb) stance than in the swing (*Figure 3G*, *right*; Pkj, $St_{MI}$ = 0.3 ± 0.03 *vs.* $Sw_{MI}$ = 0.2 ± 0.02; CbN, $St_{MI}$ = 0.3 ± 0.04 *vs.* $Sw_{MI}$ = 0.2 ± 0.03, p<0.001 both within-cell comparisons, *Wilcoxon signed-rank test*); note that hindlimb and forelimb stance and swing are out of phase with each other (*Machado et al., 2015*).

To investigate the phase relationship between stride-related modulation in Purkinje and CbN cells, we averaged the activity of all Purkinje neurons or all CbN neurons. Although the population of neurons was clearly heterogeneous, as described below, this measure permits comparison to related studies. The mean data showed that the Purkinje cell population had firing rates that peaked earlier in the stance phase of the stride than did the CbN cell population (*Figure 3H*). Since most strides were 200–300 ms long, the phase advance of approximately two bins corresponds to ~ 40–60 ms, considerably longer than a few synaptic delays. The 90° shift (i.e., one quarter of a stride) remained when the firing rates were normalized for each cell to avoid excessively weighting neurons with higher firing rates. This phase relationship differs somewhat from a similar analysis of forelimb-related Purkinje and CbN cells in cats walking at fixed rates on a motorized treadmill, which revealed an in-phase relationship for modulated Purkinje and CbN cell firing rates and greater activity on or just before the swing phase (*Armstrong and Edgley, 1984a*, *Armstrong and Edgley, 1984b*). While several differences between the studies, including species, the limb monitored, whether the stride was fixed or not, and likely the specific recording regions, might contribute to the different average phase relationships, both the present and the previous study provide evidence against a preponderance of anti-phase relationship between the averaged activity of Purkinje and CbN cells in regions that are likely to contain connected cells.

Moreover, the individual Purkinje and CbN neurons actually showed a wide range of firing patterns, with different cells showing peak firing rates at different phases of the stride. To test whether individual cells clustered into in-phase or antiphase activity patterns, we provisionally divided the cells into six discrete categories or 'modulation classes' based on the relative phase between the peak of activity and peak position of the ipsilateral hind paw: *Class I*, in phase, with activity increasing in the stance and decreasing in the swing; *Class II*, anti-phase, with activity decreasing in the stance and increasing in the swing; *Class III*, formally defined as activity leading the step cycle by 90°, but experimentally evident as activity first rising and then falling in stance; *Class IV*, formally defined as lagging the step cycle by 90°, but experimentally evident as first falling and then rising in stance; *Class V*, two peaks at ± 90° from the lift, rising then falling in both stance and swing, and *Class VI*, two troughs at ± 90° from the lift, falling then rising in both stance and swing. Sample data from cells in each category are illustrated in *Figure 4A* (Pkj) and *Figure 4B* (CbN). A subset of unclassifiable cells was grouped together in *Class VII*.

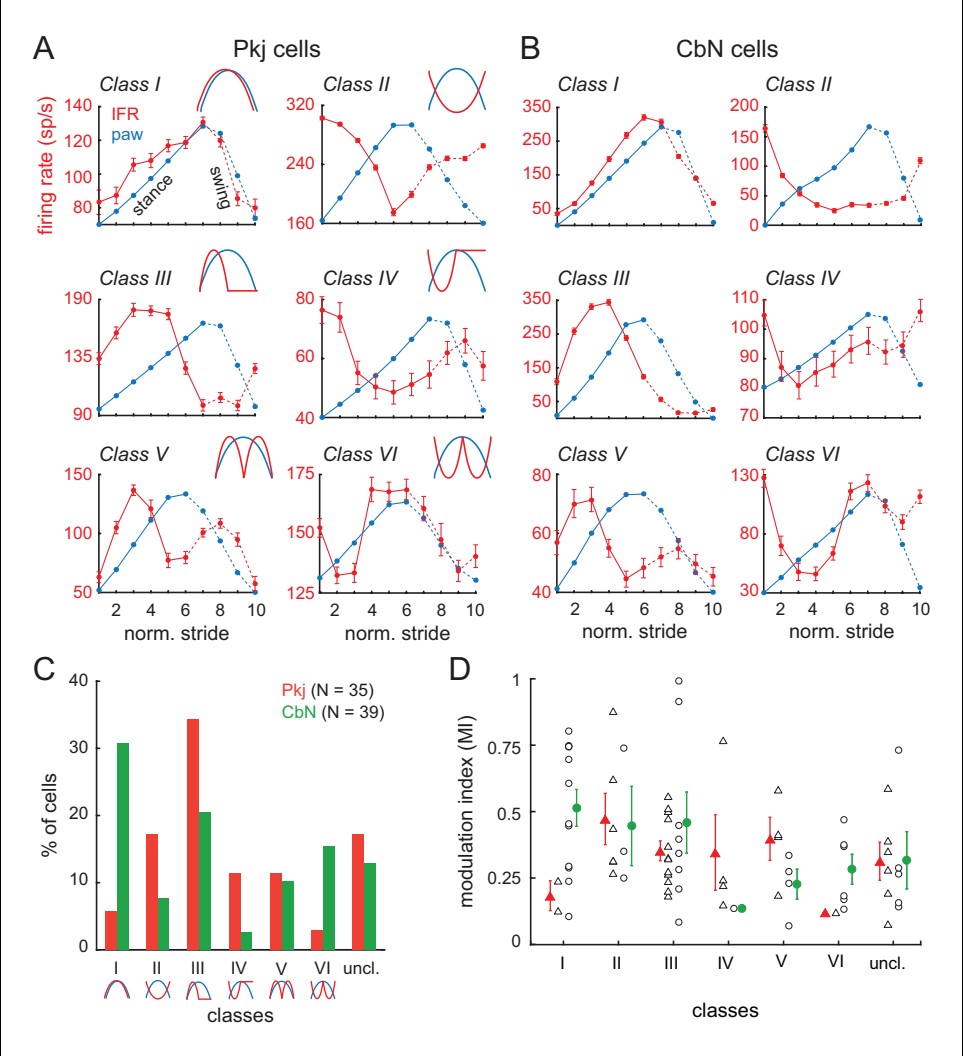

**Figure 4.** Multiple categories of stride-related firing rate modulation in Purkinje and CbN cells. (**A, B**) Instantaneous firing rates (*red, IFR*) and paw position (*blue, paw*) during normalized strides for six different Purkinje cells (**A**, Classes I-VI, N = 101, 208, 108, 77, 75, 118 strides) and CbN cells (**B**, Classes I-VI, N = 102, 67, 107, 95, 295, 54 strides). *Insets*, schematics of modulation pattern in each class. (**C**) Distribution of cells in each of the six classes. (**D**) Modulation indices sorted by class. *Triangles*, Purkinje; *circles*, CbN; *Open symbols*, individual cells, *filled symbols*, means; *uncl.*, unclassified.

DOI: https://doi.org/10.7554/eLife.29546.005

Neurons in each of the six categories were found in the population of both Purkinje and CbN cells (*Figure 4C*). Although certain patterns of activity occurred more frequently than others, for the task of well-learned, unperturbed, voluntary locomotion, there was no obvious prevalence of either in-phase relationships (i.e., similar class occupancy) or anti-phase relationships (i.e., occupancy of classes I and II, III and IV, or V and VI) for Purkinje and CbN cells. Instead, both in-phase and anti-phase categories were present. For instance, Class III was highly represented in both cell types (Pkj, 34%, CbN, 21%). In contrast, Class I was highly represented only in CbN cells (31%) with a correspondingly higher number of the anti-phase Class II Purkinje cells (17%) than in-phase Class I Purkinje cells (6%). Cells with distinct phase relationships could be located near one another; in cases in which two cells were recorded on the same penetration, different modulation categories were present in 5 of 6 Purkinje cell pairs and 3 of 5 CbN cell pairs. Additionally, the depth of stride-related modulation was comparable in most classes, for both Purkinje and CbN cells (*Figure 4D*). Together these data suggest that, although individual neuronal responses are distinct, the activity patterns across cells form

a continuum such that a subset of Purkinje and CbN cells elevate their activity at each phase of the stride.

## Alterations of gait induced by optogenetic activation of Purkinje cells

Next, we tested how perturbing CbN cell activity influenced voluntary, well-practiced locomotion. Changes in strides were monitored while ChR2-expressing Purkinje cells were optogenetically stimulated with either steps or trains of light. The choice of light patterns was informed by previous work demonstrating that CbN cells in brain slices or anesthetized mice respond differentially to inhibition from Purkinje cells activated synchronously as opposed to asynchronously (*Person and Raman, 2012a*). We reasoned that, if the degree of coherence of inhibitory inputs to CbN cells is a relevant variable in vivo, then differences in motor behavior might be evident when Purkinje cells are artificially stimulated to fire relatively more synchronously as compared to asynchronously. We predicted that light steps would increase firing rates of stimulated Purkinje cells, but with no systematic temporal correlation across cells, while trains would elicit spikes more or less simultaneously in the subset of illuminated cells. It is worth noting at the outset, however, that additional variables besides stimulus pattern might influence the experiments. For example, Purkinje cells that received optogenetic stimulation would also be subject to natural synaptic input, and CbN cells would in turn receive inhibition resulting from the summed synaptic and optogenetic drive to Purkinje cells as well as natural synaptic excitation. Since it was not possible to determine the effects of stimulation in advance, we tested a range of stimuli and interpreted the data in the context of these uncertainties.

Experiments were done on the same Purkinje and CbN cells characterized above. Once data from a neuron had been collected during voluntary rest or running, light was applied through an optical fiber included in the patch pipette. Each trial consisted of either a 1 s step or a 1 s train of illumination, with 3–10 trials per condition. Light stimuli were applied whether the mouse was running or stationary. The primary observation was that alterations of gait could be induced on a fraction of trials by optogenetic stimulation of Purkinje cells (by either steps or trains), whereas on other trials running was unimpeded.

Before analyzing the effects of steps and trains, we first examined the light-induced changes in strides. In trials with unimpeded locomotion, mice occasionally ran more rapidly. As long as strides remained regular, however, these were classified as 'non-slip' trials (*Figure 5A*). We quantified the attributes of these strides by automated stride detection (Materials and methods) that measured the plant-to-plant duration, the slope of the stance phase, and the slope of the swing phase of (1) the stride just before the light and (2) the stride during the light stimulus that differed most greatly from the pre-stimulus stride. A slope increase indicates a faster movement of the paw, that is, a more rapid gait. In 122 non-slip trials identified during recordings from CbN cells, the stride before *vs.* during the light had a duration of 288 ± 1 ms *vs.* 290 ± 3 ms (p=0.95, *paired t-test*), a swing slope of 8.0 ± 0.2 *vs.* 8.4 ± 0.2 AU/s (p=0.14, *paired t-test*, where 1 AU is the full forward-to-backward range of the paw, Materials and methods), and a stance slope of 5.4 ± 0.2 *vs.* 5.9 ± 0.2 AU/s (p=0.003, *paired t-test, Figure 5—figure supplement 1*).

In contrast, in other trials, strides were more substantially perturbed. These trials were classified as 'slip' trials, although this term does not imply a literal sliding motion but a deviation from regularity. All slip trials contained at least one stride that deviated by ≥20% from the last full stride preceding the light in at least one of the following ways: an increase in duration (a *prolonged* stride, *Figure 5B*), a decrease in duration (an *incomplete* stride, *Figure 5C*), or a decrease in either swing slope or stance slope (an *arrested* stride, *Figure 5D*). Of the 368 slips automatically identified with these criteria, 295 trials (80%) exhibited more than one of these attributes (an *altered* stride). The parameters of all slips are shown in *Figure 5—figure supplement 1*. In the 293 trials with prolonged strides, durations more than doubled, increasing by 128 ± 7% (from 244 ± 6 to 522 ± 16 ms, p<0.001, *paired t-test*). In the 97 trials with incomplete strides, durations dropped by 40 ± 2% (from 491 ± 44 ms to 233 ± 13 ms, p<0.001, *paired t-test*). In the 313 trials with arrested strides, the paw transiently stopped its usual 'sawtooth' movement, and the swing slope fell by 60 ± 2% (from 8.3 ± 0.2 to 3.2 ± 0.2 AU/s, 173 trials, p<0.001, *paired t-test*) and/or the stance slope fell by 50 ± 1% (from 5.5 ± 0.1 to 2.7 ± 0.1 AU/s, 232 trials, p<0.001, *paired t-test*); 92/313 arrested strides had reductions in both stance and swing slopes.

Comparable measurements were made for 323 non-slips and 432 slips evoked while recordings were made from Purkinje cells. Although the optical fiber was in a different position relative to the

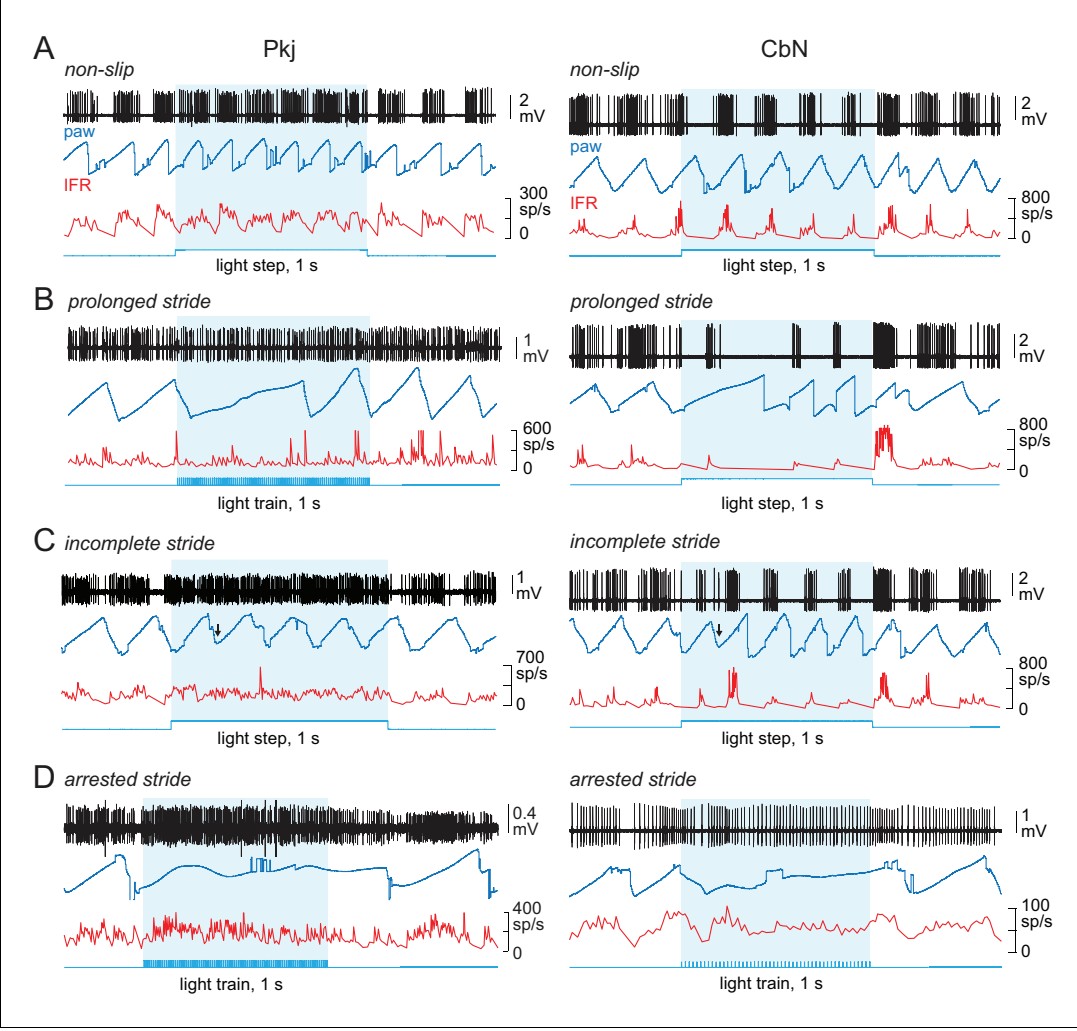

**Figure 5.** Effects of optogenetic stimulation of Purkinje cells on locomotion. (A-D) Sample recordings from a Purkinje (*left*) and CbN cell (*right*), with paw position, instantaneous firing rate, and stimulus pattern, for non-slip trials (A) and slip trials, including prolonged strides (B), incomplete strides (C), and arrested strides (D). *Blue shading*, light stimulation. *Arrows* in (C) indicate the time of the slip.

DOI: https://doi.org/10.7554/eLife.29546.006

The following figure supplement is available for figure 5:

**Figure supplement 1.** Parameters of slips.

DOI: https://doi.org/10.7554/eLife.29546.007

CbN cell recordings, the effects on locomotion were similar; for non-slips, the stride duration was unchanged (from $310 \pm 11$ to $322 \pm 19$ ms, p=0.6, *paired*) although running again tended to accelerate (swing slope, from 7.9 to 8.6 AU/s, p<0.001 *paired*; stance slope from $5.2 \pm 0.1$ to $5.4 \pm 0.1$ AU/s, p=0.15, *paired*). For prolonged strides, stride duration increased by $133 \pm 9\%$ (N = 315, p<0.001, *paired*); for incomplete strides, stride duration decreased by $40 \pm 2\%$ (N = 145, p<0.001, *paired*); for arrested strides, swing slope decreased by $57 \pm 2\%$ (N = 263, p<0.001, *paired*) and stance slope decreased by $52 \pm 1\%$ (N = 215, p<0.001, *paired*; 368 total arrested strides; 110 with both swing and stance slope reduced). Of the 432 slips, 337 trials (78%) had changes in more than one parameter and were therefore described as altered strides.

In the 315 trials in which a clear onset of the slip could be unambiguously detected, the latency to slip was ~ 120 ms, regardless of the pattern of optogenetic stimulation or the phase of the stride (steps: $120 \pm 17$ ms, N = 125 trials; trains $117 \pm 13$ ms N = 190 trials). Along with (tracked) hindlimb deviations during slips, however, irregular movements of the (untracked) forelimb and trunk were

sometimes also noted, consistent with cerebellar control of whole-body coordination, in which movements of different body regions are correlated but can be phase-shifted (*Machado et al. 2015*). Because the hindlimb did not necessarily initiate slips, the latency of ~ 120 ms places an upper bound on the lag between light onset and the first errant movement.

Control experiments demonstrated that slips of any kind during locomotion were rare when light stimuli were applied to two *Pcp2*-cre mice not expressing ChR2. Without ChR2, slips occurred on 1.3% of trials for 1.7 mW steps (1/77 trials), as compared to 93.9% for the same intensity in ChR2 expressing mice (31/33 trials, analyzed in detail below). These data indicate that slips are not visual responses to light and instead support the idea that most slips can be ascribed with confidence to the optical stimulation of Purkinje cells with ChR2.

As mentioned above, for a cell to be included in the study, a light-induced perturbation had to be observed in at least one trial with at least one step or train of any intensity or frequency (Pkj, N = 35, CbN, N = 39). In all cells, as many stimulus conditions as possible were tested. Because mice were running voluntarily, however, occasionally they would stop before or during a stimulation without resuming running afterward. These trials were excluded from the analyses, and consequently not every condition is represented in every cell. It was also necessary to control the range of the stimuli tested to ensure that slip occurrence was rare; if slips were evoked too frequently, mice tended to stop running. Therefore, if the same stimulus repeatedly evoked slips, further replications were not carried out. This constraint limited the number of slip-inducing trials that could be obtained, especially with highly effective stimuli. Often, however, slip and non-slip trials were evoked with the same stimulus condition. In these cases, slips did not cluster to the beginning or end of the series, consistent with the desired result of maintaining the stimuli near a threshold for evoking slips. The threshold nature of the response is also consistent with previous demonstrations that correlated Purkinje cell activity can vary from stride to stride (*Sauerbrei et al., 2015*); the light-evoked depolarization may therefore superimpose differently upon naturally occurring synaptic input on different trials. The observation that mice often continued running through the light stimulus also suggests that the array of stimuli led to spike rates that were within the natural range, rather than consistently pushing cerebellar output to an extreme of rapid firing or silencing. Subsequent analyses examined the relationship between light stimuli, neuronal activity, and behavior.

## Effects of optogenetic step and train stimuli on Purkinje and CbN cell firing

We first examined how the firing rates of Purkinje and CbN cells were changed by the different patterns of optogenetic stimulation. The firing rates of recorded neurons were measured during several levels of illumination (0.56, 0.65, 0.87, 1.7, and 2.6 mW); for both Purkinje and CbN cells, responses at the lowest two levels were indistinguishable, and the data were pooled. Consistent with ChR2-mediated depolarizing drive to Purkinje cells, light steps elevated Purkinje firing rates and reduced CbN firing rates throughout the 1 s stimulus. Example traces of spikes and instantaneous firing rate plots for onset of stimulation are shown in *Figure 6A and B* (*top panels*), along with the changes in rate over the full 1 s plotted for each cell as PSTHs (50 ms bins, *bottom panels*). Averaging the PSTHs across all 35 Purkinje cells showed that the elevation in firing rate increased with stimulus strength (*Figure 6C*). The time course of the effect was similar across all stimulus levels. The mean response of all trials in all Purkinje cells showed an initial transient increase in firing rate that decayed slightly at the highest strengths to a plateau that remained above the pre-stimulus 'baseline' firing rate for the duration of the step (*Figure 6C*). Likewise, the mean response averaged across all trials in all 39 CbN cells showed that firing was strongly suppressed in the first ~100 ms, followed by a transient restoration of firing, but rates remained below baseline for the remainder of the step (*Figure 6D*). Overall, the stimuli evoked firing rates that covered a wide portion of the dynamic range above baseline for Purkinje cells and below baseline for CbN cells (*Figure 6E*). CbN cell firing rate was often elevated after the offset of light steps (rate for 200 ms before *vs.* after stimulation, 89 ± 2.2 spikes/s *vs.* 106 ± 4.5 spikes/s, N = 161 trials p<0.001, *Wilcoxon signed-rank paired test*), consistent with previous studies in which Purkinje cell firing was optogenetically activated (*Lee et al., 2015*; *Witter et al., 2013*).

The effects of light trains (50-, 100-, 150-, and 200 Hz 1 ms pulses at 1.7 or 2.6 mW) were distinct from those evoked by steps. As shown for a 100 Hz train, Purkinje cell action potentials were often time-locked to each light pulse, and CbN cell spikes tended to occur between light pulses

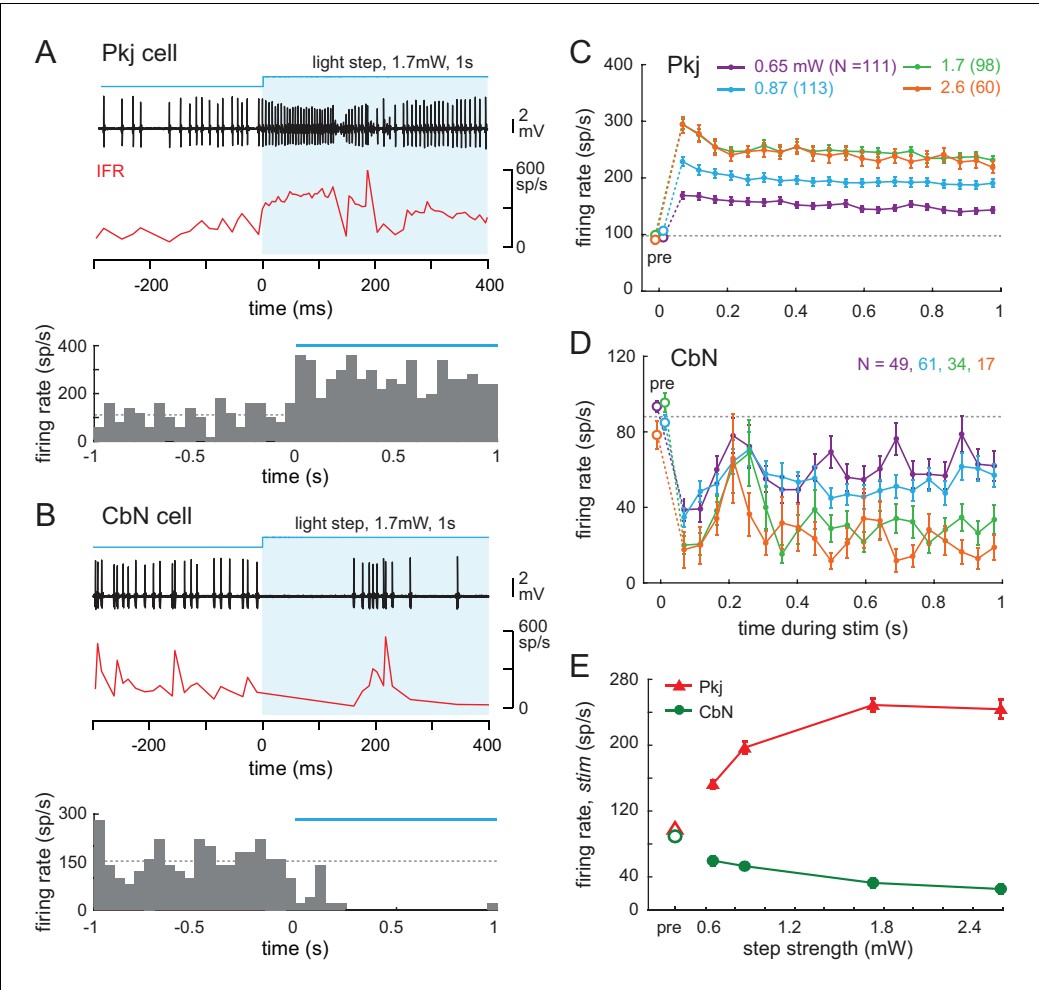

**Figure 6.** Effects of light steps on Purkinje and CbN cell firing. (**A, B**) *Top,* Sample traces from a Purkinje (**A**) and CbN cell (**B**) in response to the first 400 ms of a light step (*blue shading*). Instantaneous firing rate (*red, IFR*). *Bottom,* PSTH of the examples in (**A, B**) for the full 1 s of stimulation. Light onset, 0 ms. *Bin width,* 50 ms; *blue line,* light duration, *dotted line,* mean pre-stimulus firing rate for that trial (1 s). (**C, D**) Mean PSTHs across all trials for all step intensities for all 35 Purkinje (**C**) and all 39 CbN cells (**D**) *Bin width,* 50 ms; *pre,* pre-stimulus rates; *dotted line,* mean pre-stimulus rates in all conditions. (**E**) Mean firing rates over the 1 s stimulus for all cells at each stimulus strength.

DOI: https://doi.org/10.7554/eLife.29546.008

(*Figure 7A, 7BFigure 7A and B,* top panels). The precision of response timing throughout the 1 s stimulation was examined by plotting spike times as PSTHs (1 ms bins) relative to the onset of each pulse in the train. The illustrated Purkinje cell showed a well-timed threefold increase in spike probability, and the CbN cell showed a well-timed drop to 0 spike probability. This temporal structure was consistent throughout the 1 s train, as shown by grouping all light pulses for the early, middle, and late phases of the stimulation window (*Figure 7A, 7B,* bottom panels). Averaging the PSTHs across all 35 cells and all trials showed that Purkinje spike probability peaked 2–3 ms after stimulus onset, dropping back to near pre-stimulus values by 4 ms (*Figure 7C*). At higher frequencies, the mean elevation of spike probability from pre-stimulus values was lower, consistent with the relatively slow ChR2 currents not evoking a spike after every light pulse, but Purkinje cell spikes continued to time lock (p<0.001 at each frequency, *Rayleigh test*). This time-locked pattern of response was consistent across all Purkinje cells. Since the optical stimulation was not limited to a single neuron, the light trains likely had the desired effect of increasing the degree of synchrony of Purkinje cells. In contrast

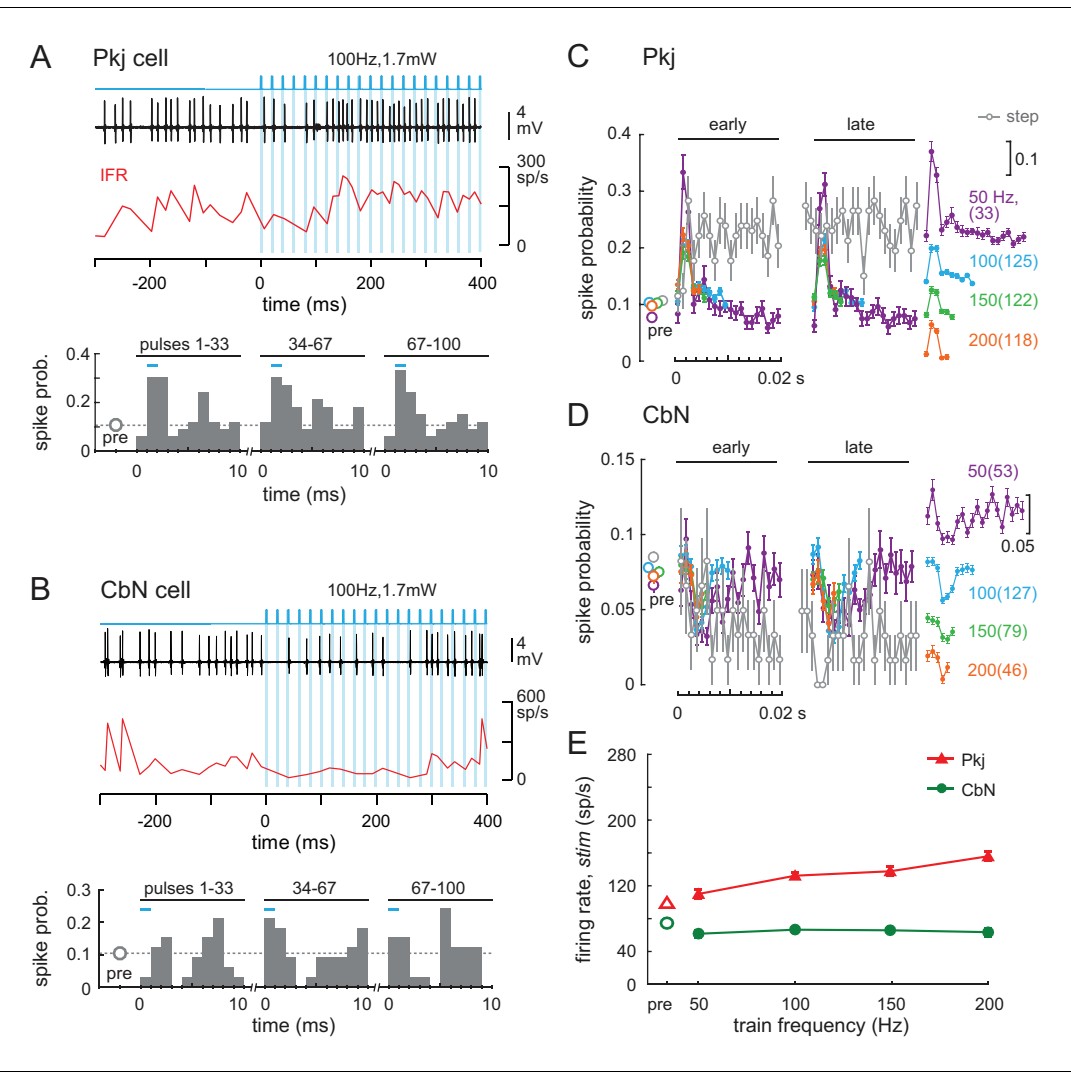

**Figure 7.** Effects of light trains on Purkinje and CbN cell firing. (**A, B**) *Top,* Sample traces from a Purkinje (**A**) and CbN cell (**B**) in response to the first 400 ms of a 100 Hz light train (*blue shading*). Instantaneous firing rate (*red, IFR*). *Bottom,* Mean PSTH across cycles for the early (1-33), middle (34-67) and late (68-100) pulses in the train, showing resetting of spike times. *Bin width,* 1 ms; inter-pulse interval, 10 ms; *pre,* mean spike probability per 1 ms over a 1 s pre-stimulus epoch. (**C, D**) Mean PSTHs across all 35 Purkinje and 39 CbN cells for all train frequencies for *early* and *late* cycles analyzed as in (**A**) and (**B**). *Far right, inset,* PSTH traces from the early group at different frequencies are offset for clarity (scale = 0.1 change in spike probability). *Grey open circles,* PSTH of responses to light step (0.87 mW) from ***Figure 6***, binned at 1 ms and overlaid for comparison; *dotted grey line,* mean pre-stimulus firing rates for all train conditions. (**E**) Mean firing rates over the 1 s stimulus for all cells at each frequency.
DOI: https://doi.org/10.7554/eLife.29546.009

The following figure supplement is available for figure 7:

**Figure supplement 1.** Local field potentials of Purkinje cell responses to optical stimulation.
DOI: https://doi.org/10.7554/eLife.29546.010

to the response to trains, when the responses to steps were binned at 1 ms, the spike probability remained elevated across all bins, throughout the step (***Figure 7C***, *grey*).

CbN cell spikes also time-locked to the stimulus at all frequencies, briefly dropping their firing rates during the window between 2 and 6 ms after light pulse onset (***Figure 7D***, p<0.001 at each frequency, *Rayleigh test*). As in Purkinje cells, spike probability deviated less from pre-stimulus values at higher frequencies, but the resetting of spike times remained evident. Since each CbN cell

receives convergent input from 30 to 50 Purkinje cells (e.g., *Person and Raman, 2012a*), the consistent gaps in firing indicate that a measurable proportion of converging Purkinje cells must have provided coherent inhibition at regular intervals that matched the stimulus train frequency. The observation that similar response patterns were present across all 39 cells, for all trials, at each train frequency made it reasonable to conclude that the trains led to multiple Purkinje cells firing action potentials in the same 2–3 ms time window, and therefore that they increased the synchrony of Purkinje cell firing. For CbN cells, like for Purkinje cells, superimposing the responses to steps plotted in 1 ms bins confirmed that the temporal structure of spiking imposed by trains was distinct from the lack of structure imposed by steps. Specifically, after the first 10 ms, the spike probability fell below ~0.05 per 1 ms bin throughout the stimulation, consistent with steps generating asynchronous Purkinje cell firing. Trains had a smaller effect on firing rates than did steps. As train frequencies increased from 50 to 200 Hz, Purkinje cell firing rates increased from about 110 to 160 spikes/s, whereas rates of CbN cells remained at about 60–70 spikes/s at all train frequencies (*Figure 7E*).

Because the optical fiber was inside the recording pipette, however, the fiber was necessarily in different locations during the recordings from Purkinje and CbN cells. To obtain a measure of Purkinje cell activity with light stimulation applied in the locations associated with CbN recordings, we broke the tip of the electrode to be ~15 µm diameter, positioned the fiber at the depths that it reached during CbN cell recordings, and recorded local field potentials from Purkinje cells. This method had the added advantage of permitting simultaneous recordings from cells activated by the light stimulation, providing a direct investigation rather than just the inference of synchrony. Recordings were made in anesthetized mice to facilitate data acquisition, since testing whether high-frequency stimulation of ChR2 elicited well-timed Purkinje spikes simultaneously in multiple cells did not require awake animals. Light steps led to a brief period of coherent high-frequency firing that lasted ~50 ms; for the next ~950 ms, evidence of synchronous firing was not detectable (*Figure 7 Supplement 1AFigure 7—figure supplement 1A*). In contrast to the responses to steps, light trains led to coherent spiking at frequencies that matched the train and that lasted throughout the 1 s stimulus (*Figure 7—figure supplement 1B–E*). Consistent with the change in spike rates seen in the PSTHs from CbN cells in awake mice, the amplitude of the signal was highest for 50 Hz trains, illustrating that Purkinje cells followed the stimulus most effectively at the lowest frequency tested. Even with high-frequency trains, however, FFTs of the local field potentials in both mice showed peaks at the stimulation frequency and harmonics for all trains, verifying that some Purkinje cells fired synchronously on each cycle of the stimulus across the range of frequencies tested (*Figure 7—figure supplement 1F*).

Thus, the consistent time-locking of Purkinje and CbN cell firing to trains, as well as the local field potentials, confirmed that light trains could be used to restructure the temporal pattern of firing of stimulated Purkinje cells in a way that would (1) differ from firing patterns induced by steps, and (2) favor coherent firing by Purkinje cells, that is, increase the degree of synchrony of spikes. It is worth noting that the resetting of spike times in the CbN cells occurred on a time scale of 5–20 ms, a much shorter period than the duration of a stride.

## Relation between firing rates and stride cycle during optogenetic stimulation

Before the recorded CbN cell activity could be related to normal and perturbed locomotion, it was necessary to test the extent to which activity of an individual CbN cell could be correlated with movement, since the firing by a single cell could not be assumed to give an accurate readout of behavior. Therefore, we first pooled results from steps and trains and compared neuronal activity during *non-slip* trials during stimulation ('light on') to control strides when no stimulation was applied ('no light'). The same analysis, that isi.e., binning and averaging instantaneous firing rates across all strides, was applied to light-on and no light conditions. As shown for a CbN cell (*Figure 8A*), even though light changed the firing rates during non-slip trials, the spike rates rose and fell at the same phase of the stride as for control strides. The binned firing rates during the stride for the light-on and no-light conditions were plotted, as shown for a Purkinje and a CbN cell (*Figure 8B and C*, *left*), and the correlation between light-on and no-light binned firing rates was calculated by linear regression (*Figure 8B and C*, *right*). On non-slip trials, most Purkinje and CbN neurons had high positive correlations between the light-on and no-light conditions (*Figure 8D*), with $\geq$ 70% exceeding r = 0.4 (Purkinje cells, N = 21/30; CbN cells, N = 15/20). Thus, despite changes in mean firing rate across

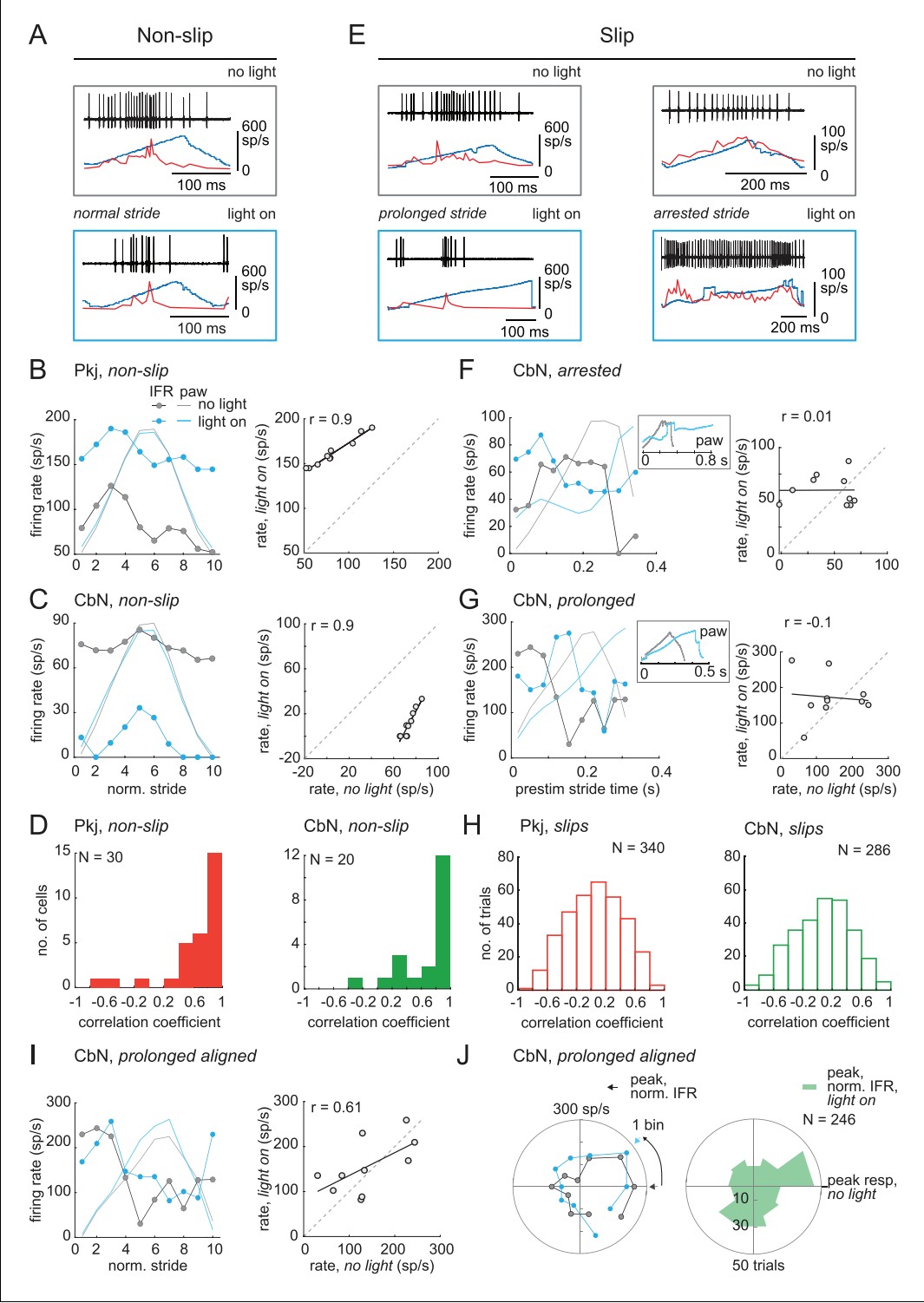

**Figure 8.** Correlation between firing rate modulation of individual neurons and slip/non-slip trials. (**A**) Comparison of CbN cell activity (*black*), paw position (*blue*), and instantaneous firing rate (*red*) without light (*top*, 'no light') and during light stimulation (*bottom*, 'light on') in non-slip trials. (**B, C**) *Left*, Instantaneous firing rate (IFR) and paw position during normalized strides for a Purkinje (**B**) and a CbN cell (**C**) for non-slip trials. *Right*, Binned IFRs from left panels for 'light on' *vs.* 'no light' conditions; *r*, correlation coefficient; *dotted line*, unity. (**D**) Histogram of correlation coefficients measured as in (**B**), (**C**) for all Purkinje (*left*) and CbN (*right*) cells. (**E**) As in (**A**) for two slip trials. Note differences in time base between 'no light' and 'light on' pairs. (**F, G**) As in (**C**), for slip trials for the

*Figure 8 continued*

illustrated CbN cells. *Insets,* overlay of paw traces during the pre-stimulus (*grey*, 'no light') stride template and the slip epoch (*blue,* 'light on'). (H) As in (D), for all slip trials. (I). As in (C) for prolonged stride aligned to the lift of a control stride ('no light'), for the same slip trial as shown in (G). (J) *Left.* Polar plots from (I) of control and prolonged lift-aligned stride. 0 indicates bin with maximal firing rate in control stride. *Right.* Mean of all prolonged lift-aligned stride trials, showing the peak response is near 0, which is the bin with the peak firing rate in control ('no light').

DOI: https://doi.org/10.7554/eLife.29546.011

the duration of each stride, on non-slip trials, the phase relation between the firing rate of individual cells and the step cycle was relatively unperturbed. In other words, normal stride-related modulation was preserved on non-slip trials.

Next, we compared spiking during *slip* trials between light-on and no-light conditions, with a goal of testing the extent to which changes in stride-related modulation of firing in individual cells and slips were mutually predictive. Sample traces are shown for different CbN cells during an arrested stride and a prolonged stride (*Figure 8E*). Because each slip was distinct, these analyses were done on a trial-by-trial basis. Also, since by definition strides were disrupted during slips, firing rates could not be binned against a clearly defined concurrent stride. We reasoned, however, that if the slip and the activity of the recorded neuron were independent, then the pre-stimulus firing rate modulation should persist throughout the stimulation. We therefore tested whether firing rates during slips continued to be modulated as they were during unperturbed locomotion, by binning firing rates during the slip against a template stride just preceding light onset (Materials and methods). The binned firing rates for the light-on and no-light conditions were plotted, as illustrated for the arrested and prolonged stride (, *left*), and correlations were calculated (*Figure 8F, 8GFigure 8F and G*, *right*). Unlike for non-slip trials, a low proportion of perturbed strides showed correlations $\geq$0.4 (Purkinje cells, N = 69/340, 20% trials; CbN cells, N = 60/286, 21% trials, *Figure 8H*, *open bars*), indicating that disruption of firing rate modulation of individual cells and disruption of strides were tightly associated.

The prolonged strides provided a special case for examining this relationship, as inspection of the data suggested that firing rate modulation with respect to the unperturbed stride might be maintained but extended to match the lengthened stride. Therefore, prolonged strides were aligned to the lift and binned, and the correlation between light-on and no-light conditions was examined (*Figure 8I*). To quantify this effect across trials, the binned no-light and light-on firing rates were plotted on polar coordinates (*Figure 8J*, *left*), with 0° indicating the bin with the peak firing rate for the control stride ('no light'). The distribution for all prolonged strides is shown in *Figure 8J* (*right*). The skew toward the right half of the plot suggests that the maximal firing rates were achieved in the same phase (stance or swing) of the prolonged and control strides (Pkj, N = 277, p=0.002; CbN, N = 246, p=0.002, *Rayleigh test*). Together, the data suggest that modulation of firing rates individual Purkinje and CbN cells correlates well with the regularity of the stride cycle: when modulation persists during optogenetic stimulation, strides continue relatively normally, whereas when modulation is disrupted, slips occur. The most plausible explanation for this result seems to be that direct stimulation of Purkinje cells evokes similar responses (of altering or failing to alter modulation) in multiple CbN cells, which together exert an influence over locomotion that is ultimately manifested in hind paw movement. The association between slips and the loss of stride-related modulation in single cells also justifies an analysis of the ability of light steps and trains to disrupt single-cell firing and perturb movements.

With this information in hand, we tested whether CbN cells responded differentially to steps and trains (asynchronous and synchronous inhibition) in a way that could be related to the measured behavior. We first compared the effects of light steps of different strengths on CbN cell firing rates during slip and non-slip trials. The mean firing rate during the 1 s light-on stimulation period was plotted against the firing rate for 1 s just before stimulation on each trial (*Figure 9A–9D*, every trial from N = 13, 18, 10, 7 cells). In all trials, since the mouse was running before the light-on period, the value represents the mean rate during one or more strides. As the stimulus strength increased, CbN firing rates were more likely to fall below the pre-stimulus baseline, and a larger number of slip trials were evident. Across all trials, the probability of evoking a slip with a light step was 76% with the

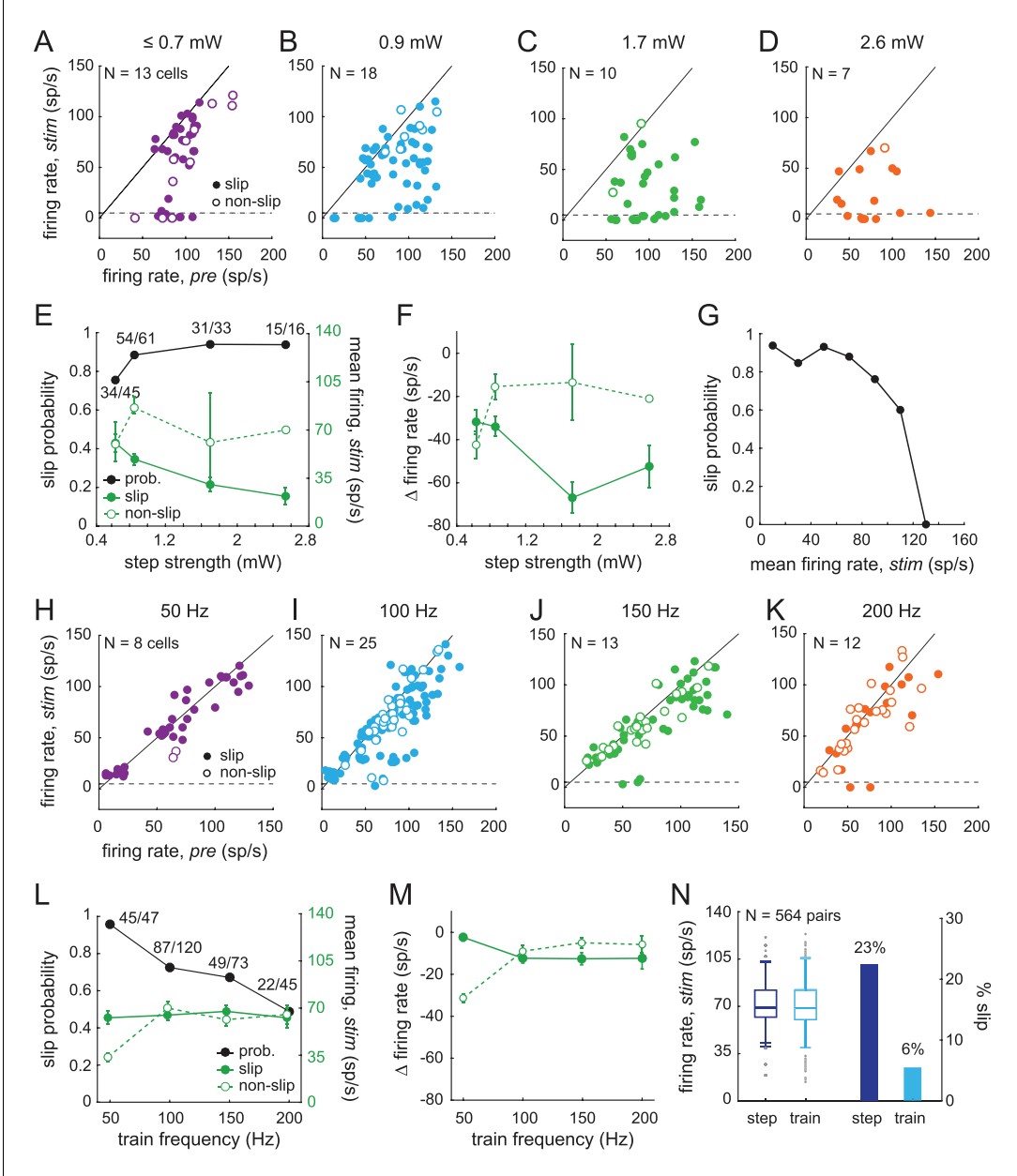

**Figure 9.** Sensitivity of smooth locomotion to optogenetic changes in CbN firing rates and patterns. (**A-D**) Scatter plots for each light intensity of mean CbN cell firing rates during light steps (1 s) *vs.* pre-stimulus firing rates (1 s) for single trials. *Filled symbols*, slip; *open symbols*, non-slip trials; *black line*, unity; *dashed line*, five spikes/s. (**E**) Slip probability (*black*) and mean CbN cell firing rates (*green*) for slip (*solid symbols*) and non-slip (*open symbols*) trials. *Numbers*, slip/total trials (**F**) Change in CbN cell firing rates compared to pre-stimulus firing rates in slip and non-slip trials *vs.* light step strength. Symbols as in (**E**). (**G**) Slip probability *vs.* binned mean firing rates across all step intensities (bin = 20 spikes/s). (**H–K**) As in (**A–D**) for each light train frequency. (**L**) As in (**E**), for trains. (**M**) As in (**F**), for trains. (**N**) *Left,* Box plot of firing rates for 564 pairs of trials selected for similar mean firing rates during a step and a train of light stimuli applied to the same cell. *Grey symbols*, outliers; *whiskers*, two standard deviations. *Right,* Bar graph of fraction of paired trials in which a slip was evoked only by the step (*dark blue*) or only by the train (*light blue*).

DOI: https://doi.org/10.7554/eLife.29546.012

The following figure supplement is available for figure 9:

**Figure supplement 1.** Reanalysis of data of *Figure 9* with slip measurements constrained to the period within the trial when locomotion was perturbed.

*Figure 9 continued on next page*

*Figure 9 continued*

DOI: https://doi.org/10.7554/eLife.29546.013

weakest stimuli and increased to 94% with the strongest stimuli (*Figure 9E*, *black symbols*, $\chi^2$ (7, N = 155)= 7.1, p=0.07, *chi-square test*). The firing rates of CbN cells on slip trials but not non-slip trials showed a corresponding decrease as the step intensity was raised (*Figure 9E*, *green symbols*). Plotting the change in firing rate relative to the pre-stimulus period likewise showed greater firing rate decreases at higher intensities for slip as opposed to non-slip trials (*Figure 9F*, *two-way ANOVA* F (3,147) = 3.91, p=0.05, interaction between step intensity and slip/non-slip, p=0.08). Interestingly, almost all trials on which CbN firing rates were suppressed below $\leq$ 5 spikes/s resulted in slips (*Figure 9A–D*). The only exceptions were with the weakest stimulation, which likely recruited the fewest Purkinje cells. The observation that suppressing firing in CbN cells almost invariably correlated with discontinuities of movement (*Figure 9G*) suggests that even well-practiced motor behaviors, such as locomotion, relies on signals that can be disrupted by sudden and transient loss of input from CbN cells.

The same analysis applied to trials with light trains showed a different distribution of CbN cell firing rates before and during stimulation. The mean rates measured over the 1 s stimulus period continued to cluster around or just below the unity line, illustrating that trains were less effective at suppressing CbN cell firing rates, at all frequencies of stimulation (*Figure 9H–K*, trials from N = 8, 25, 13, 12 cells). Nevertheless, slip as well as non-slip trials were present, with $\geq$49% of trials eliciting slips at each frequency. Slip and non-slip trials could not be well distinguished on the basis of firing rate, which remained near 60 spikes/s regardless of train frequencies (except two non-slip trials with 50 Hz trains; *Figure 9L*), nor could they be distinguished on the basis of CV of the interspike intervals. For trials with $\geq$2 interspike intervals during light stimulation, CVs on slip and non-slip trials, respectively, were 0.90 ± 0.02 (N = 201) and 0.93 ± 0.05 (n = 82) for trains and 1.2 ± 0.07 (N = 111) and 1.0 ± 0.17 (N = 18) for steps. Although the CV was lower for trains than for steps (p=0.007, F (1,408) = 7.23, *two-way ANOVA*), the analysis revealed neither an effect of slip/non-slip on CV (p=0.4) nor an interaction between slip/non-slip and step/train (p=0.2). The data are therefore consistent with temporal restructuring of CbN cell spikes (on a millisecond time scale) by trains of synchronous inhibition, but the consequent decrease in CV was not predictive of whether or not a slip occurred.

The probability of train-evoked slips varied inversely with frequency, with 50 Hz stimulation evoking a high slip probability (0.96), and 200 Hz stimulation evoking the lowest slip probability (0.49, *Figure 9L*, $\chi^2$ (7, N = 285)=25.4, p<0.001, *chi-square test*). The slip probability therefore was not simply a direct consequence of the amount of illumination time as a readout of the magnitude of inhibition. Instead, it may have related to the efficacy of recruiting Purkinje cells to time-lock to the stimulus; both the PSTHs and the local field potentials suggested that the 50 Hz trains favored synchrony of more Purkinje cells. The larger number of Purkinje cells recruited to a perfectly regular, physiologically arbitrary stimulus pattern likely made them more effective at restructuring CbN cell spike timing (on a time scale of milliseconds) in a manner that led to deviation from the normal stride-related modulation pattern, thus favoring slips even without greatly suppressing firing rates.

Indeed, the change in firing rates compared to baseline was similar between slip and non-slip trials at the different train frequencies (*Figure 9M*, *two-way ANOVA* F(3,277) = 0.51, p=0.5, interaction between train frequency and slip/non-slip, p=0.09). These data further suggest that factors such as synaptic depression were not the primary determinants of whether a slip occurred; such a scenario predicts that strong inhibitory synaptic depression would cause high firing rates without slips, while less depression would do the opposite. Since the absolute and relative firing rates were similar in slip and non-slip trials and only slightly changed relative to no-light conditions, the simplest interpretation is that the train stimuli altered stride-related modulation on slip-trials in a way that interfered with movement.

In these analyses, firing rates were all averaged across the full 1 s light stimulus. As stated above, however, slips occurred with a lag and did not always persist throughout the stimulus window. Therefore, for the subset of trials in which slip onset and offset could be identified with certainty, the

measurements were repeated with the analysis window constrained to the duration of the slip. This analysis of the data gave indistinguishable results ().

Although trains could induce slips, pooling all stimulus conditions showed that they did so with a 15% lower probability than steps (steps, 0.86; trains 0.71, *Fisher's exact test,* p<0.001); as in the case of steps, however, in the few instances (six trials) when trains brought CbN cell firing rates below five spikes/s, slips always occurred (*Figure 9H–9K*). One possibility, therefore, is that the different slip probabilities elicited by steps and trains related simply to their differential effects on rate. To test this possibility, we selected the subset of trials in which a step and train applied to the same cell resulted in similar firing rates (within 10 spikes/s). Pairs of trials with firing rates below 10 spikes/s (which always gave high slip probabilities) were excluded. This analysis yielded 564 pairs of trials, for which the mean firing rate for both steps and trains was 71 ± 1 spike/s (*Figure 9N*, *left*, N = 13 cells, 65 distinct step trials, 141 distinct train trials).

Even with comparable firing rates elicited by comparable light intensities, the probability of evoking a slip remained higher for steps than for trains (0.89 vs. 0.72, p<0.001, *McNemar's test for paired data*). Stated differently, a slip was evoked only by the step in 127 pairs (22.5% of all trials), and only by the train on 31 pairs (5.5% of all trials, *Figure 9N*, *right*); on the other trials the response to steps and trains was the same. Since the firing rates were similar in these within-cell pairs, it seems likely that similar numbers of Purkinje cells were activated and that the total inhibition during the paired trials was comparable. Therefore, the higher incidence of non-slip trials during trains suggests that trains interfered less often than did steps with normal stride-related modulation.

Indeed, when the spiking patterns during these rate-matched pairs of trials were compared, the records associated with slips showed an altered stride-related modulation compared to that seen over the time period of a single pre-stimulus stride. In contrast, the records associated with non-slips showed a stride-related modulation comparable to the pre-stimulus control (shown for two rate-matched pairs, *Figure 10A and B*, *left* and *middle*). These changes were evident when firing rate was plotted against paw position for the no-light and light-on conditions, regardless of whether a step or a train generated the slip (*Figure 10A, 10B*, *right*). We therefore reexamined the data of *Figure 8E*, that is, the correlation of firing rates during and before light stimulation for *all* slips associated with non-zero CbN cell firing rates, and separated the responses of steps and trains (). Again, steps gave a larger fraction of slips (83/102 trials, 81%) than did trains (203/306 trials, 66%). Thus, experimentally applied asynchronous Purkinje cell inhibition was relatively more effective than synchronous inhibition at disrupting the stride-related modulation of CbN spikes in a manner that induces irregularities of locomotion.

## Discussion

These results provide evidence that smooth execution of well-trained, voluntary locomotion in mice is sensitive to the cyclic modulation of CbN cell firing rates associated with the stride cycle. Not only did silencing CbN cells via optogenetic activation of Purkinje cells reliably disrupt ongoing movement, but perturbing the modulation of CbN cells was also associated with irregularities in the step cycle, even when CbN cell firing was not fully suppressed by experimentally increased inhibition. Conversely, maintenance of the modulation pattern correlated well with continued walking, even when CbN firing rates were experimentally reduced. In addition, the results suggest that cerebellar output is sensitive to the degree of coherence of Purkinje cell firing, since CbN cells responded differentially to experimentally applied asynchronous or synchronous inhibition superimposed on natural activity patterns. Light steps, which increased Purkinje spike rates without consistently re-patterning spike timing on a millisecond time scale, slowed CbN cell firing and reliably elicited slips. Light trains, which increased Purkinje spike rates to a lesser extent but consistently increased the synchrony of convergent inhibition, generated slips on >50% of trials, suggesting that a resetting of the timing of inhibition could be sufficient to disrupt cerebellar output and the associated behavior. Nevertheless, relative to steps, trains were more frequently associated with non-slip trials. Since non-slip trials were associated with stride-related firing rate modulation, trains appear more permissive of continued modulation, which probably arises from direct mossy fiber excitation of CbN cells. Together, the data are consistent with the idea that simple spike synchrony can create synchronous gaps in inhibition on the millisecond time scale, during which (modulated) excitation from mossy fibers can more effectively elicit (modulated) firing in CbN cells.

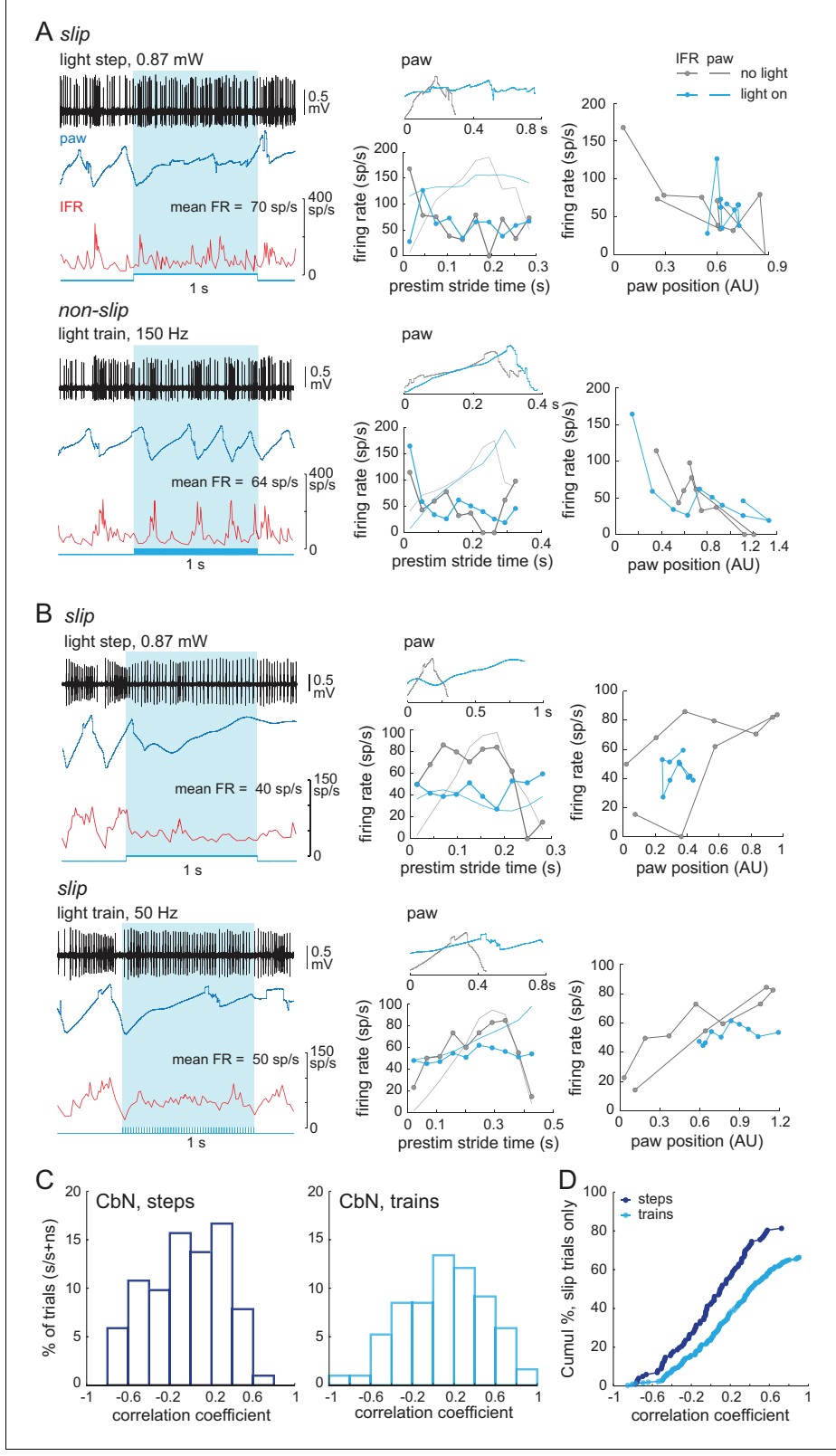

**Figure 10.** Loss of stride-related firing rate modulation during slips evoked by steps or trains. (**A**) *Left,* Rate matched trials from a single CbN cell in which a step led to a slip (*top*) and a train led to a non-slip (*bottom*). *Middle,* Overlaid paw records from the pre-stimulation and stimulation period (*upper plots*) and binned instantaneous firing rates (10 bins) during the pre-stimulus stride and during the stimulation over an equivalent

*Figure 10 continued on next page*

*Figure 10 continued*

period (*lower plots*) for the trials at left. *Right,* firing rate *vs.* paw position for the trials at left. (B) As in (A) for a different CbN cell in which both the step and train led to a slip. (C) Data from *Figure 8H* showing the correlation between the firing rate during the pre-stimulus stride and during the stimulus on slip trials, separated by steps and trains, and normalized to the total number of step trials (122) or train trials (286). (D) Cumulative histograms for the percentage of trials with a given correlation coefficient, for slip trials only. The maximal value of each histogram corresponds to the percentage of slip trials resulting from steps (81%) or trains (63%).

DOI: https://doi.org/10.7554/eLife.29546.014

## Cerebellar output and the persistence of well-trained movements

Cerebellar lesions and alterations of cerebellar physiology give rise to ataxia (*Manto, 2008*; *Morton and Bastian, 2007*; *Zackowski et al., 2002*), and specifically reducing CbN cell output interferes with natural, ongoing behavior (*Thach and Bastian, 2004*). For instance, after portions of the cerebellar nuclei are inactivated with muscimol reaching and grasping movements in primates and cats show slowed reaction times, increased reach trajectory length, and general 'clumsiness' (*Goodkin and Thach, 2003*; *Mason et al., 1998*; *Milak et al., 1997*), as in the prolonged strides and other slips seen here. Similarly, in mice, silencing CbN cells by optogenetic stimulation of Purkinje cells alters the set point of whiskers without preventing whisking (*Proville et al., 2014*). Additionally, the sufficiency of elevated CbN cell activity to induce movements in stationary mice has been demonstrated by optogenetically altering firing exclusively of Purkinje cells (*Heiney et al., 2014*; *Lee et al., 2015*; *Witter et al., 2013*).

Nevertheless, dissecting how patterns of CbN cell firing relate to movements is complicated by the difficulty in determining to what extent changes in firing rate report sensory feedback and/or actively drive motor output. Passive movements like paw flexion are sufficient to alter both Purkinje and CbN cell firing patterns (*Arshavsky et al., 1980*; *Casabona et al., 2010*; *Cody et al., 1981*; *McDevitt et al., 1987*), and treadmill-induced locomotion in decerebrate animals produces modulated activity in Purkinje cells and target neurons. Transiently obstructing limb movement during locomotion in decerebrate ferrets or cats elevates simple spike rates for ~100–200 ms in Purkinje cells (*Lou and Bloedel, 1992a*; *Lou and Bloedel, 1992b*) and abolishes modulation of interpositus cell firing (*Schwartz et al., 1987*), but locomotion persists even with loss of cerebellar output (*Orlovsky, 1972b*). Such observations have led to the suggestion that such modulation normally seen in cerebellar output neurons represents sensory feedback associated with movement, which may facilitate but may not be necessary for locomotion. Additionally, despite the extensive inhibition of CbN cells by Purkinje cells, changes in gait can be subtle when exclusively Purkinje cell firing is changed (; *Levin et al., 2006*; *Machado et al., 2015*; *Mullen et al., 1976*; *Walter et al., 2006*). Thus, to what extent CbN cell firing patterns and Purkinje-mediated inhibition of CbN cell firing facilitates, produces, or reports movements may vary with the behavior and/or conditions.

The present experiments on voluntary locomotion in intact mice demonstrate that transiently suppressing modulation of CbN cell firing by directly stimulating Purkinje cells is sufficient to disrupt smooth, well-practiced locomotion. Additionally, slips were correlated with the loss of modulation rather than the absolute firing rate, suggesting that CbN cells do not simply provide an excitatory tone to downstream regions. The latency of the disruption (~120 ms) suggests either an indirect multi-synaptic signaling before the motor response becomes evident in the hind paw record or that the hindlimb slip is secondary to shorter latency disruption in other muscle groups under cerebellar regulation (e.g., *Machado et al., 2015*). In either case, these data provide evidence that cerebellar output can actively participate in locomotion.

## Modulation of Purkinje and CbN cells during ad-lib locomotion

The repetitive nature of the step cycle makes locomotion a useful behavior for examining the activity of likely-connected Purkinje and CbN cells. Such data isare of interest since CbN cells receive concomitant synaptic excitation from mossy fibers and inhibition from Purkinje cells, which are themselves indirectly excited by mossy fibers via granule cells. Thus, excitation and inhibition may rise and fall together in CbN cells with lags of only two synaptic delays. Here we find that both Purkinje cells and CbN cells showed an elevation of mean firing rates during locomotion relative to rest.

Similar firing rate increases in both cell types have been observed in walking cats (*Armstrong and Edgley, 1984a*, *Armstrong and Edgley, 1984b*) and primates during rapidly alternating arm movements (*Thach, 1968*), raising the question of how inhibition from Purkinje cells interacts with mossy fiber excitation.

Because the goal of the study was to collect data from as many modulated neurons as possible, we recorded from sites restricted enough to ensure that cells showed hindlimb-related activity, but broadly distributed enough to permit random sampling of the population (~400×400 x 600 µm for CbN cells). This approach revealed a wide range of stride-related modulation patterns in different cells. It remains possible that a topographic organization of phase modulation exists, but the resolution of the present work did not permit any such arrangement to be detected. Nevertheless, a reasonable prediction is that the population activity of Purkinje and CbN cells may reveal whether stride-related firing rate modulation in the two cell groups is inverted, indicating that Purkinje cell inhibition overrides excitation and sets CbN firing rates, or co-modulated, suggesting that mossy fiber excitation dominates. In cats, recordings from cells with forelimb receptive fields showed that population averages of firing rates of Purkinje and CbN cells rose and fell at the same phases of the step cycle (*Armstrong and Edgley, 1984a*; *Armstrong and Edgley, 1984b*). The corresponding analysis of the present data with respect to hindlimb movement in mice gave mean peak Purkinje firing rates that were somewhat phase-advanced to that of CbN cells. In neither study, however, did inverse firing patterns predominate. Also, in both studies, firing rates showed one or sometimes two peaks of activity at specific phases of the stride, and the peak firing rates of either Purkinje cells or CbN cells could occur at any phase of the stride. A similar range of stride-related activity patterns of Purkinje cells was reported in a study of freely moving rats, which further demonstrated that the variety of activity patterns across Purkinje cells, and even across strides within a single Purkinje cell, may result from the relation of Purkinje cell firing rates to different aspects of locomotion, including speed, acceleration, pitch, and roll (*Sauerbrei et al., 2015*). Here, the pattern of modulation with respect to the stride across a given Purkinje or CbN cell did not depend strongly on stride duration; one possibility is that the range of values assumed by variables such as pitch and roll was reduced by head fixation.

## Responses of CbN cells to the rate and timing of Purkinje-mediated inhibition

Although the connection patterns between response classes could not be deduced from the present data, Purkinje neurons with different modulation patterns may well converge onto CbN cells. For instance, specific saccadic eye movements in primates may be encoded by convergence of Purkinje cells with different activity patterns (*Herzfeld et al., 2015*). Thus, several factors may govern the outcome of synaptic interactions among excitatory and inhibitory inputs to CbN cells.

A series of in vitro studies have suggested that the degree of coherent firing, or synchrony, of convergent Purkinje cell inputs may be one such factor. The intrinsic ion channels of CbN cells generate spontaneous action potential firing at 70–100 spikes/s (*Mercer et al., 2016*; *Person and Raman, 2012a*; *Raman et al., 2000*; *Telgkamp and Raman, 2002*) even with no synaptic input. CbN cells therefore require active hyperpolarization and/or shunting from Purkinje cells to prevent them from firing. About 30–50 Purkinje cells provide strong synaptic contacts to each CbN cell, but each simple spike evokes unusually brief GABA$_A$ receptor-mediated IPSCs in CbN cells, with decay kinetics of 2–2.5 ms (*Person and Raman, 2012a*). If converging Purkinje cells fire asynchronously, the inhibitory currents overlap in time and CbN cell firing can be effectively suppressed. Conversely, if Purkinje cells synchronize, brief periods of inhibition are followed by windows of reduced inhibition when CbN cells can fire (*Person and Raman, 2012a*; *Wu and Raman, 2017*). The degree of synchrony influences the effect of synaptic excitation: lower jitter among Purkinje inputs permits CbN cells to fire more action potentials in response to a fixed amount of excitation. In other words, the efficacy of excitation is raised by the synchrony of inhibition (*Wu and Raman, 2017*).

In the present experiments, synchrony could not be measured directly, since it was not possible to identify and record simultaneously from multiple converging Purkinje cells. Nevertheless, all CbN cells showed a consistent spike suppression on repeated cycles of stimulation at fixed latencies with millisecond-level precision, reflecting coherent inhibition that was interpreted as arising from increased synchrony of firing by converging Purkinje cells. Supporting this idea, Purkinje cell spike probability increased reliably with each cycle of the light train, and local field potentials in

anesthetized mice demonstrated Purkinje spike-related signals at rates that matched the train stimuli. In contrast, steps did not generate well-timed inhibition of target cells. Although step-evoked local field potentials began with a ~50 ms damped oscillation of ~300 Hz, no corresponding pattern emerged in recordings of CbN cells in awake mice, possibly owing to an anesthetic-induced increased coherence in the local field potential, a lack of convergence of Purkinje cells sampled in the local field recording, or an absence of gaps in inhibition of the target cells. For the remaining 950 ms of stimulation, the millisecond-level timing of CbN cell spikes diverged for steps and trains, and the disruption of locomotion often persisted well into this later stimulus period. Taken together, it seems reasonable to infer that trains increased the synchrony of Purkinje cell firing throughout the stimulus.

These in vitro observations and in vivo analyses provide a context for interpretation of the step and train data in the present study. In many respects, the effects of steps and trains were similar: if light stimulation altered stride-related modulation, then a slip occurred. Purkinje and CbN cell firing rates, however, were reduced to a lesser extent by trains than steps, possibly owing to the shorter illumination times associated with trains and/or the limited responsiveness of ChR2 to high-frequency light pulses. Nevertheless, imposing an arbitrary, regular pattern of relatively synchronous inhibition often restructured CbN cell spike timing (on a millisecond time scale) was enough to interfere with stride-related modulation. With such disruption, the step cycle could be perturbed even without decreasing the mean rate of CbN cell firing, providing evidence that spike patterns of CbN cells on the time scale of a single stride can have behavioral consequences.

Of note, however, are the other trials, in which the natural pattern of stride-related firing modulation persisted. In these trials, the modulated excitation from mossy fibers (*Powell et al., 2015*) presumably contributed to the persistence of stride-related modulation in CbN cells. In other words, on about half the trials, the naturally occurring excitation successfully overcame synchronous optogenetically applied inhibition. Moreover, when steps and trains elicited the same firing rate in a given cell but different behavioral outcomes, steps were nearly four times more likely than trains were to generate slips, that is, to disrupt the modulation pattern. This observation is consistent with in vitro data demonstrating that excitation was most effective during gaps in inhibition induced by coherent Purkinje cell firing (*Wu and Raman, 2017*).

In vivo, synchrony of simple spikes has been repeatedly observed in recordings of neighboring and likely converging Purkinje cells, and the frequency of observation of synchrony reportedly increases during cerebellar behaviors (*Bell and Grimm, 1969*; *Ebner and Bloedel, 1981*; *Heck et al., 2007*; *MacKay and Murphy, 1976*; *de Solages et al., 2008*; *Wise et al., 2010*; reviewed by *Person and Raman, 2012b*), and tetrode recordings in freely moving rats illustrate correlated activity between stride-modulated Purkinje cells (*Sauerbrei et al., 2015*). The extent to which Purkinje cell synchrony occurs during movements remains unknown, but, given the high firing rates of Purkinje cells and the common inputs they receive from granule cells and inhibitory neurons, it seems highly improbable that inhibitory input to CbN cells will always be maximally asynchronous. An interesting possibility, yet to be tested, is that the degree of synchrony might vary systematically during each stride. Since stride-related firing rate modulation is correlated with smooth locomotion, and since excitation from mossy fibers appears sufficient to generate modulated firing in target cells (*Powell et al., 2015*), a plausible outcome of naturally occurring synchronous Purkinje cell activity might be to create physiologically appropriate gaps in which mossy fibers might evoke CbN cell spikes more efficiently, providing a relatively more permissive state for smooth execution of natural behaviors.

## Materials and methods

**Key resources table**

| Reagent type (species) or resource | Designation | Source or reference | Identifiers | Additional information |
|---|---|---|---|---|
| strain, strain background (*Mus musculus*) | B6.Cg-*Gt(ROSA)26Sor*$^{tm27.1(CAG-COP4*H134R/tdTomato)Hze}$/J | Jackson Laboratory | Jackson stock: 012567 | - |

*Continued on next page*

Continued

| Reagent type (species) or resource | Designation | Source or reference | Identifiers | Additional information |
|---|---|---|---|---|
| strain, strain background (*Mus musculus*) | B6.129-Tg(Pcp2-cre)2Mpin/J | Jackson Laboratory | Jackson stock: 004146 | - |
| recombinant DNA reagent | AAV9.CBA.Flex.ChR2(H134R)-mCherry.WPRE.SV40 | Addgene | Addgene: 18916 | - |
| recombinant DNA reagent | AAV9.EF1a.DIO.hChR2(H134R)-eYFP.WPRE.hGH | Addgene | Addgene: 20298 | - |
| software, algorithm | custom paw tracking software | Actimetrics | from Actimetrics www.actimetrics.com | free by contacting company |
| software, algorithm | irradiance calculator | Deisseroth Lab Optogenetics Resources | https://web.stanford.edu/group/dlab/ | - |

## Animals

All procedures conformed to the NIH guidelines and all protocols were approved by Northwestern University's Institutional Animal Care and Use Committee (Animal Welfare Assurance Number, A3283-01, protocol IS00000242, IMR). Ai27D (B6.Cg-*Gt(ROSA)26Sor*$^{tm27.1(CAG-COP4*H134R/tdTomato)Hze}$/J, stock 012567) and *Pcp2*-Cre (B6.129-Tg(*Pcp2*-cre)2Mpin/J, stock 004146) Jackson Laboratory, Bar Harbor, ME) mice were crossed to generate offspring heterozygous for channelrhodopsin-2 (hChR2 (H134R)), referred to as ChR2, expression in Purkinje cells. Mice had free access to food and water and were maintained under a reverse day-night cycle (12:12 hr) for at least two weeks before the training phase of experiments began. Home cages included running wheels (Bio-Serv, Flemington, NJ) to provide experience in running. Experiments were performed on adult mice (P45-P70). Initially, both male and female mice underwent training to run on a treadmill while head-fixed (see *Behavior* below); however, while 38/44 males (86%) ran consistently, only 4/13 (31%) females did. Since running was essential to the study, only male mice were used for experiments. The data presented are from 23 adult male mice.

## Surgical procedures

Mice underwent surgery for a craniotomy to allow electrode access and for installation of a head-plate for head-fixation. They were anesthetized with ketamine (120 mg/kg, intra-peritoneal) and xylazine (3 mg/kg, intra-peritoneal) and mounted in a stereotaxic apparatus (Stoelting Co., Wood Dale, IL). Body temperature was maintained at 37°C with a heating pad (Harvard Apparatus, Holliston, MA) and oxygen was delivered through the nose. Lidocaine (2%, topical) was applied to the skin and buprenorphine SR-LAB (1 mg/kg, subcutaneous, ZooPharm, Windsor, CO) was administered peri-operatively for analgesia. A rectangular metal headplate with a central window (*Dombeck et al., 2007*) was attached to the skull with Metabond (Parkell, Inc., Edgewood, NY), just posterior to the bregma. A plastic well was also cemented around the recording site to hold agarose and the ground electrode. A craniotomy was made over the paravermal region of the cerebellum (bregma −6.3 mm, 1.85 mm lateral,~0.3–0.5 mm diameter). The dura was left intact and covered with Kwik-Sil (WPI, Inc., Sarasota, FL) until the time of recording. Saline solution (0.9% NaCl, 50 ml/kg, subcutaneous) was administered to aid hydration while animals recovered from surgery.

For local field potential recordings only, experiments were done in mice (N = 2) anesthetized with intraperitoneal injection of ketamine (120 mg/kg) and xylazine (3 mg/kg). Mice were held in a stereotaxic apparatus and kept on a warming pad to maintain body temperature at 37°C and oxygen was delivered through the nose. Vital signs were monitored and the animal was re-dosed with 20–50% of the initial dose of ketamine when the toe-pinch reflex began to reappear.

## Behavior

At least one week after surgery, mice began training to run on a cylindrical, freely rotating, non-motorized, 18.6 cm diameter Styrofoam treadmill (*Domnisoru et al., 2013*; *Heys et al., 2014*). The setup was located in a darkened room with white noise delivered through speakers to minimize startle from external noises. Head-fixed mice were placed on the treadmill for 30 min, increasing to 2 hr over the course of week. Most male mice (86%) ran regularly on the treadmill by days 7–10. After

mice ran consistently for at least 60% of a 2 hr session, recordings were made (see *Electrophysiological Recording* below) during head-fixed running for up to 3 hr per day for 1–2 weeks. During recordings, the ipsilateral hind paw was monitored from the side with a high-speed, infrared video camera (240 fps, Tamron, Commack, NY). The x-position (anterior-posterior) and y-position (dorsal-ventral) of the paw was tracked (240 fps) with a custom-written program (Actimetrics, Wilmette, IL). Image contrast was aided by painting the treadmill black and illuminating the paw with a red flashlight. During running the paw position varied mostly in the x-domain, which was stored for analysis.

## Injections and anatomy

To identify Purkinje cells that projected to the CbN cells associated with locomotion, tracers were pressure injected with a picospritzer (150–400 pulses, 6 psi, 5 ms pulse) into the region of the interpositus targeted for recordings (N = 3 mice). In wild-type mice, the retrograde tracer cholera-toxin subunit B conjugated to Alexa Fluor-488 (Invitrogen) was injected, and in *Pcp2*-cre mice, a mixture of viruses (200–500 nL) was applied to express ChR2(H134R) with a fluorescent marker (AAV2/9 conjugated with either mCherry (AAV9.CBA.Flex.ChR2(H134R)-mCherry.WPRE.SV40; Addgene 18916) or eYFP (AAV9.EF1a.DIO.hChR2(H134R)-eYFP.WPRE.hGH; Addgene 20298; U Penn viral vector core). With either technique, labeled Purkinje cells were consistently found lateral to the primary fissure in coronal sections, that isi.e., in the lobulus simplex of the cerebellar cortex, following injections in the central regions of anterior/posterior interpositus nucleus. These regions were therefore targeted for recordings.

## Electrophysiological recordings and optical stimulation

Electrophysiological recordings were made from head-fixed, running mice (*Harvey et al., 2009*). Borosilicate micropipettes (4–6 MΩ) were pulled on an P97 Sutter micropipette puller (Novato, CA), heat polished with a microforge MF-900 (Narishige International Inc., East Meadow, NY), and filled with (mM) 125 NaCl, 5 KCl, 10 D-Glucose, 10 HEPES, 5 $CaCl_2$, pH = 7.4, 300 mOsm). A Ag-AgCl ground electrode was placed on the surface of the skull near the craniotomy and held by 2.5% agarose, which also covered the exposed brain surface and stabilized the recording electrode. An optic fiber (240 μm, NA 0.63, Doric Lenses, Québec, Canada) was inserted in the recording pipette through the side port of the electrode holder (Warner instruments, Holliston, MA) to permit simultaneous electrophysiological recordings and optical stimulation (*Katz et al., 2013*). The end of the fiber was 4–5 mm from the tip of the pipette.

Recordings were made beneath the craniotomy (bregma −6.3 mm, 1.85 mm lateral,~0.3–0.5 mm diameter) at a depth of up to 1900 μm for Purkinje cells in the lobulus simplex and 1900–2500 μm for CbN cells in the posterior and less commonly the anterior or dorsolateral horn divisions of the interpositus nucleus. Light stimuli (465 nm, 8–53 mW/mm$^2$) were generated by an LED coupled to an LED driver (Doric Lenses). Stimuli were calibrated with a photometer (PMD100, S140C, Thorlabs, Inc., Newton, NJ). The maximum measured light intensity at the pipette tip of ~ 40 mW/mm$^2$ was estimated to attenuate to ~ 20 mW/mm$^2$ about 50 μm from the tip (based on (*Yizhar et al., 2011*) https://web.stanford.edu/group/dlab/). For this fiber size, this intensity is equivalent to 0.9 mW, which corresponds to the threshold for action potential activation in pyramidal neurons in cortical slices from Ai27 mice (*Madisen et al., 2012*).

Loose-cell attached recordings ($R_{seal}$ > 10 MΩ) were made in current clamp mode. Electrical signals were acquired at 10 kHz with an Axoclamp-2B or Axopatch-200B amplifier, a Digidata-1550, and pClamp software (Molecular Devices, Sunnyvale, CA). Action potential waveforms were identified by a voltage threshold of the electrical trace following low-pass digital filtering at 1 kHz. In trials from 8 Purkinje cells representing cells with high and low firing rates, slips and non-slips, and steps and trains, complex spikes were identified in 1 s windows around the light stimulation and found to have rates of 1.5 ± 0.4 spikes/s (before), 2.9 ± 0.7 spikes/s (during), and 1.1 ± 0.4 spikes/s (after). An elevation of complex spikes during the stimulation, which was evident in only 3 of 8 cells, is consistent with an experimentally driven increase in inhibition of nucleo-olivary cells and increase of inferior olivary firing. Because it was not possible to disambiguate the direct effect of optically raising Purkinje cell spikes from biological effects, e.g., associated with sensory or motor signals, complex spikes were not examined further. Additionally, when complex spikes were extracted from the record, the total firing rates were reduced by < 2%. Therefore, both simple and complex spikes

were included in all analyses that are presented, but the values are overwhelmingly dominated by simple spikes.

Both in the cerebellar cortex and nuclei, recordings were made from regions where light stimulation evoked altered movement of the ipsilateral hindlimb (described further in *Results*). Each cell was recorded continuously for 10 trials of 7 s each without optical stimulation, followed by several conditions of light steps and light trains. Both steps and trains were applied for 1 s with 2 s recording before ('pre-stim') and after stimulation ('post-stim'). 3–10 trials of each stimulus condition were repeated with a 10 s inter-trial interval. Steps and trains were tested in increasing order of their strength (0.56, 0.65, 0.9, 1.7 and 2.6 mW) or frequencies (50, 100, 150, 200, Hz at 1.7 or 2.6 mW), respectively. Multiple penetrations were made per session. Between sessions, the surface of the brain was sealed with Kwik-Sil (WPI, Inc.).

At the end of the final session, the recording pipette was withdrawn and a glass injection pipette (taper ~3.5 mm, tip 20–25 μm, WPI, Inc.) was inserted at the same stereotaxic coordinates. The recording locations and track were marked by pressure injection (~1–3 psi) of 1–2.5% Alexa Fluor-488 fluorescent dye conjugated with dextran (10K, anionic, fixable; Life Technologies, Carlsbad, CA). Mice were euthanized with Euthasol (100 mg/kg, intraperitoneal, Virbac, Fort Worth, TX) and transcardially perfused with paraformaldehyde (4% in PBS, ThermoScientific, Skokie, IL). Brains were kept in 4% PFA overnight, then triple-rinsed. Coronal cerebellar sections (100 μm) were cut at 4°C in PBS (0.01 M, Sigma Life Science, St. Louis, MO) on a vibratome (VT1000S, Leica Biosystems, Nussloch, Germany). Sections were imaged on a confocal microscope (Leica Biosystems, TCS SP5, Biological Imaging Facility, Northwestern University) and processed with the Fiji version of ImageJ software (*Schindelin et al., 2012*). The injection location was taken as the site of the maximum intensity (laser power and gain held constant for all imaged sections).

For local field potential recordings from Purkinje cells, the pipette tip was broken to give an electrode resistance of 0.8–1 MΩ, which permitted the fiber to be positioned as it was for direct recordings from CbN cells. Each optical stimulation condition was applied ten times. The ten sweeps in each condition were averaged and the signal was bandpass filtered from 50 Hz to 1 kHz. FFTs were done on the resulting sweeps and spectral power density was plotted. The power of the unfiltered residual electrical noise at 60 Hz was blanked for clarity.

## Data analyses

Data are presented as mean ± SEM except as noted. All analyses including statistical tests were performed in Matlab (Mathworks). For inclusion in the study, Purkinje and CbN cells had to show appropriate firing responses to light (increases for Purkinje and decreases for CbN) that were statistically significant (one-sided paired t-test, $p < 0.05$), as well a behavioral 'slip' (described in Results) of the ipsilateral hindlimb in any light stimulus condition.

## Identification of strides

For quantification of paw movement, a region of interest (ROI) was set to cover the maximal range of hind paw movements during locomotion. Tracked paw position was normalized to the ROI window and therefore has arbitrary units of magnitude (*Figure 1A*, green box). 'Rest' periods, that is, when the mouse was not running, were evident in the paw-tracking record as a minimal change in paw position (<10% of that during strides). Neuronal activity during rest periods was measured from cells in which the total duration of all the rest epochs (in the absence of light stimulation) exceeded 2 s.

'Run' periods were identified as sawtooth-like sections of the paw-tracking record indicative of the hind paw moving forward and backward as it completed each stride (plant-stance-lift-swing-plant). Occasionally the tracking system would lose the paw, resulting in instantaneously rising transients in the tracking record that resembled brief square pulses. Tracking usually resumed within a few samples, so these transients were viewed as recording noise and were disregarded during analyses. Strides were identified from smoothed paw traces (sliding-average filter, 50 ms duration) as a trough-to-trough cycle. At least 30 strides, each 100–500 ms in duration were required for a cell to be included in the study. All strides were aligned to their peak (the lift). The ratio of the stance phase (trough-to-peak) to the swing phase (peak-to-next-trough) was calculated for each stride. The median value of this ratio for all strides associated with a recording was used to normalize strides by

dividing each stride into stance and swing bins, for a total of 10 bins. Specifically, if the stance accounted for 70% of the stride, seven bins would be allotted to the period before the lift and three bins to the period after the lift. The stance phase generally accounted for 60–70% of the whole stride, so 6–7 bins were usually stance bins. In this way, the stance and swing bins would be of comparable (though not identical) durations, and firing rates in stance and swing could be examined separately. Reanalysis of the data with 10 equal bin widths yielded similar results and did not alter classification schemes, confirming that the differences in bin widths were minimal.

Instantaneous firing rates were calculated for each neuron. Mean firing rates during a stride were calculated from averaging the instantaneous firing rates within each bin. Binned firing rates and the binned paw displacement were averaged across all strides for a cell. The depth of modulation was quantified using a modulation index (MI) defined as $(FR_{max} - FR_{min})/(FR_{max} + FR_{min})$, where $FR_{max}$ and $FR_{min}$ are the maximal and minimal binned firing rates during the normalized stride.

## Classification of firing rate modulation patterns

Binned firing rates of Purkinje and CbN cells relative to paw position during the stride were classified independently by both investigators, as described in *Results*. Relative firing rates rose and fell at consistent phases of the stride for both Purkinje and CbN cells across stride durations, which justified normalizing strides (*Armstrong and Edgley, 1984a*, *Armstrong and Edgley, 1984b*). Nevertheless, because absolute firing rates could be variable (e.g., *Sauerbrei et al., 2015*), firing rates were plotted against paw velocity (derived from paw position) and analyzed by linear regression in separate analyses. 40% of Purkinje cells (14/35) and 21% of CbN cells (8/39) showed a significant correlation of firing rate and velocity. This relationship was not investigated further, but is encompassed in Class III, IV, V, and VI categorizations.

## Analysis of optogenetically induced slips

Optogenetic stimulation during locomotion could generate disruptions of movement that were evident in the paw-tracking record. Because the treadmill was not motorized, the running speeds of mice were not held constant, giving rise to variability in the duration of strides as well as the rate of paw movement in the stance and swing. Consequently, it was not possible to use a fully automated analysis to identify slips. We therefore used the following approach: The paw record associated with each trial (1 s pre-stim, 1 s light stimulation, 1 s post-stim) was first inspected and classified manually, blind to stimulus condition, to identify potential slip trials. Trials in which running rate changed greatly just before the stimulus or in which running ceased during the light stimulus were discarded. If strides proceeded regularly throughout the 1 s, the trial was categorized as a 'non-slip'. If at any point in the 1 s light stimulus, however, at least one stride showed (1) a greatly lengthened or shortened duration relative to pre-stimulus strides, (2) a strongly reduced slope of the stance or swing phase relative to pre-stimulus strides, or (3) a deviation from a sawtooth pattern, the trial was categorized as a potential 'slip.' These potential slip trials were then evaluated quantitatively. The last complete pre-stimulus (control) stride was detected automatically (as above) and strides during optogenetic stimulation were compared to the control stride. For a trial to remain classified as a slip trial, it had to have at least one stride with either a duration ≥20% longer than control (prolonged stride), a duration ≥20% shorter than control (incomplete stride), or a slope of the stance or swing phase ≥20% lower than control (arrested stride). Several trials fit into more than one of these slip categories (altered strides). Of the 971 putative slip trials, the automated analysis successfully detected a complete stride before and during stimulation on 859 trials; 800 (93%) fulfilled the criteria of at least one category and were confirmed as slips, whereas 59 (7%) did not reach threshold for slips but did not look like non-slips so were discarded. For the other 112 strides, the automated analysis was unsuccessful owing to factors such as a pre-stimulus stride extending for some tens of ms into the stimulus period, or the lack of an identifiable stride at any point during the stimulation. We counted these trials as slips, with an awareness that 7% (~8 trials) might not have been true slips. For a total of 912 slip trials, the misclassification error for slips would therefore be <1%. With this approach, although small optogenetically induced alterations of gait may have been ignored and treated as non-slips, slip trials are highly likely to represent real, experimentally generated changes in movement. The analysis therefore errs on the side of incorrect rejections but avoids false positives.

In cells included in the study, a light-induced change in gait (including the onset of locomotion from rest) occurred on at least 50% of all trials in at least one stimulus condition.

The activity of each cell with and without optogenetic stimulation was examined with respect to paw movement. For non-slip trials, binned firing rates averaged across all strides during light stimulation were compared with binned firing rates from strides without light stimulation. For slip trials, the last stride before the light stimulus served as a template, and the firing rate of the cell during the slip period was compared to that during the template. Binned firing rates during the template stride were compared to those during the slip epochs using the same histogram bins as the template. For the special case of the 'prolonged stride' slip category, prolonged strides were aligned to the lift and analyzed as for undisrupted strides and binned firing rates between normal and slip strides were directly compared.

## Statistical testing

The normality of each data set was assessed by a Shapiro-Wilk test. For normal data, t-tests (paired or unpaired) were performed. For non-normal data, signed-rank (paired) or Wilcoxon rank sum tests (unpaired) were performed. Tests were two-tailed, except when testing for optogenetically induced increases or decreases in firing rates, in which case the tests were one-sided. Rayleigh tests for circular non-uniformity or Kuiper's tests (as noted) were used to test for non-uniform distributions of data, e.g., over a stride cycle or following successive pulses in light trains (toolbox for circular statistics with Matlab, *Berens, 2009*). Pearson's correlation coefficients were used to evaluate the linear correlation between firing rates in two conditions. Two-way ANOVAs were performed to assess the relation among light stimuli, behavioral responses, and changes in firing rates. Fisher's exact tests and $\chi^2$ tests for independence were used to assess dependence of slip probability on light pattern or light parameters. McNemar's tests for paired nominal data were used to test for slip probability in pairs of trials with matched firing rates. All p-values are reported. In figures, one, two, or three asterisks indicate $p<0.05, <0.01$, or $< 0.001$.

## Acknowledgements

We are grateful to Dr. Daniel Dombeck for advice on recording from awake head-fixed running mice, Dr. David Ferster for the paw-tracking program and comments on the manuscript, Dr. Izumi Sugihara (Tokyo Medical and Dental University) for consultation on anatomy, Dr. Mitra Hartmann for advice on local field potentials, as well as members of the Raman lab for helpful discussions. Confocal imaging was done in the Biological Imaging Facility, Northwestern University.

## Additional information

### Competing interests

Indira M Raman: Reviewing editor, *eLife*. The other author declares that no competing interests exist.

### Funding

| Funder | Grant reference number | Author |
|---|---|---|
| National Institute of Neurological Disorders and Stroke | R37-NS39395 | Indira M Raman |

The funders had no role in study design, data collection and interpretation, or the decision to submit the work for publication.

### Author contributions

Rashmi Sarnaik, Conceptualization, Formal analysis, Validation, Investigation, Visualization, Writing—original draft, Writing—review and editing; Indira M Raman, Conceptualization, Formal analysis, Supervision, Funding acquisition, Validation, Visualization, Writing—original draft, Project administration, Writing—review and editing

## Author ORCIDs

Rashmi Sarnaik (iD) http://orcid.org/0000-0003-1977-2740
Indira M Raman (iD) http://orcid.org/0000-0001-5245-8177

## Ethics

Animal experimentation: All procedures conformed to the NIH guidelines and all protocols were approved by Northwestern University's Institutional Animal Care and Use Committee (Animal Welfare Assurance Number, A3283-01, protocol IS00000242, IMR).

## Decision letter and Author response

Decision letter https://doi.org/10.7554/eLife.29546.017
Author response https://doi.org/10.7554/eLife.29546.018

## Additional files

**Supplementary files**
• Transparent reporting form
DOI: https://doi.org/10.7554/eLife.29546.015

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
