## [Decision Letter]

Thank you for submitting your article "Control of free and optogenetically perturbed running by spike rate and timing of neurons of the mouse cerebellar nuclei" for consideration by *eLife*. Your article has been reviewed by three peer reviewers, and the evaluation has been overseen by Gary Westbrook as the Senior Editor. The reviewers have opted to remain anonymous. The reviewers have discussed the reviews with one another and the Senior Editor has drafted this decision to help you prepare a revised submission.

Summary:

All the reviewers thought the topic was very important and that the quality of the work is high. However, there were concerns as to the extent to which the results prove causality between the neural activity and locomotion, as most clearly stated in reviews 2 and 3. Overall there was broad agreement among the reviewers concerning the specific comments raised by all reviewers regarding required changes to analysis and interpretation. Because the reviews include substantial amounts of text, we have included the reviews verbatim rather than try to summarize subtleties. We expect that addressing the comments will require additional analysis and perhaps some additional experiments. A few issues also arose in the discussion that we would like you to address:

1) Due to the lack of measurement of front paw movement and the previous recordings from other species, the focus of the paper might be best directed to the question of synchronous/asynchronous activity. However, synchrony vs asynchrony with the different optogenetic stimulation paradigms has not been demonstrated, and it needs to be. The authors could address this with LFP recordings from cerebellar cortex during continuous and pulsed stimulation (at different frequencies).

2) An additional concern is the possibility of direct stimulation of Purkinje cell axon terminals, particularly in the experiments recording from CbN neurons as it appears that the distance between fiber and pipette tip was fixed in the experiments). We recognize that the recordings were done at a max depth of 2.5 mm with the fiber tip 4-5 mm above the pipette tip, thus there would have been a lot of scatter through the 2.5 mm of brain above the DCN. Nonetheless the targets of stimulation (Purkinje cells) were at a systematically different distance between the DCN recordings and Purkinje cell recordings.

*Reviewer #1:*

The authors made recordings from Purkinje cells (PCs) and neurons in the interpositus deep cerebellar nucleus (CbNs) in locomoting mice. They used optogenetic stimulation of PCs to alter activity in these neurons and their targets in the deep nuclei and attempt to correlate these perturbations in activity with alterations in locomotor behavior. The electrophysiological recordings in this study are of high quality, and the experiments have the potential to describe a causal relationship between cerebellar activity patterns and the execution of ongoing motor programs.

These findings build on previous work showing that PC and CbN firing patterns are periodically modulated with step cycle – described by Armstrong and colleagues over 30 years ago – and that cerebellar perturbations of various kinds (with lesions, microstimulation, or optogenetics) can alter ongoing motor programs. The potential novelty of the present study lies in its ability to draw causal relationships between the types of perturbations induced in PC activity (steps vs. trains), and specific alterations in locomotor behavior. In this regard, the authors could dig deeper into their data in order to strengthen the conclusions drawn and enhance the novelty of the findings.

Major comments:

1) A main piece of analysis that is lacking is a description of any systematic relationship between the timing and types of optogenetic perturbations and the resulting defects in the animals' gait. Here are a few questions that could be addressed:

- Does some feature of the underlying activity explain why the optogenetic stimulus is effective at causing 'slips' on some trials and not others?

- Is there a particular phase of the step cycle which was more 'vulnerable' than others to the optogenetic perturbation?

- Is the stimulation more likely to cause a 'slip' at certain running speeds than others?

- Did trains or steps induce greater numbers of different types of slips over the course of single recordings (i.e. the same cell)? In general, it is unclear if trains, steps and different powers were applied over the course of single recordings or whether each cell was only subjected to a subset of stimuli. Comparing across stimuli within cells, when the optogenetic stimulus was constant, would be a useful group of analyses.

- Is there a systematic relationship between the type of 'slip' – prolonged, altered, or incomplete stride – and any of the abovementioned criteria?

These analyses would substantially improve our understanding of how cerebellar activity can affect ongoing motor programs. Otherwise, this study could be seen as being largely confirmatory, validating a set of previously documented findings with contemporary methods (optogenetics).

2) It is somewhat worrying that 'slip' events were detected manually and (apparently) without blinding. It would be good if the authors could develop automated criteria for what they call a 'slip'. Paw tracking is admittedly a difficult analysis challenge, but formalizing it in order to remove potential bias is important. This is particularly worrying in a few of the examples shown, which seem to contain high frequency events (e.g. the paw trace in 5A, left panel just at the end of the optogenetic step) that look aberrant. Are these errors in the paw tracking itself or actual movements?

3) The authors should show explicitly that their optogenetic perturbations (light flash + ChR2) induce stride perturbations at higher rates than in baseline (no light flash) and negative control (light flash but no ChR2) conditions. Description of these measurements is mentioned deep in the Materials and methods section, but no data are shown. Adding these comparisons in various places throughout the paper (for example in Figure 8, or as a supplement) would contextualize interpretation, since variability in step-cycle PC firing is related to other behavioural parameters (see Sauerbrei et al., 2015).

4) Complex spiking in Purkinje cells appears to be completely neglected in this study. The authors should at least attempt to extract complex spikes from their recordings and analyze their response properties during the optogenetic stimulation. Is complex spiking increased during 'slips' to a greater extent than in non-slip trials? Related to the point above – does the flashing of the optogenetic stimulus induce complex spiking (this is likely a novel/unexpected sensory stimulus)?

5) The authors mention in the Materials and methods that a subset of their neurons were modulated by running speed. Was this consistent across individual step cycles? If so, it would be a departure from the findings of Armstrong and colleagues, who showed that firing rates, when normalized to step cycle, were invariant to different speeds of walking on a motorized treadmill. It would be valuable to know if during free running, the depth of modulation varies with speed (consistent with the findings of Sauerbrei et al., 2015).

*Reviewer #2:*

How the cerebellum contributes to already learned, ongoing movement is an important and much debated question. While most studies have focused on the role of cerebellar circuits in motor learning, the idea that the cerebellum plays a significant role in ongoing movement has only recently become more widely accepted. However, the same modern optogenetic tools that have permitted causal demonstrations that Purkinje cell firing can drive motor output have also suggested an overly simplified relationship between Purkinje cell output, cerebellar nuclear cell (CbN) firing, and movement by implying a strict inverse relationship between Purkinje cell firing and CbN firing. In contrast, in vivo recordings from CbN neurons and Purkinje cells have suggested a more complex relationship between cerebellar output and movement. Here, Sarnaik and Raman seek to address the important question of how the cerebellum contributes to ongoing movement by combining in vivo recordings from both the CbN and cerebellar cortex in combination with optogenetic manipulations designed to test how the firing patterns of neurons in each location are causally related to motor output. By tailoring optogenetic stimulation of Purkinje cells in a manner intended to either promote or reduce population synchrony, in combination with high-speed measurements of hind limb position during locomotion, they conclude that Purkinje cell synchrony is favorable for normal movement, likely by creating widows of excitability for ongoing mossy fiber excitation. If true, this conclusion is significant, and I would be in favor of publication. However, I am skeptical of the central conclusion, as I am not yet convinced that the data supports the claim that enhanced synchrony is favorable for ongoing movement. Moreover, the relationship between movement, CbN spiking and synchrony overall is not fully explored, and paper would benefit greatly from further analysis regarding the relationship between these parameters.

Major comments:

1) Movement errors may actually be highest when PC synchrony is highest for light pulse trains. Based on Figure 7, it is challenging to interpret the relationship between Chr2 train frequency and PC synchrony. Because PCs were recorded one at a time, it is unclear how light spreads from the stimulation/recording pipette to the surrounding tissue activate nearby PCs. A nearby LFP electrode could have been useful to assess relative population synchrony in absence of distance dependent paired PC recordings. However, from the current measurements, it is clear that peak PC spike probability is inversely proportional to train frequency, strongly suggesting the likelihood of decreased synchrony in the population with increasing stimulation frequency. This hypothesis is consistent with the CbN recordings. Here, the largest suppression (peak transient reduction in spike probability) happens for the lowest stimulation frequency of 50 Hz. This could suggest PC synchrony is highest for the 50 Hz stimulation, because CbN neurons integrate 50-80 PC inputs and suppression should be largest when synchrony is highest. It seems unlikely that the difference in CbN suppression is purely due to short-term depression from PC synapses, because the spike probability in PCs at 100 Hz is nearly half that at 50 Hz, which should normalize for depression between these frequencies. Hence, though the measurements are indirect, both PC and CbN recordings suggest highest PC synchrony at 50 Hz stimulation. Comparing these data with Figure 9, the highest probability of movement error occurs for 50 Hz stimulation. More specifically, in Figure 9L, though mean spike rate is invariant, motor errors decrease dramatically with simulation frequency. In combination with the data shown in Figure 7, these results suggest the possibility that with the same type of stimulation (pulse trains), motor errors occur with higher probability as PC synchrony increases, conflicting with the authors' current interpretation of the data.

2) The relationship between movement, movement errors, and CbN spiking is unclear. Figure 8 seems intended to show that locomotion can proceed normally as long as PC and CbN neuron firing maintains the same phase relationship with respect to movement during light stimulation (modulated in the same way, with only a change in spike rates). Conversely, the authors suggest that movement is preferentially disrupted when the relationship between spiking and the step cycle is impaired. However, in panels F and G, the movements being compared are totally different. Hence, I don't understand the meaning of the firing rate correlations, which are no longer paired in a meaningful way with respect to movement. How is this different from a random shuffle of movement and spiking prior to correlation measurements? Moreover, if I understand panel I correctly, when prolonged movement traces are corrected by aligning with normal movement, the firing rates then show a similar relationship to movement as for non-slip/control strides. Do these data suggest that the relationship between spiking and movement is actually the same for slip and control trials, and that the only difference lies in the altered kinematics of movement, which is definitionally different for slips? From this analysis it is not clear how disrupted Purkinje cell spiking is causing the slip as opposed to reflective of the change in motor program. The authors should justify why it is reasonable to correlate firing rates across distinct movements, and explain more clearly how this informs the role of CbN spiking in motor output.

3) How do light steps vs light trains differentially affect the relationship between CbN spiking and movement across the step cycle? Figure 8 implies that disrupting the specific phase relationship between PC/CbN spiking and movement is a central cause for movement errors. Figure 9 shows that light trains are less likely to cause movement errors than light steps for equivalent mean changes in spike rates. The authors suggest that together these observations imply that light steps are more likely to disrupt the spiking/movement phase relationship. This is a central point, but it is not demonstrated. To make sense of how light steps and trains differ in causing movement errors, it is necessary to show how they differentially affect the relationship between spiking and the step cycle. This should also be shown for different light train frequencies as well, as it remains unclear why different frequencies produce differential motor error probabilities despite producing the same net effect on spiking.

4) It is unclear whether movement errors relate to PC synchrony, or just irregularity in firing. Rodent models of cerebellar ataxia have suggested that mean PC firing rates in ataxic animals are not different from control animals with normal coordinated gait, but instead differ in regularity of firing (as measured by CV). It seems that a major difference between the pulse and train stimuli delivered here might simply be a change in firing regularity instead of a difference in population synchrony, but this is never measured. In particular, Figure 9N shows that for the same mean firing rates, light steps are more likely to produce movement errors than pulse trains. Could this simply be due to differences in firing CV? Analysis of firing regularity across conditions should be considered as an explanation for the effects on movement.

*Reviewer #3:*

1) This is a well-written, high quality study investigating how the temporal organization of Purkinje cell activity affects CbN activity and locomotor behavior. Unfortunately, weaknesses in experimental design undermine the main conclusions of the paper. The paper is motivated in two ways – both as examining the role of Purkinje cells/cerebellar nucleus neurons in locomotion and as examining the physiological relevance of synchronous vs. asynchronous Purkinje cell activation. Most of the conclusions are really about the latter, but the Title and Introduction, as currently written, are overly focused on the former.

2) The authors use optogenetics, with continuous vs. pulsed illumination, to generate 'asynchronous' vs. 'synchronous' responses in Purkinje cells, respectively. But, only single unit responses are shown, and the assumption that continuous stimulation will give asynchronous responses across the Purkinje cells population is not validated. The authors only claim that Purkinje cells spike rates are increased without 'consistently affecting spike timing' – but this is across trials, not across neurons. Any Purkinje cell synchronicity that exists in vivo would be largely due to shared inputs, which as the authors acknowledge, are still there in the optogenetics experiments. So even if an individual Purkinje cell fires with different temporal patterns in response to continuous light stimulation on subsequent trials it is still very possible that a set of Purkinje cells will respond synchronously with each other on those same trials. Establishing whether the stimulus conditions used here result in synchronous or asynchrounous activity across Purkinje cells on individual trials – which is crucial to support the authors' conclusions – would likely require recording or imaging across multiple Purkinje cells (or CbN) simultaneously.

3) A related point: it was interesting that the relative inefficacy of trains of stimuli in changing CbN firing rates or evoking slips was frequency dependent. That made me wonder whether differences in short term plasticity could be responsible. If so, it may not be the synchronous nature of the response to the trains that matters, so much as the fact that it is extremely regular. Would trains with more natural temporal patterns (drawn from a typical distribution of Purkinje cell interspike intervals) be more effective? With the current experimental design, synchronicity and regularity are conflated.

4) Given these caveats, I have a general worry that the findings about continuous vs. pulsed stimulation may be a consequence of the relatively unnatural case of optogenetic stimulation, rather than telling us anything fundamental about the role of physiological synchrony, which is what the authors want to investigate.

5) The authors state that "Since the goal was to track continuity or discontinuity of the stride, rather than to correlate neuronal activity with actions of specific limb muscles, only the hindlimb was monitored." For the experiments that aimed to look at synchronous vs. asynchronous activation, this may be OK. However, given 1) that the stance phase for a front paw is (to a first approximation, and depending on walking speed) the swing phase for the ipsilateral hindpaw, and 2) the important role of the cerebellum in interlimb coordination in mice (Machado et al. 2015, Vinueza Veloz et al. 2014), only having measurements of the ipsilateral hindpaw is limiting for any analysis of stride phasing of recorded responses (e.g. all of Figures 3,4). Moreover, given the nature of the treadmill used, if one paw's movement is altered by optogenetic stimulation, the treadmill movement will change, and all paws will appear to slip. This could be one reason for the seemingly long (160 ms) latency between stimulation and slips that the authors report. Careful latency comparisons of front vs. hind paw kinematics, or EMG recordings, would be necessary to conclude that any slips (or neural responses) observed were due to changes in the ipsilateral hindpaw (measured) and not secondary consequences of primary direct effects on ipsilateral front paws.

6) The authors are fairly careful in their wording, to say 'modulated CbN firing is necessary for normal, ongoing locomotion,' but I am concerned that most readers will read this as 'modulation of CbN firing is necessary…', which has not been shown. Acute shutdown of spontaneously active neurons cannot reveal whether tonic activity or modulated activity was required. The authors clearly know this, but they should address it explicitly, and may want their main conclusions (Impact statement, Abstract, first line of Discussion) to be less susceptible to confusion on this point.

[Editors' note: further revisions were requested prior to acceptance, as described below.]

Thank you for sending your article entitled "Control of voluntary and optogenetically perturbed locomotion by spike rate and timing of neurons of the mouse cerebellar nuclei" for peer review at *eLife*. Your revised article has been evaluated by two of the original peer reviewers, and the evaluation has been overseen by Gary Westbrook as the Senior Editor.

Summary:

The revised manuscript was seen by two reviewers, both of whom have substantial remaining concerns. Under usual circumstances, *eLife* would not consider a manuscript further at this point. However, the reviewers recognize that the raw data are very high quality and the results are interesting, but the interpretation and presentation fall short. The study has evolved its focus to understand how the level of coherence of Purkinje cell inhibition influences the way neurons in the cerebellar nuclei support locomotion. This is an interesting point, but what is missing is direct evidence for a relationship between these phenomena in awake animals.

The reviewers discussed whether the authors could conduct a single experiment, involving simultaneous LFP recording in the cerebellar cortex and optogenetic stimulation of Purkinje cells during locomotion. This experiment would allow for single trial analysis that would relate the presence or absence of behavioral slips to the degree of coherence in the cortical LFP, which if successful could provide support for a direct relationship.

We also suggest that the authors restrict their analyses to the period of time (<120 ms) preceding the slips rather than the whole 1-second optogenetic stimulations (both steps and trains). The stated claim in the manuscript is that steps produce less synchronous inhibition, and this is true over the whole second stimulation. However, there are very clear oscillations in the anesthetized LFP recordings during the first ~60 ms of the recordings in both conditions, which confuses the claim that optogenetic steps produce a lower degree of synchronous firing than trains (over the time interval relevant for behavior). This is why single trial analysis during behavior is a crucial experiment, and the inherent trial-by-trial variability of spiking may be a useful 'bug' that can be exploited to provide the necessary support for the authors' claims.

Before we make a final decision on the manuscript, we would like the authors to address whether they think this approach is something they wish to pursue. If so, please provide an action plan and a time frame for us to consider. The original comments of the reviewers are also included below.

Reviewer #1:

The authors have added new experiments and analyses to their manuscript, which is now significantly improved. Specifically, they have added some additional analyses as well as LFP recordings from the cerebellar cortex in anaesthetized mice during their optogenetic stimulation protocols.

I have the following comments about the new experiments:

1) The LFP recordings demonstrate – when viewed over the whole 1 s stimulation time window – that there are obvious differences in the coherence of Purkinje cells between step and train stimuli. What is worrying, however, is that in the initial ~60 ms, there is a clear oscillatory response present in their recordings (most probably reflecting synchronous firing). Given that the average latency to a slip is ~120 ms (reduced from 160 ms in the initial manuscript), this could be an important time frame for consideration, especially if the hindlimbs are 'followers' of a forelimb slip.

Related to this point, the authors bin across the PC and CbN firing rates in Figure 6 over 50 ms, which prevents the assessment of short term synchronous dynamics. In Figure 7, meanwhile, they assess spiking over the relevant (single millisecond) timescales.

2) It is a little disappointing (though understandable given the technical challenge) that the authors did not perform these LFP recordings under the same conditions as the rest of the experiments in the manuscript – i.e. in awake, running mice. While the conditions for the requested LFP recordings were not explicitly stated in the "essential reviews", it would obviously have been better to have these results obtained under the same conditions, especially considering the comments of Reviewer 3, Major point 3. Nevertheless, having seen the data from anaesthetized mice, it would be helpful to know whether:

(a) Varying the duration of light steps (say down to the 60 ms range) produces similar results to the 1 second stimulation, and

(b) Whether non-slip trials show a greater degree of LFP oscillation (Purkinje cell synchrony) than slip trials.

If the authors have this data it would be helpful to include it in the final version of the manuscript.

3) The methodological description of the LFP recordings is a little confusing. The authors state that (1) the fiber was normally ~4-5 mm from the tip of the pipette, (2) the CbN recordings were made 1.9-2.5 mm below the cerebellar surface, and (3) for LFP recordings, the pipette tip was broken at the level of the optic fiber and placed as if for CbN recordings, implying a 1.5-3.1 mm distance (4-5 minus 1.9-2.5) between the recording pipette and cerebellar surface. Is this correct?

Reviewer #3:

While the authors have added helpful new experimental data, I continue to have major difficulties with this paper. I am unconvinced that the revisions have resolved the serious issues raised by all three reviewers in the previous round. I outline my major concerns below. In summary, in my view, once the caveats with the experimental approach that must be considered are taken into account, the conclusions left do not constitute a major advance in our understanding of cerebellar circuit processing and locomotion.

1) Here is the logic as it is laid out by the authors in the abstract (and elsewhere):a. 'Trains suppressed CbN firing less effectively' than steps;b. Trains 'altered spike timing';c. Steps or trains that perturbed 'stride related modulation' (which the authors confusingly use to refer to timing) correlated well with irregularities of movement;d. However, 'Unperturbed locomotion continued more often during trains than steps'.

In other words, trains altered spike timing more effectively, but didn't cause as many slips, yet the authors' conclusion is that modulation of spike timing is what matters for inducing slips. As the authors acknowledge with the 'however', this is fundamentally contradictory. If trains truly altered spike timing more effectively, and spike timing cause irregularities in movement, then trains simply should not allow locomotion to continue unperturbed more often than during steps. The authors speculate that 'the modulation of CbN spiking required for locomotion may persist more easily during synchronous inhibition' and that this could account for the apparent discrepancy, but this is highly speculative, not to mention extremely confusing for me as an expert reviewer, and presumably even more so for a broad readership. Insufficient data is presented to support this idea and in my view, it cannot be used to explain the apparently contradictory findings, given the necessary caveats in interpreting the experiments as they were done (using sudden disruption of neural activity with optogenetics).

2) I strongly disagree with the repeated claims in the manuscript and the rebuttal that the finding that ongoing locomotion is perturbed by optogenetic stimulation provides any evidence at all that 'modulated CbN cell firing is necessary for normal, ongoing locomotion.' I pointed this problem out in my previous review, at which time I thought that the authors had simply not been careful with their language. However, now they are insisting on this point. To show that modulated CbN firing is necessary for normal, ongoing locomotion it would be necessary to effectively clamp CbN firing at a certain level and prevent it from modulating with the stride. This is not currently possible with optogenetic stimulation in spontaneously active neurons and the kind of acute perturbation of spiking presented here is particularly unsuited to making any claims about what natural patterns of spike timing may be doing during locomotion.

(See for example:

1. From the rebuttal: 'we can safely conclude that transient disruption of CbN output rapidly induces discontinuities of gait' [Ok] and 'we can begin to infer that cerebellar output, and indeed modulated cerebellar output, is necessary for smooth, learned locomotion'. [No, we cannot; this has not been shown and is presented as a primary result of the manuscript].

2. From the rebuttal, 'These observations… permit us to conclude that cerebellar output consisting of modulated CbN activity is required even for well-trained ongoing repetitive movements like locomotion.' Again, I fundamentally disagree that they do.)

3) There are many statements throughout the paper and rebuttal (including the statement quote in point #2 above) relating to the idea that one important contribution of this paper is to demonstrate that sudden optogenetic perturbations of CbN neuron activity perturbs even well-practiced, ongoing locomotion. But this is entirely expected. It has been shown repeatedly, for example by the Lena, Medina, and Otis groups that optogenetic perturbation of Purkinje cell activity affects movement acutely in naïve animals. Disrupting movement of a limb of a mouse on a freely rotating cylinder will obviously, necessarily, induce 'slips', particularly given that the authors are not even claiming that the slips they observe are a direct result of perturbation of control of the hindlimb (this is in fact their main defense for not analyzing forelimb movements.) Moreover, 'To what extent CbN cell activity is assistive, predictive, reactive, or generative remains open to discussion' is used to motivate this study. But this study does not speak to those questions. CbN neurons are deep within the brain. They are not purely sensory or motor. Disturbing their activity with a sudden optogenetic perturbation of many CbN neurons at once perturbs movement, as expected, because they project directly to motor output areas. But this does not inform what signals drive their normal activity under physiological conditions.

4) According to the authors, after the primary demonstration that disruptions of CbN activity perturb movement, their second main result is the demonstration that 'steps and trains of light that disfavor or favor Purkinje cell synchrony evoke different responses in CbN cells and different behavioral outputs.' Although the authors have added LFP data to support the idea that the optogenetic stimulation resulted in more synchrony with trains than steps (though steps also induced synchrony for the first ~100ms or so), I am still not convinced that the paper presents results that speak to the question of how physiological PC/CbN synchrony might affect locomotion. In some places in the rebuttal for example, the authors admit that they cannot speak to physiological synchrony. But sometimes they say things like: 'The results provide a framework for interpreting systems-level observations in light of observations from cellular physiology.' The link here between the two levels is too tenuous and statements like this are unsupported by actual data. Moreover, if their claims are limited to claims about optogenetic synchrony (which I think they necessarily are), I do not see this as a meaningful advance.

5) The first part of the paper is meant to be a description of PC/CbN activity in mouse cerebellum during locomotion, which the authors argue, and I agree, is a necessary step in addressing their later questions. However, this analysis is incomplete and basic aspects of this characterization are lacking. For this part of the paper to represent an advance on its own, given the many similar published studies from other species, the authors would need to thoroughly analyze the speed dependence of the physiological responses they observe, for example. Given that they cannot speak to whether they were in a fore or hindlimb region, they would also need to at least consider changes in interlimb coordination and timing across speeds, to assess whether those could account for changes in neural responses, either across speeds in the same cells, or even possibly across cells, if the animals were locomoting at different speeds in different recordings. For example, in Figure 4, could changes in interlimb phasing across speeds account for these different categories? How common are the different categories of responses across speeds? Are these even distinct categories of responses? Similarly, in Figure 3, fifth paragraph of the Results, Figures 3E, F do not show that across speeds/stride durations 'firing rates tended to rise and fall at consistent phases of the stride cycle for both Pkj and CbN cells'. 3E, F are each one representative neuron, and there is no analysis of the relative phase modulation at different stance durations. The fact that there is still an average modulation when normalizing for stride duration does not speak to whether there is a consistent dependence of phase on stride duration. This is relevant, again, because fore/hind limb phasing also depends on speed.

6) In the sixth paragraph of the Results and throughout the paper: The modulation index and the subsequent use of the word 'modulation' to refer both to changes in spike rate and timing is extremely confusing. In the rebuttal the authors try to explain that they are using modulation to refer to changes only in temporal patterns but the definition itself is spike-rate based. This needs to be fixed.

[Editors' note: further revisions were requested prior to acceptance, as described below.]

Thank you for resubmitting your work entitled "Control of voluntary and optogenetically perturbed locomotion by spike rate and timing of neurons of the mouse cerebellar nuclei". Your revised article and responses have been evaluated by Gary Westbrook (Senior editor) in consultation with two reviewers. Following this revision, the conclusions of the paper are now more in harmony with the data with respect to interpretation. However the reviewers had several comments that we would like you to consider and address so that the Senior editor can make a final decision on the manuscript. These are included below. Some are primarily for your information, but some will require adjustments in the text.

1) Interpretationsa) The changes reflected in the rewriting of the first and the last sentences of the first paragraph of the Discussion, and the results are major improvements over their predecessors, which attempted to make larger claims about what was "required" during normal locomotion. These findings are now presented in a way that is adequately supported by the data.

b) The changes in the Discussion (which now reads "In either case, these data provide evidence that cerebellar output can actively participate in locomotion") and elsewhere are similarly crucial.

c) Synchrony: The authors have taken increased care in discussing physiological synchrony throughout the paper.

d) Terminology: Clarifications of the use of terms such as spike rate, spike modulation, and spike timing/synchrony are a major improvement. The authors could go further in this regard, perhaps using "depth of modulation" more consistently, and avoiding the term "spike timing" if at all possible (because "timing" alone still could refer to time relative to the stride). These suggested changes may be helpful for *eLife*'s broad readership.

2) Beyond interpretation, the major technical issue was to what extent synchronous vs. asynchronous responses had been demonstrated in response to trains vs. steps. The reviewers asked for a demonstration in awake animals that steps did not elicit synchrony because recordings in the anesthetized animal may not reflect the situation in awake animals. Without such a direct comparison, the conclusions need to be adjusted accordingly. Much of the toning down in the revised text helps here, but this caveat (anesthetized vs. awake LFP recordings, and the issue of the synchrony at the start of the steps) should be explicit in the Discussion.

3) Proville et al. did show that stimulating Purkinje cells during whisking caused whisker deflections (see Figure 6C of their paper). They call this a change in setpoint, and argue that it did not interfere with ongoing whisking. However, the stimulation did in fact induce a movement. If one changes the position (setpoint) of whiskers, the mouse can keep whisking (as Proville et al. found). However, if one changes, for example, the position (setpoint) of a front limb during ongoing locomotion, it will necessarily "disrupt locomotion" at the level measured in the current study – because it will prevent that limb from touching down at the expected time (if the paw is in the air) or it will directly perturb the wheel motion (if the paw is on the surface when the stimulation comes). The consequent change in support pattern or wheel motion will cause compensatory changes in the movement of all limbs, which will resemble hind limb "slips" as measured in the current study. Until now, the authors seemed to be assuming that their stimulations were directly disrupting locomotion itself – whereas it is entirely possible that stimulation simply induces movement of a limb, and that locomotor patterns are altered as a result. For example, the same slips in the hindpaws and the same changes in CbN activity might occur if one physically halted the mouse's front limb while it was in the air during a step. The authors now acknowledge this possible alternative interpretation at several points in the manuscript.

4) In response to the concerns about the completeness of the description of neural responses during locomotion, the authors state in their rebuttal, "Pilot analyses verified that the phase relationship between firing and the stride cells did not change with speed, and that eliminating especially long or especially short strides did not significantly alter the plots of stride-related modulation, justifying collapsing the data across stride durations." This is quite useful information that we recommend should be incorporated in the Results section.

5) Introduction, paragraph two: "Consequently, to what extent and during which behaviors CbN cell activity may be assistive, predictive, reactive, or generative remains open to discussion." This sentence should be deleted as its placement gives a misleading impression of what the paper is about. The question in the third paragraph asks the same thing in a way that is more relevant for the paper, and the same point is made elsewhere, including in the Discussion.

6) Paragraph beginning "The present experiments on voluntary locomotion in intact mice demonstrate that transiently suppressing modulation of CbN cell firing by directly altering Purkinje cell activity is sufficient to disrupt smooth, well-practiced locomotion”: Because the authors highlight the effects of stimulation (as opposed to inhibition) of Purkinje cell activity, they could be more specific here and say "stimulation of" instead of "altering." At the end of this paragraph, "In either case, these data provide evidence that cerebellar output can actively participate in locomotion" is a major improvement, as it does not infer causality that has not been demonstrated. The authors could also say something like "and cerebellar output is tightly linked to locomotor performance," if they wanted to strengthen this statement.

7) Subsection “Modulation of Purkinje and CbN cells during ad-lib locomotion”, passage "Here, we found that the pattern of modulation with respect to the stride across a given Purkinje or CbN cell did not depend strongly on stride duration; possibly the range of values assumed by variables such as pitch and roll was reduced by head fixation.": This is an interesting point, but it is also possible that this could be a species difference, perhaps resulting from speed-dependent interlimb phasing differences in rats vs mice.

8) Second paragraph subsection “Cerebellar output and the persistence of well-trained movements”: Suggestion: "Such observations suggest that such modulation normally seen in cerebellar output neurons… may facilitate but may not be necessary for locomotion." Delete: "represents sensory feedback associated with movement, which", because the paper does not address this issue.

9) In the same paragraph: Replace "may vary depending on the behavior and the conditions" with "is unclear".

10) The x axis label for the bottom (histogram) panels of 6A, B should be seconds not milliseconds.

---

## [Author Response]

Summary:All the reviewers thought the topic was very important and that the quality of the work is high. However, there were concerns as to the extent to which the results prove causality between the neural activity and locomotion, as most clearly stated in Reviews 2 and 3. Overall there was broad agreement among the reviewers concerning the specific comments raised by all reviewers regarding required changes to analysis and interpretation. Because the reviews include substantial amounts of text, we have included the reviews verbatim rather than try to summarize subtleties. We expect that addressing the comments will require additional analysis and perhaps some additional experiments. A few issues also arose in the Discussion that we would like you to address:1) Due to the lack of measurement of front paw movement and the previous recordings from other species, the focus of the paper might be best directed to the question of synchronous/asynchronous activity. However, synchrony vs asynchrony with the different optogenetic stimulation paradigms has not been demonstrated, and it needs to be. The authors could address this with LFP recordings from cerebellar cortex during continuous and pulsed stimulation (at different frequencies).

We have made and included the requested LFP recordings, which are shown as Figure 7—figure supplement 1 and discussed in the associated text.

2) An additional concern is the possibility of direct stimulation of Purkinje cell axon terminals, particularly in the experiments recording from CbN neurons as it appears that the distance between fiber and pipette tip was fixed in the experiments). We recognize that the recordings were done at a max depth of 2.5 mm with the fiber tip 4-5 mm above the pipette tip, thus there would have been a lot of scatter through the 2.5 mm of brain above the DCN. Nonetheless the targets of stimulation (Purkinje cells) were at a systematically different distance between the DCN recordings and Purkinje cell recordings.

Below, we clarify the comparisons that were made in the response to reviewers as well as in the text of the revised manuscript. The point was to stimulate Purkinje cells at any position (somata or axons). Comparisons were not made between data obtained from positions used for Purkinje stimulation and positions used for CbN cell stimulation. The LFP data directly illustrate the pattern of activity of Purkinje cells with stimulation positions used for recordings of CbN cells.

Reviewer #1:[…] Major comments:1) A main piece of analysis that is lacking is a description of any systematic relationship between the timing and types of optogenetic perturbations and the resulting defects in the animals' gait. Here are a few questions that could be addressed:- Does some feature of the underlying activity explain why the optogenetic stimulus is effective at causing 'slips' on some trials and not others?

Yes. Identifying the features of CbN cell activity that correlate with slips and non-slips is a major point of the manuscript. The first factor is firing rate. When the firing rate (measured in single CbN cells) falls below ~5 Hz, the slip probability is very high. This is a key point: it indicates that silencing cerebellar output disrupts even well-trained locomotion. These data are quantified throughout Figure 9. When firing rates are not reduced, if the stride-related modulation pattern is disrupted, slips are induced (measured and quantified in Figure 8). This is another key result: it shows that seeing a disruption of stride related modulation in a single CbN cell is indeed predictive of the slip, which one could not have anticipated with certainty before the experiments and analyses were done. This idea is now explained more fully in the text associated with Figure 8. A slip-inducing decrease in firing rate or disruption of modulation can be achieved with either a step or a train, but because trains are less effective at silencing cells, slips associated with trains more often show disrupted modulation than suppressed rate. This has been explained more fully in the text associated with Figure 9.

Regarding the questions below, please note that it was necessary to ensure that slips were relatively infrequent, since voluntary running by mice tended to cease when slips were evoked too often. Therefore, stimuli were at a threshold level, necessarily limiting the data set. We have inserted text, which addresses many of the points below, in the Results to clarify the stimuli used.

- Is there a particular phase of the step cycle which was more 'vulnerable' than others to the optogenetic perturbation?

No. The latency of the slips was 120 ± 17 ms for steps and 117 ± 13 ms for trains. We clarify that these latencies were independent of the step phase.

- Is the stimulation more likely to cause a 'slip' at certain running speeds than others?

Slips did not correlate with speed but with attributes of the firing patterns in the CbN. This is now directly evident as the distribution of slope of stance (rate of paw backward movement) and slope of swing (rate of paw forward movement) overlaps for trials that led to slips and non-slips, now shown in Figure 5—figure supplement 1.

- Did trains or steps induce greater numbers of different types of slips over the course of single recordings (i.e. the same cell)?

No. Each stimulus was repeated 3-10 times and slips did not occur earlier or later in the series of repeated stimuli, clarified in the Results.

In general, it is unclear if trains, steps and different powers were applied over the course of single recordings or whether each cell was only subjected to a subset of stimuli.

Multiple replications of each stimulus condition were applied to each cell, clarified in the Results.

Comparing across stimuli within cells, when the optogenetic stimulus was constant, would be a useful group of analyses.

The analysis of Figure 9N compares across stimuli within cells (with rate-matched trials).

- Is there a systematic relationship between the type of 'slip' – prolonged, altered, or incomplete stride – and any of the abovementioned criteria?

The descriptors “prolonged,” “incomplete” and “arrested” do not separate slips into three different categories or types. They are attributes that were indicative of a slip (now quantified and illustrated in Figure 5—figure supplement 1), but most slips (632/800, or 79%) had more than one of these attributes (“altered”), so a separation among slip “types” would not be appropriate or meaningful. This is clarified in the description of slips (paragraph five, subsection “Alterations of gait induced by optogenetic activation of Purkinje cells”).

These analyses would substantially improve our understanding of how cerebellar activity can affect ongoing motor programs. Otherwise, this study could be seen as being largely confirmatory, validating a set of previously documented findings with contemporary methods (optogenetics).

As indicated above, a number of the requested analyses were in fact done, and our textual revisions clarify these points. Nevertheless, independently of these finer grain analyses, the work is not simply confirmatory. Reversible manipulations of neuronal activity during locomotion have not previously been made. The first half of the present work is intentionally descriptive, since a baseline of measurements is necessary to the investigative part of the paper, and since the results from cats on motorized treadmills from Armstrong and Edgley 1984a and 1984b cannot be extrapolated to the voluntarily running head-fixed mouse. The second half of the work manipulates Purkinje cell activity, and brings us toward the realm of causality. The slips are certainly caused by the optogenetic stimulation of Purkinje cells (see quantification of light without ChR2 below and the new automated validation of slips); the question remains of how many intervening steps are involved, but since trained mice rarely slip on the treadmill, and since slips only occurred consistently with measurable disruptions of CbN activity, we can safely conclude that transient disruption *of CbN* output rapidly induces discontinuities of gait. From these results, we can begin to infer that cerebellar output, and indeed modulated cerebellar output, is necessary for smooth, learned locomotion. This has not been shown and is the primary result of the manuscript. The secondary result is the demonstration that steps and trains of light that disfavor or favor Purkinje cell synchrony evoke different responses in CbN cells and different behavioral outputs. These points have been re-emphasized by textual revision throughout the manuscript.

2) It is somewhat worrying that 'slip' events were detected manually and (apparently) without blinding. It would be good if the authors could develop automated criteria for what they call a 'slip'. Paw tracking is admittedly a difficult analysis challenge, but formalizing it in order to remove potential bias is important. This is particularly worrying in a few of the examples shown, which seem to contain high frequency events (e.g. the paw trace in 5A, left panel just at the end of the optogenetic step) that look aberrant. Are these errors in the paw tracking itself or actual movements?

We developed an automated analysis method and re-analyzed all trials, described in the Materials and methods as well as Results. The attributes of all slips are now illustrated in Figure 5—figure supplement 1. The automated analysis extracts stride duration, stance slope and swing slope and compares these values for strides before the stimulus to those during the stimulus. The method provides a stringent identification of slips; as described in the Materials and methods, false positives are low. The analysis led to a rejection of only 7% of the slips identified by eye (which was in fact blinded, now stated, in subsection “Analysis of optogenetically induced slips”). Consequently, although a few numbers have been updated, no changes in results or interpretation come from the reanalysis. We also clarify that the brief transients are noise from when the tracker transiently lost the paw signal and do not reflect actual movement (subsection “Identification of strides”).

3) The authors should show explicitly that their optogenetic perturbations (light flash + ChR2) induce stride perturbations at higher rates than in baseline (no light flash) and negative control (light flash but no ChR2) conditions. Description of these measurements is mentioned deep in the Materials and methods section, but no data are shown. Adding these comparisons in various places throughout the paper (for example in Figure 8, or as a supplement) would contextualize interpretation, since variability in step-cycle PC firing is related to other behavioural parameters (see Sauerbrei et al., 2015).

As the reviewer noted, we had described these control experiments in the Materials and methods. We moved the text to the Results to incorporate it better into the narrative, and added a sentence to stress that the data provide evidence that the optical stimulation of Purkinje cells is responsible for most slips. Because these slip events in control mice were so rare (1 in 77 trials, 1.3%, without ChR2, compared to 31/33 trials, 93.9%, with ChR2), and because the investment in surgery and training mice was quite high, it did not make sense to run the complete set of experiments on many control mice.

Like Sauerbrei et al., 2015, who studied freely moving rats, we see modulated Purkinje cell firing across the step cycle as well as variability of stride duration. Sauerbrei et al. indicate that some of the variability comes from acceleration, roll, and pitch parameters as the animal moves freely. Sauerbrei et al. also illustrate that the activity simultaneously recorded Purkinje cells co-vary, so the variability is not just noise. This piece of data informs the present study in two ways: (1) they help account for why the optical stimulation may be effective on some trials and not on others, and (2) they provide evidence that synchronized Purkinje cell activity may be a real aspect of cerebellar function during locomotion. These points are now included in the Results and the Discussion.

4) Complex spiking in Purkinje cells appears to be completely neglected in this study. The authors should at least attempt to extract complex spikes from their recordings and analyze their response properties during the optogenetic stimulation. Is complex spiking increased during 'slips' to a greater extent than in non-slip trials? Related to the point above – does the flashing of the optogenetic stimulus induce complex spiking (this is likely a novel/unexpected sensory stimulus)?

Complex spikes were intentionally not studied because the questions under investigation are about firing patterns in CbN cells and their relation to simple spike mediated inhibition from Purkinje cells and to ongoing practiced movement. We originally provided information on complex spike rates in the Materials and methods, and we now added measurements from additional Purkinje cells and separated firing rates into windows before, during and after the light step. As expected, complex spike rate is low (1-2 Hz) and can go up during the light-induced activation of Purkinje cells (2-3 Hz), which is predicted to inhibit nucleoolivary cells, leading to disinhibition of the inferior olive. Because it is not possible to disambiguate the direct effects (olivary disinhibition by increase Purkinje cell firing) from biological effects (e.g., sensory or motor input to the inferior olive), further analysis is not likely to be informative. We have therefore left this as incidental information in the Materials and methods for completeness but have not pursued the studies further.

5) The authors mention in the Materials and methods that a subset of their neurons were modulated by running speed. Was this consistent across individual step cycles? If so, it would be a departure from the findings of Armstrong and colleagues, who showed that firing rates, when normalized to step cycle, were invariant to different speeds of walking on a motorized treadmill. It would be valuable to know if during free running, the depth of modulation varies with speed (consistent with the findings of Sauerbrei et al., 2015).

As indicated in the Results, aligning strides to the lift revealed that relative firing rates rose and fell at consistent phases of the stride for both Purkinje and CbN cells across stride durations. Nevertheless, for completeness we tested whether firing rates varied with velocity. As reported in 8 CbN cells (21%), firing rate correlated with velocity. Because these were a minority of cells targeted in our recordings and because this question was tangential to the questions at hand, we reported the data but did not pursue it further.

Reviewer #2:How the cerebellum contributes to already learned, ongoing movement is an important and much debated question. While most studies have focused on the role of cerebellar circuits in motor learning, the idea that the cerebellum plays a significant role in ongoing movement has only recently become more widely accepted. However, the same modern optogenetic tools that have permitted causal demonstrations that Purkinje cell firing can drive motor output have also suggested an overly simplified relationship between Purkinje cell output, cerebellar nuclear cell (CbN) firing, and movement by implying a strict inverse relationship between Purkinje cell firing and CbN firing. In contrast, in vivo recordings from CbN neurons and Purkinje cells have suggested a more complex relationship between cerebellar output and movement. Here, Sarnaik and Raman seek to address the important question of how the cerebellum contributes to ongoing movement by combining in vivo recordings from both the CbN and cerebellar cortex in combination with optogenetic manipulations designed to test how the firing patterns of neurons in each location are causally related to motor output. By tailoring optogenetic stimulation of Purkinje cells in a manner intended to either promote or reduce population synchrony, in combination with high-speed measurements of hind limb position during locomotion, they conclude that Purkinje cell synchrony is favorable for normal movement, likely by creating widows of excitability for ongoing mossy fiber excitation. If true, this conclusion is significant, and I would be in favor of publication. However, I am skeptical of the central conclusion, as I am not yet convinced that the data supports the claim that enhanced synchrony is favorable for ongoing movement. Moreover, the relationship between movement, CbN spiking and synchrony overall is not fully explored, and paper would benefit greatly from further analysis regarding the relationship between these parameters.

The reviewer has laid out the questions clearly but has overstated our conclusions. We have no data on physiological synchrony during locomotion. The actual conclusions of the manuscript are derived directly from the results, which indicate that alterations of Purkinje cell activity that (a) suppress CbN cell firing rates or (b) disrupt temporal modulation of CbN cell firing rates on the time scale of a stride are associated with slips. These observations, which have to our knowledge have never been made explicitly before, permit us to conclude that cerebellar output consisting of modulated CbN cell activity is required even for well-trained ongoing repetitive movements like locomotion. As part of the series of manipulations, we compare different types of activation of Purkinje cells, namely asynchronous and synchronous firing. These can both induce slips, but influence CbN cells differently, with asynchronous inhibition producing lower firing rates. To control for rate (and other variables), we made within-cell comparisons of pairs of trials in which a step and a train led to comparable changes in firing rate. In these rate-matched trials, steps and trains often elicited a common response (either a slip or a nonslip). However, in 29% of these trials, a step but not a train elicited a slip, or vice versa—and step-slip/train-nonslip trials were nearly four times more likely than step-nonslip/train-slip. Since we know that slips correlate with loss of modulation and non-slips correlate with continued modulation, we can conclude that modulation can persist more easily with trains (synchrony) than with steps.

To illustrate this point more clearly, we have added Figure 10. The raw data for rate matched step and train response pairs from individual cells and all subsequent transformations are shown, to illustrate that, for either steps or trains of light, when slips occurred, stride-related modulation was disrupted, and when nonslips occurred, modulation persisted. In addition, the correlation between firing rate modulation during the slip and before the slip (shown for all slips from Figure 8H) is parsed by step or train as stimulus. The plots show directly that a higher fraction of step trials than train trials were non-modulated.

In the Discussion, we bring up previous observations (by others) of synchrony occurring under physiological conditions of normal movements. Given the evidence that CbN cell modulation is required for locomotion, and that modulation is less often disrupted by applied synchronous inhibition than applied asynchronous inhibition, we propose the idea that a consequence of physiological synchrony might by to permit modulation of CbN firing (e.g. by mossy fiber inputs) to proceed. We mention that the conclusions raise this idea as a possibility because it serves to put the experimental results into context and (we hope) direct future research. We have edited the text throughout to distinguish our interpretations relating to the artificial synchrony tested here and our proposals regarding physiological synchrony that may occur in vivo.

Major comments:1) Movement errors may actually be highest when PC synchrony is highest for light pulse trains. Based on Figure 7, it is challenging to interpret the relationship between Chr2 train frequency and PC synchrony. Because PCs were recorded one at a time, it is unclear how light spreads from the stimulation/recording pipette to the surrounding tissue activate nearby PCs. A nearby LFP electrode could have been useful to assess relative population synchrony in absence of distance dependent paired PC recordings. However, from the current measurements, it is clear that peak PC spike probability is inversely proportional to train frequency, strongly suggesting the likelihood of decreased synchrony in the population with increasing stimulation frequency. This hypothesis is consistent with the CbN recordings. Here, the largest suppression (peak transient reduction in spike probability) happens for the lowest stimulation frequency of 50 Hz. This could suggest PC synchrony is highest for the 50 Hz stimulation, because CbN neurons integrate 50-80 PC inputs and suppression should be largest when synchrony is highest. It seems unlikely that the difference in CbN suppression is purely due to short-term depression from PC synapses, because the spike probability in PCs at 100 Hz is nearly half that at 50 Hz, which should normalize for depression between these frequencies. Hence, though the measurements are indirect, both PC and CbN recordings suggest highest PC synchrony at 50 Hz stimulation. Comparing these data with Figure 9, the highest probability of movement error occurs for 50 Hz stimulation. More specifically, in Figure 9L, though mean spike rate is invariant, motor errors decrease dramatically with simulation frequency. In combination with the data shown in Figure 7, these results suggest the possibility that with the same type of stimulation (pulse trains), motor errors occur with higher probability as PC synchrony increases, conflicting with the authors' current interpretation of the data.

Everything the reviewer states is correct, except the last phrase. We have revised the text throughout to remove the mistaken impression that we have claimed that extreme artificially induced synchrony will facilitate movement. Based on the data, this claim would indeed be incorrect. The data indicate that what facilitates movement is *stride-related modulation of CbN cell spiking*. Stride-related modulation of spiking can be disrupted by (1) inhibition that reduces the rate of firing enough to suppress CbN cell output altogether or (2) inhibition that alters the stride-related timing of spikes whether or not the rate is suppressed. As the reviewer correctly points out, compared to the other train stimuli tested here, 50-Hz stimulation of Purkinje cells is an unusually strong pattern of artificial regular inhibition. We have added LFP recordings, which confirm this statement. It effectively disrupts stride-related modulation by retiming the spikes. Nevertheless, as described in the response to the reviewer’s general comments, when pairwise comparisons are made, controlling for rate changes, even artificially regular trains with no relation to the stride cycle are less likely to induce slips, i.e., they are less likely to interfere with the CbN cell firing rate modulation. These ideas are now emphasized still more strongly in the manuscript in the text associated with Figure 9L and 9M, as well as Figure 10. These observations raise the possibility that if Purkinje cell activity during stepping were to be correlated across cells (as suggested by data of Sauerbrei), then inhibition might be interspersed with physiologically appropriate gaps in which mossy fiber activity would be able to generate spikes more effectively. We bring up this possibility in the last paragraph of the manuscript, as it is a rational inference from the data, which might be helpful in future experimental design, even though it is not tested in the present study.

2) The relationship between movement, movement errors, and CbN spiking is unclear. Figure 8 seems intended to show that locomotion can proceed normally as long as PC and CbN neuron firing maintains the same phase relationship with respect to movement during light stimulation (modulated in the same way, with only a change in spike rates). Conversely, the authors suggest that movement is preferentially disrupted when the relationship between spiking and the step cycle is impaired. However, in panels F and G, the movements being compared are totally different. Hence, I don't understand the meaning of the firing rate correlations, which are no longer paired in a meaningful way with respect to movement. How is this different from a random shuffle of movement and spiking prior to correlation measurements? Moreover, if I understand panel I correctly, when prolonged movement traces are corrected by aligning with normal movement, the firing rates then show a similar relationship to movement as for non-slip/control strides. Do these data suggest that the relationship between spiking and movement is actually the same for slip and control trials, and that the only difference lies in the altered kinematics of movement, which is definitionally different for slips? From this analysis it is not clear how disrupted Purkinje cell spiking is causing the slip as opposed to reflective of the change in motor program. The authors should justify why it is reasonable to correlate firing rates across distinct movements, and explain more clearly how this informs the role of CbN spiking in motor output.

The underlying assumption that the reviewer is making is firing rate in an individual cell is correlated with movement, but that is an assumption that cannot be validated without these experiments and analyses. Only afterward can we be confident that the single-cell firing rates and movement are linked. The text associated with Figures 8-10 has been edited to stress these points. The key ideas are as follows:

In panels 8F and 8G, what is being correlated is not movement but firing rates before and after stimulation. To interpret the rest of the data, we must verify that firing patterns of individual neurons can be used to make inferences about movements. What we observe (Figures 8B, 8C, 8D) is that when movements are not perturbed by light stimulation (non-slips), firing rate modulation continues as it does in the unstimulated condition (although rates can change). This observation provides evidence against the idea that movement persists normally even when firing rates in the recorded cell are altered.

Then we check the converse: does modulation of firing rates in the recorded cell continue even when movement is disrupted? Please refer to Figures 8F and 8G. We test the correlation between the firing rate during the normal stride (no light) with the firing rate over an identical time period during the slip. If the slip is independent of the activity in the recorded cell, firing rate modulation might persist regardless of the movement. But instead, we see that modulation is disrupted (as is the movement). Like the measurement of modulation during non-slips, this is a crucial control that permits us to say that modulation in the CbN cell is predictive of whether or not there is a slip. This is a necessary baseline for the analyses of Figure 9 (and 10).

The prolonged stride is a special case because we can identify strides clearly, and see that firing rate modulation now occurs on a longer time scale. This is interesting because it strengthens the relationship between the modulation in a single CbN cell and the locomotor output. Regarding causality, it is reasonable to say that the optogenetic activation of Purkinje cells (whose output is to CbN cells) caused the slip, but we cannot (and do not) make claims about whether it is a proximal or distal cause. Nevertheless, what matters here is to test whether a single-cell readout can be used to make inferences about movement. The text associated with Figures 8-10 has been edited to re-emphasize these points.

3) How do light steps vs light trains differentially affect the relationship between CbN spiking and movement across the step cycle? Figure 8 implies that disrupting the specific phase relationship between PC/CbN spiking and movement is a central cause for movement errors. Figure 9 shows that light trains are less likely to cause movement errors than light steps for equivalent mean changes in spike rates. The authors suggest that together these observations imply that light steps are more likely to disrupt the spiking/movement phase relationship. This is a central point, but it is not demonstrated. To make sense of how light steps and trains differ in causing movement errors, it is necessary to show how they differentially affect the relationship between spiking and the step cycle. This should also be shown for different light train frequencies as well, as it remains unclear why different frequencies produce differential motor error probabilities despite producing the same net effect on spiking.

We have added Figure 10 which shows paired examples of steps and trains that elicited similar firing rates and illustrates the disruption of modulation during slips but not non-slips (for both steps and trains), as well as the higher fraction of slip trials with non-zero firing rates, which were non-modulated, for trains vs. steps (from data in Figure 8).

4) It is unclear whether movement errors relate to PC synchrony, or just irregularity in firing. Rodent models of cerebellar ataxia have suggested that mean PC firing rates in ataxic animals are not different from control animals with normal coordinated gait, but instead differ in regularity of firing (as measured by CV). It seems that a major difference between the pulse and train stimuli delivered here might simply be a change in firing regularity instead of a difference in population synchrony, but this is never measured. In particular, Figure 9N shows that for the same mean firing rates, light steps are more likely to produce movement errors than pulse trains. Could this simply be due to differences in firing CV? Analysis of firing regularity across conditions should be considered as an explanation for the effects on movement.

We measured CV in all step and train trials. As expected, the CV is lower in the trains than in the steps, consistent with regularization by the optically induced regular synchronous inhibition, but was not predictive of whether a slip or non-slip would occur. We have included this information but this question seems to be predicated on a misunderstanding. The rodent studies that the reviewer is presumably referring to show regular/irregular firing in slices without synaptic input, not regularity or irregularity in vivo, and as such are not directly comparable or relevant to the present work. In the present experiments, regular trains of synchronous inhibition would be the mechanism underlying the outcome of a reduced CV.

Reviewer #3:1) This is a well-written, high quality study investigating how the temporal organization of Purkinje cell activity affects CbN activity and locomotor behavior. Unfortunately, weaknesses in experimental design undermine the main conclusions of the paper. The paper is motivated in two ways – both as examining the role of Purkinje cells/cerebellar nucleus neurons in locomotion and as examining the physiological relevance of synchronous vs. asynchronous Purkinje cell activation. Most of the conclusions are really about the latter, but the Title and Introduction, as currently written, are overly focused on the former.

We tried to be systematic in building up to the key experiments and explicit about the multiple variables that contribute to the phenomena under study. Therefore, the manuscript begins with a descriptive portion (Figures 1-4) that lays the foundation of Purkinje and CbN cell activity during locomotion before the experimental portion that manipulates Purkinje cell activity optogenetically. After illustrating the effects of the manipulation (Figure 5), Figures 6-8 provide tests, assessments, and validation of the manipulations (Purkinje and CbN activity during optogenetic stimulation; correlation of non-slips with continued modulation, hence interpretability of individual CbN cells). Only then do we proceed with the detailed analysis of slips (Figure 9 and now the new Figure 10), of which the comparison of steps vs. trains is a subset. We think this approach makes the experimental design quite solid, especially given the multiple variables that make investigative (rather than exclusively descriptive) experiments on the relation between neuronal firing patterns and behavior quite challenging.

The manuscript is actually not only (or even primarily) about synchrony vs. asynchrony, although previous work on this topic certainly helped motivate the study. As stated in other responses above, we think that the strongest experiments and results of the paper are as follows. We present some of the first recordings of Purkinje and CbN cell activity in awake, running mice both during voluntary locomotion and during transient, reversible, experimental elevations of inhibition to the cerebellar nuclei. These results show definitively that direct manipulations of Purkinje activity that either (a) suppress the activity of CbN cells, or (b) disrupt the stride-related modulation of CbN cells, produce measurable disruptions of locomotion. These results matter for the following reasons: The experimental manipulation is new and direct. The demonstration that tracking an individual CbN cell’s activity is highly correlated with behavioral output could not have been predicted without direct measurement. The observation that silencing the output of a CbN cell (which likely indicates comparable silencing of multiple CbN cells) is sufficient to alter movement provides evidence that cerebellar activity is required for persistence even of a well-trained motor task. The observation that disruption of modulation of a CbN cell (which likely indicates disrupted modulation of multiple CbN cells) is sufficient to alter movement provides evidence that modulated cerebellar output is required for persistence even of a well-trained motor task. None of these points could have been stated before the experiments were done. We have edited throughout to ensure that the emphases are appropriate to make these points clear.

2) The authors use optogenetics, with continuous vs. pulsed illumination, to generate 'asynchronous' vs. 'synchronous' responses in Purkinje cells, respectively. But, only single unit responses are shown, and the assumption that continuous stimulation will give asynchronous responses across the Purkinje cells population is not validated. The authors only claim that Purkinje cells spike rates are increased without 'consistently affecting spike timing' – but this is across trials, not across neurons. Any Purkinje cell synchronicity that exists in vivo would be largely due to shared inputs, which as the authors acknowledge, are still there in the optogenetics experiments. So even if an individual Purkinje cell fires with different temporal patterns in response to continuous light stimulation on subsequent trials it is still very possible that a set of Purkinje cells will respond synchronously with each other on those same trials. Establishing whether the stimulus conditions used here result in synchronous or asynchrounous activity across Purkinje cells on individual trials – which is crucial to support the authors' conclusions – would likely require recording or imaging across multiple Purkinje cells (or CbN) simultaneously.

First, we have made local field potential recordings from Purkinje cells with the optical fiber placed as it was for CbN cell recordings. These data confirm that trains but not steps evoke responses whose FFTs have power at the optical stimulation frequency (and harmonics), consistent with multiple cells firing simultaneously at the applied stimulation frequency. These data are now included as Figure 7—figure supplement 1.

Second, we edited the text to clarify that the data in Figure 7 show the spike timings (in 1-ms bins) per condition averaged across all trials in all Purkinje cells (7C) and all CbN cells (7D). Although the cells are not recorded simultaneously, they are signal averaged with respect to the optical stimulus, so the data really are likely to reflect the response “across neurons.” If there were a consistent temporal relationship across Purkinje cell spikes, it would emerge in the mean PSTH for Purkinje cells and for CbN cells. Such patterns do emerge for trains but not for steps (grey mean PSTHs). Steps evoke comparatively constant spike probabilities in each 1-ms bin throughout the early and late windows, distinct from the patterns seen with trains, in which the spike probabilities consistently rise and fall (across Purkinje cells) or fall and rise (across CbN cells) in the same windows.

3) A related point: it was interesting that the relative inefficacy of trains of stimuli in changing CbN firing rates or evoking slips was frequency dependent. That made me wonder whether differences in short term plasticity could be responsible. If so, it may not be the synchronous nature of the response to the trains that matters, so much as the fact that it is extremely regular. Would trains with more natural temporal patterns (drawn from a typical distribution of Purkinje cell interspike intervals) be more effective? With the current experimental design, synchronicity and regularity are conflated.

We, too, wondered whether higher frequency trains are ineffective because inhibition by Purkinje cells is reduced by synaptic depression, but this idea is unlikely. The depth of modulation of Purkinje cells (Figure 7C, Figure 7—figure supplement 1) was lowest at high train frequencies, suggesting that fewer Purkinje cells fire on each cycle of the stimulus (even though every cycle has at least some Purkinje cells firing). Indeed, the mean firing rates of Purkinje cells were comparable at low and high frequencies (Figure 7E) giving rise to comparable firing rates of CbN cells across the range of frequencies tested (Figure 7E), suggesting that the total inhibition provided by Purkinje cells was not much lower for the high-frequency trains. Still more direct evidence comes from the analysis of rates in Figure 9. If inhibitory synaptic depression or failures of Purkinje cell spikes to follow the stimulus were the primary determinant of slips, then slips and non-slips would segregate either by absolute firing rate or by the size of the reduction in firing rate, which is not the case (Figures 9H-9M). This is in fact the reason why we did the analyses in Figures 9H-9M. These explanations have been added more explicitly to the manuscript (Results).

4) Given these caveats, I have a general worry that the findings about continuous vs. pulsed stimulation may be a consequence of the relatively unnatural case of optogenetic stimulation, rather than telling us anything fundamental about the role of physiological synchrony, which is what the authors want to investigate.

It is important to stress again here that this manuscript neither tests nor makes claims about physiological synchrony, either in the original or revised version. Optogenetic stimulation is in fact *completely* unnatural; it is an experimental manipulation that is being used to test how and with what probability movements are altered by interference with normal Purkinje cell firing patterns. We have edited throughout to try to clarify what we are actually studying (summarized below).

The question under investigation is what patterns and strengths of inhibition are likely to be most effective (and ineffective) at changing CbN cell firing in a manner that has behavioral consequences? Or as is stated more broadly in the Introduction, we are testing “how CbN cells respond to modulated inhibition from Purkinje cells and how these responses relate to behavior.” The choice of stimulus parameters is influenced by the awareness that biophysical studies of the Purkinje-CbN synapse indicate that firing by CbN cells is sensitive to the relative timing (the degree of coherence) of inhibitory inputs as well as the rate of inhibition, as well as the fact that both Purkinje cell simple spike synchrony and asynchrony has been repeatedly reported in vivo. We therefore manipulated two variables: the rate of Purkinje cell firing (with steps of different intensities, which keeps relative timing asynchronous) as well as the relative timing of Purkinje cell firing (with trains of different frequencies, which keeps firing rate relatively constant) to observe the effect on CbN cell firing (rates and patterns) as well as on behavior.

By doing so, we provide answers to the question under investigation: movement is altered when the firing rates of CbN are fully suppressed. Movement can also be disrupted when CbN cell firing rates are relatively unchanged. In the latter case, two explanations are possible. Either the cell’s firing *pattern* has changed and may correlate with the slip, or its firing pattern is unchanged and a single-cell readout is uninformative. We consistently find that the CbN cell modulation pattern changes during slips, and not during non-slips. Regardless of steps and trains, if modulation persists, locomotion is unimpeded even when CbN cell firing rates are strongly reduced (Figure 8C, 8D); conversely, when modulation is lost, locomotion is disrupted even when CbN firing rates are not reduced (Figure 8F, 8G, 8H).

Further analysis indicates that normal locomotion persists somewhat more during trains than steps (Figure 9N, new Figure 10). This means that trains (even artificial ones) are less effective at suppressing modulation. These experimental results let us think about biophysics, which indicates that mossy fiber excitation is (not surprisingly) most effective when it is not impeded by inhibition, i.e. during gaps in inhibition, which occur between synchronous IPSPs (Wu and Raman, 2017). In other words, if Purkinje cells synchronize, then their inhibition will be less likely to shunt modulated excitation. This lets us consider the physiological scenario. We know that under normal (unmanipulated) conditions, CbN cells are modulated. This raises the possibility, which we present in the discussion, that physiological Purkinje cell synchrony, which almost certainly is not a regular train of spikes if/when it happens, could be more permissive of modulation of CbN cell firing rates by incoming mossy fiber excitation. The present experiments do not purport to test physiological synchrony, but provide data that permit us to think about what would happen.

5) The authors state that "Since the goal was to track continuity or discontinuity of the stride, rather than to correlate neuronal activity with actions of specific limb muscles, only the hindlimb was monitored." For the experiments that aimed to look at synchronous vs. asynchronous activation, this may be OK. However, given 1) that the stance phase for a front paw is (to a first approximation, and depending on walking speed) the swing phase for the ipsilateral hindpaw, and 2) the important role of the cerebellum in interlimb coordination in mice (Machado et al., 2015, Vinueza Veloz et al., 2014), only having measurements of the ipsilateral hindpaw is limiting for any analysis of stride phasing of recorded responses (e.g. all of Figures 3,4). Moreover, given the nature of the treadmill used, if one paw's movement is altered by optogenetic stimulation, the treadmill movement will change, and all paws will appear to slip. This could be one reason for the seemingly long (160 ms) latency between stimulation and slips that the authors report. Careful latency comparisons of front vs. hind paw kinematics, or EMG recordings, would be necessary to conclude that any slips (or neural responses) observed were due to changes in the ipsilateral hindpaw (measured) and not secondary consequences of primary direct effects on ipsilateral front paws.

Yes, the reviewer is exactly right that multiple limbs may be affected and that the stride phasing cannot be determined. We expanded our statement of this idea and state paragraph four of subsection “Firing rate modulation with stride” that the phase relationship is shifted relative to Armstrong and Edgley’s tracking of forelimb (which brings the two studies much closer into alignment); the same idea is now stated earlier in the text. The most important part for the questions at hand, however, is that the Purkinje and CbN cells that we recorded from are in regions that project to one another, so the relative phase of modulation with respect to the hindlimb step cycle can be compared. The point of Figure 3-4 is not to relate the activity to a specific limb, but to look at the distribution of modulation patterns with respect to the stride and for Purkinje vs. CbN cells.

Regarding the optogenetics, which paw initiates the irregularity of movement is actually immaterial to the study and its conclusions. Pilot studies indicated that the hindlimb was the most reliably tracked during normal and altered locomotion. All slips included a change of hindlimb movement and correlated with a loss of modulation in the recorded Purkinje or CbN cell. The recording regions were consistent, as were the slip latencies. Therefore, it seems reasonable to conclude that the data can be analyzed to address the desired question:what alterations of CbN cell activity as a consequence of experimental manipulation of Purkinje cell activity give rise to disruptions of gait? We are not trying to say that the hind paw movement is causal of the slip, but it is a readout of the slip. We have added text (in addition to the places where it already was stated) to clarify these points. We also edited the sentences on latency in the Discussion.

6) The authors are fairly careful in their wording, to say 'modulated CbN firing is necessary for normal, ongoing locomotion,' but I am concerned that most readers will read this as 'modulation of CbN firing is necessary…', which has not been shown. Acute shutdown of spontaneously active neurons cannot reveal whether tonic activity or modulated activity was required. The authors clearly know this, but they should address it explicitly, and may want their main conclusions (Impact statement, Abstract, first line of Discussion) to be less susceptible to confusion on this point.

We edited the Abstract to define “modulated,” in addition to the definition that already was present in the Introduction. The data support the idea that slips occur when modulation is disrupted, even without suppression of firing. This is shown in Figure 8D: non-slips remain modulated, i.e. there is a high correlation between firing rate modulation before and during the light stimulus; 8H: slips become unmodulated, i.e., the correlation between firing rate modulation before and during the stimulus is lost. It is shown in another way in Figure 9ABCD; 9HIJK, note slips on unity line, i.e. with unmodulated (tonic) output. It is not really the case that it remains ambiguous whether tonic output is sufficient. The text associated with Figures 8 and 9 has been edited to make these points clearer.

[Editors' note: further revisions were requested prior to acceptance, as described below.]

Summary:The revised manuscript was seen by two reviewers, both of whom have substantial remaining concerns. Under usual circumstances, eLife would not consider a manuscript further at this point. However, the reviewers recognize that the raw data are very high quality and the results are interesting, but the interpretation and presentation fall short. The study has evolved its focus to understand how the level of coherence of Purkinje cell inhibition influences the way neurons in the cerebellar nuclei support locomotion. This is an interesting point, but what is missing is direct evidence for a relationship between these phenomena in awake animals.The reviewers discussed whether the authors could conduct a single experiment, involving simultaneous LFP recording in the cerebellar cortex and optogenetic stimulation of Purkinje cells during locomotion. This experiment would allow for single trial analysis that would relate the presence or absence of behavioral slips to the degree of coherence in the cortical LFP, which if successful could provide support for a direct relationship.We also suggest that the authors restrict their analyses to the period of time (<120 ms) preceding the slips rather than the whole 1-second optogenetic stimulations (both steps and trains). The stated claim in the manuscript is that steps produce less synchronous inhibition, and this is true over the whole second stimulation. However, there are very clear oscillations in the anesthetized LFP recordings during the first ~60 ms of the recordings in both conditions, which confuses the claim that optogenetic steps produce a lower degree of synchronous firing than trains (over the time interval relevant for behavior). This is why single trial analysis during behavior is a crucial experiment, and the inherent trial-by-trial variability of spiking may be a useful 'bug' that can be exploited to provide the necessary support for the authors' claims.Before we make a final decision on the manuscript, we would like the authors to address whether they think this approach is something they wish to pursue. If so, please provide an action plan and a time frame for us to consider. The original comments of the reviewers are also included below.

Thank you for this summary and the opportunity to respond to it. The requests for further revisions apparently stem from the desire for some key data and concerns about novelty and interpretation, centering on the following requests:

(a) Evidence for increases in synchrony of Pkj cells with trains and not steps,

(b) Shifting of the analysis window,

(c) Analyses of single trials,

(d) Curtailment of inferences, and

(e) Distinction from previous work.

We have edited the full text based on the editor’s and reviewers’ comments to address the points listed above. Below, we explain why the experiments and analyses already in the manuscript provide the answers they seek and why the reviewers’ newly suggested experiments and analyses would fail to answer the central questions. We also clarify key points of the work and its distinction from previous studies.

We hope that the edits and these explanations will clarify why we think the key points have been addressed.

(a) Evidence for increases in synchrony of Purkinje cells with trains and not steps:

The reviewers’ key question is whether Purkinje (Pkj) cells synchronize in response to trains.

Answers provided by existing experiments: In our original study, we provided evidence that synchrony was elevated by trains, from direct recordings from individual Pkj cells as well as from direct recordings from individual CbN cells. Since each CbN cell is targeted by a few dozen Pkj cells, the response of each cell reflects the combination of convergent inhibitory inputs. If inhibition (assessed as the decrease of spike probability in a 1-ms bin) of a CbN cell is measurably time-locked to the stimulus train applied to ChR2-expressing Pkj cells, the most reasonable interpretation is that the inhibition is coherent (relatively synchronous). This offers a direct measure of the activity of Pkj cells that are (1) stimulated by light and (2) converging. We found that individual CbN cells and the population of CbN cells indeed showed evidence of Pkj cell synchrony in the form of inhibition time-locked on the time scale of ms. This was true for trains of all frequencies, but not for steps (Figure 6B, 6D for steps, Figure 7B, 7D for trains).

Explanation of shortcomings of LFP recordings: Nevertheless, the reviewers requested LFP data from Pkj cells stimulated by steps and trains of light. To comply with reviewer requests, we obtained the data requested. The problem with LFP recordings, however, is that there is no way to know how many Purkinje cells contribute to the measurement or whether they converge in the CbN. Thus, LFP recordings are actually uninterpretable in the context of the question that the reviewers ask about. Although they provide corroborative evidence in light of the direct recordings, they cannot not stand on their own.

Evidence for lack of transmission of 50-ms LFP oscillation to the CbN: The reviewers now express concern that an oscillation at ~280-300 Hz seen in the first ~50 ms of the LFP recordings in two anesthetized mice during steps of light is responsible for all the slips detected. However, a major role for this brief oscillation can be ruled out because it is not present in the CbN cell records, even when the data are binned at 1 ms (Figure 7D, grey, now moved to front in the revision). These records show no coherent inhibition beyond the first 10 ms. The brief period of coherence is expected, given that the onset of light initially evokes spikes synchronously (as it does for a brief pulse), after which the firing rates of individual Purkinje cells diverge: after the first 10 ms, the mean spike probability in CbN cells remains below ~0.05 per bin with no systematic or cyclic variation throughout the interval, suggesting that the incoming inhibition must be asynchronous. Thus, the ~50 ms oscillation in the LFP translates into ~2 inhibitory gaps within ~10 ms in CbN cells and nothing more. Moreover, although steps and trains both start with coherent firing in the first 10 ms, they diverge in the CbN spiking behavioral outcomes they generate. The reviewers’ proposal that slips primarily result from the 50-ms oscillation in the steps therefore is unfounded, especially since all the other analyses give a selfconsistent picture: slips arise with either complete suppression of CbN cell spiking or a loss of stride-related modulation concurrent with the slip. We therefore think that further LFP experiments would not be helpful and would in fact be uninterpretable.

(b) Shifting of the analysis window:

Based on their interest in the early oscillation, the reviewers suggest that we restrict our analyses to the first 120 ms of each trial. Given that the oscillation is

in the LFP of anesthetized mice, and does not appear to be transmitted, this proposal does not seem well justified.

– It assumes without evidence that the activity in the cerebellar nuclei that is related to the slip necessarily precedes the slip rather than being concurrent with the slip.

– It ignores the fact that slips often last for several hundred milliseconds of the 1-second stimulation (shown in Figure 5, Figure 10).

– It is based on an unlikely assumption that the hindlimb movement for the full 1 second is secondary to the forelimb movement in the first 120 ms, and that this would be evident at the level of the firing by cells selected for having hindlimb-associated activity.

– It gives no justification for ignoring firing patterns/rates during the slip itself. As the reviewers point out, the relation between cerebellar activity and movement can be quite complex and cells cannot be treated as simply sensory or motor.

We were aware of the ambiguity in identifying an ideal analysis window, however, we did the analysis in two ways (since the first submission). We first used the most assumption-free approach by analyzing the entire 1-second and testing for relationships between firing and behavior. Figure 9 illustrates the attributes of firing that evoked slips or non-slips during steps and trains. All data that were gathered are represented, without selection. Clear patterns emerged, discussed in the manuscript. Then we used a more conservative approach and reanalyzed all the data restricting the window to the time of the slip, for slips in which a clear onset and offset could be detected (Figure 9—figure supplement 1). These gave similar results, suggesting that the relevant measures are associated with neuronal activity that coincides with the slip. Thus, excluding the first 120 ms of the spike records gives results that are predictive of the slip outcome, which emphasize the inappropriateness of discarding all but the first 120 ms.

(c) Analyses of single trials:

Please note that we have already included trial-by-trial measures. Since the first submission, we made use of the “bug” of trial-by-trial variability referred to in the message from the Editor. Every trial is represented individually in Figure 9 and Figure 9—figure supplement 1. The difference between the non-slip and slip trials was whether or not the instantaneous firing rate of CbN cells continued to vary with respect to the phase of the stride in the same pattern as during normal locomotion, albeit with a lower firing rate (which correlated with a non-slip), or an altered pattern relative to normal locomotion (which correlated with a slip). This is illustrated in Figure 8 and Figure 10. The reviewers apparently overlooked this central aspect of the data.

(d) Curtailment of inferences:

Although we do not agree with the reviewers’ stance, we have edited the full manuscript to cut out all inferences that CbN cell firing rate modulation might be necessary for locomotion (inferred from the observation that locomotion was disrupted when modulation was lost) as well as all references to physiological synchrony, except the last paragraphs of the Discussion, since our including these ideas seemed to be central to the reviewers’ objections.

(e) Distinction from previous work:

Three groups have all used optogenetic stimulation that elevated CbN cell activity in stationary animals, which led to muscle contraction, and one group showed that Purkinje cell activation alters whisker set point but not movement. We cited this work. We are not aware of work published by others in which it is demonstrated that an ongoing repetitive well-trained movement is interrupted by the suppression of CbN cell activity (not during rebound spiking).

Reviewer #1:The authors have added new experiments and analyses to their manuscript, which is now significantly improved. Specifically, they have added some additional analyses as well as LFP recordings from the cerebellar cortex in anaesthetized mice during their optogenetic stimulation protocols.I have the following comments about the new experiments:1) The LFP recordings demonstrate – when viewed over the whole 1 s stimulation time window – that there are obvious differences in the coherence of Purkinje cells between step and train stimuli. What is worrying, however, is that in the initial ~60 ms, there is a clear oscillatory response present in their recordings (most probably reflecting synchronous firing). Given that the average latency to a slip is ~120 ms (reduced from 160 ms in the initial manuscript), this could be an important time frame for consideration, especially if the hindlimbs are 'followers' of a forelimb slip.

Please see response above on the LFP data and the direct recordings.

Related to this point, the authors bin across the PC and CbN firing rates in Figure 6 over 50 ms, which prevents the assessment of short term synchronous dynamics. In Figure 7, meanwhile, they assess spiking over the relevant (single millisecond) timescales.

This is why we plotted spiking responses with two bin widths: in Figure 6, with 50- ms bins so that the firing rates would be evident; in Figure 7, with 1-ms bins for comparison of spike timing. The latter permitted us to assess the short-term dynamics, and to observe that the steps gave consistent reductions in CbN cell activity after the first 10 ms, whereas the trains gave coherently alternating decreases and increases in spike probability.

2) It is a little disappointing (though understandable given the technical challenge) that the authors did not perform these LFP recordings under the same conditions as the rest of the experiments in the manuscript – i.e. in awake, running mice. While the conditions for the requested LFP recordings were not explicitly stated in the "essential reviews", it would obviously have been better to have these results obtained under the same conditions, especially considering the comments of Reviewer 3, Major point 3. Nevertheless, having seen the data from anaesthetized mice, it would be helpful to know whether:(a) Varying the duration of light steps (say down to the 60 ms range) produces similar results to the 1 second stimulation, and(b) Whether non-slip trials show a greater degree of LFP oscillation (Purkinje cell synchrony) than slip trials.If the authors have this data it would be helpful to include it in the final version of the manuscript.

Please see comments above on LFP recordings and their limitations.

3) The methodological description of the LFP recordings is a little confusing. The authors state that (1) the fiber was normally ~4-5 mm from the tip of the pipette, (2) the CbN recordings were made 1.9-2.5 mm below the cerebellar surface, and (3) for LFP recordings, the pipette tip was broken at the level of the optic fiber and placed as if for CbN recordings, implying a 1.5-3.1 mm distance (4-5 minus 1.9-2.5) between the recording pipette and cerebellar surface. Is this correct?

This has been clarified as follows: “For local field potential recordings, the pipette tip was broken to give an electrode resistance of 0.8-1 MW, which permitted the fiber to be positioned as it was for direct recording from CbN cells.”

Reviewer #3:While the authors have added helpful new experimental data, I continue to have major difficulties with this paper. I am unconvinced that the revisions have resolved the serious issues raised by all three reviewers in the previous round. I outline my major concerns below. In summary, in my view, once the caveats with the experimental approach that must be considered are taken into account, the conclusions left do not constitute a major advance in our understanding of cerebellar circuit processing and locomotion.1) Here is the logic as it is laid out by the authors in the abstract (and elsewhere):a. 'Trains suppressed CbN firing less effectively' than steps;b. Trains 'altered spike timing';c. Steps or trains that perturbed 'stride related modulation' (which the authors confusingly use to refer to timing) correlated well with irregularities of movement;d. However, 'Unperturbed locomotion continued more often during trains than steps'.In other words, trains altered spike timing more effectively, but didn't cause as many slips, yet the authors' conclusion is that modulation of spike timing is what matters for inducing slips. As the authors acknowledge with the 'however', this is fundamentally contradictory. If trains truly altered spike timing more effectively, and spike timing cause irregularities in movement, then trains simply should not allow locomotion to continue unperturbed more often than during steps. The authors speculate that 'the modulation of CbN spiking required for locomotion may persist more easily during synchronous inhibition' and that this could account for the apparent discrepancy, but this is highly speculative, not to mention extremely confusing for me as an expert reviewer, and presumably even more so for a broad readership. Insufficient data is presented to support this idea and in my view, it cannot be used to explain the apparently contradictory findings, given the necessary caveats in interpreting the experiments as they were done (using sudden disruption of neural activity with optogenetics).

The reviewer has confused two issues: the time scale of what we call “stride-related modulation,” which is the increase and decrease of instantaneous firing rate on the time scale of a stride (200-300 ms), and the time scale of spike resetting by trains, on the time scale of the interpulse intervals of the trains (5-20 ms). It is possible to reset the timing of spikes by a train and still retain stride-related modulation on the time scale of the stride. (For example, if in control locomotion, most spikes occur in the first 100 ms of the stride, with the train stimuli, most spikes might still occur in the first 100 ms, but their regularity might be altered.) We added the following text: “We refer to this change in instantaneous firing rate on the time scale of the stride (~150-400) ms as “stride-related modulation.” Later, we include the text: “It is worth noting that the consequent resetting of spike times in the CbN cells, occurred on a time scale of 5-20 ms, a much shorter period than the duration of a stride.” We have also edited throughout the manuscript to make sure that the time scales are made clear.

What we find in this work is (1) that when slips occur, stride-related modulation is altered (evident in single-trial analyses). Note that this is true when Purkinje cell activity was altered by both steps and trains of optogenetic stimulation; i.e., it is independent of whether Purkinje cell synchrony was experimentally elevated. We also find that (2) when rate is controlled for, slips (i.e., loss of stride-related modulation) occurred 3x-4x more often with steps than with trains. Stating the converse, stride-related modulation was retained 3x-4x more often with trains than steps. This observation stands to biophysical reason: the gaps in experimentally applied inhibition by trains permit windows in which excitation (from mossy fibers) can effectively produce spikes, with rates that increase and decrease over the time scale of the stride, in a manner similar to control conditions. So there are not actually any contradictory findings.

The reviewer is inaccurate in saying that we claim that the “modulation” of spike timing on the time scale of ms by trains correlates with normal locomotion. We have tried to make this point clear by referring to “stride-related modulation” (fluctuations in instantaneous firing rate on the time scale of 200-300 ms) at several points in the manuscript. What we note is that (1) because nonslip trials persist more during trains, and (2) because nonslip trials correlate with stride-related modulation, it seems that trains—despite the resetting of spike timing on the time scale of 5-20 ms—are more permissive of stride related modulation. This is an observation, not an interpretation. The “however” contrasts the similarity between steps and trains (they both can cause slips) with the difference between them (one is more effective). There is no contradiction.

2) I strongly disagree with the repeated claims in the manuscript and the rebuttal that the finding that ongoing locomotion is perturbed by optogenetic stimulation provides any evidence at all that 'modulated CbN cell firing is necessary for normal, ongoing locomotion.' I pointed this problem out in my previous review, at which time I thought that the authors had simply not been careful with their language. However, now they are insisting on this point. To show that modulated CbN firing is necessary for normal, ongoing locomotion it would be necessary to effectively clamp CbN firing at a certain level and prevent it from modulating with the stride. This is not currently possible with optogenetic stimulation in spontaneously active neurons and the kind of acute perturbation of spiking presented here is particularly unsuited to making any claims about what natural patterns of spike timing may be doing during locomotion.(See for example:1. From the rebuttal: 'we can safely conclude that transient disruption of CbN output rapidly induces discontinuities of gait' [Ok] and 'we can begin to infer that cerebellar output, and indeed modulated cerebellar output, is necessary for smooth, learned locomotion'. [No, we cannot; this has not been shown and is presented as a primary result of the manuscript].2. From the rebuttal, 'These observations… permit us to conclude that cerebellar output consisting of modulated CbN activity is required even for well-trained ongoing repetitive movements like locomotion.' Again, I fundamentally disagree that they do.)

Since loss of stride-related modulation correlates with loss of smooth movement, we had inferred (more exactly “we began to infer”) that stride-related modulation might be necessary for smooth movement. To comply with the reviewer, we removed this inference throughout the manuscript and leave it to others to reason or not as follows:

– Nonslips retain stride-related modulation of firing rates at same phase as control (no-light), with or without suppression of firing rates.

– Slips lack stride-related modulation of firing rates at same phase as control (no-light), with or without suppression of firing rates.

– From this we “begin to infer” (trying to put it cautiously, but still considering implications of the data) that stride-related modulation of firing rates at same phase as control (no-light), with or without suppression of firing rates, may be necessary for locomotion: because when it goes away, smooth locomotion goes away.

– The reviewer’s desire to clamp CbN firing at a certain level and preventing it from modulating was effectively done by analyzing trials with equivalent firing rates Figure 9H, I, J, K, L, M. In many of these trials, the firing rates are unchanged from control (within-trial measures) and the same as each other (overlapping open and closed symbols) yet there were slips and non-slips. The non-slips retained modulation. The slips lost modulation. This is why we think that modulation is related to nonslips.

These observations follow directly from the data and we are puzzled how they can be subject to the “fundamental disagreement” expressed by the reviewer.3) There are many statements throughout the paper and rebuttal (including the statement quote in point #2 above) relating to the idea that one important contribution of this paper is to demonstrate that sudden optogenetic perturbations of CbN neuron activity perturbs even well-practiced, ongoing locomotion. But this is entirely expected. It has been shown repeatedly, for example by the Lena, Medina, and Otis groups that optogenetic perturbation of Purkinje cell activity affects movement acutely in naïve animals. Disrupting movement of a limb of a mouse on a freely rotating cylinder will obviously, necessarily, induce 'slips', particularly given that the authors are not even claiming that the slips they observe are a direct result of perturbation of control of the hindlimb (this is in fact their main defense for not analyzing forelimb movements.) Moreover, 'To what extent CbN cell activity is assistive, predictive, reactive, or generative remains open to discussion' is used to motivate this study. But this study does not speak to those questions. CbN neurons are deep within the brain. They are not purely sensory or motor. Disturbing their activity with a sudden optogenetic perturbation of many CbN neurons at once perturbs movement, as expected, because they project directly to motor output areas. But this does not inform what signals drive their normal activity under physiological conditions.

Three groups have done experiments, which we cited, that involve optogenetic stimulation that lead to elevations of CbN cell activity in stationary animals and contraction of muscles. One group also has published that Purkinje cell activation changed whisker set point but not whisking for exploration, which we also cited. We are not aware of work that has shown the results in the present manuscript, namely that an ongoing repetitive well-trained, complex movement, such as locomotion, is interrupted by the transient suppression of CbN cell activity.

The differing perspectives presented by the reviewers seem to support the statement that it is open to discussion whether CbN cell activity assists (facilitates), predicts (carries a copy of a motor command that may be canceled elsewhere, having little effect on movement), reacts (makes corrections to movements that have happened), or generates (drives) movement. However, we edited the sentence to include the idea that the particular movement might matter. The question of what CbN cells do or don’t do probably does not have a unique answer. However, the observation that transiently suppressing firing altogether or transiently disrupting stride-related modulation by altering the activity of Purkinje neurons (that project solely to CbN cells) correlates with movement errors does start to give hints about how CbN cells might contribute to this specific behavior.

4) According to the authors, after the primary demonstration that disruptions of CbN activity perturb movement, their second main result is the demonstration that 'steps and trains of light that disfavor or favor Purkinje cell synchrony evoke different responses in CbN cells and different behavioral outputs.' Although the authors have added LFP data to support the idea that the optogenetic stimulation resulted in more synchrony with trains than steps (though steps also induced synchrony for the first ~100ms or so), I am still not convinced that the paper presents results that speak to the question of how physiological PC/CbN synchrony might affect locomotion. In some places in the rebuttal for example, the authors admit that they cannot speak to physiological synchrony. But sometimes they say things like: 'The results provide a framework for interpreting systems-level observations in light of observations from cellular physiology.' The link here between the two levels is too tenuous and statements like this are unsupported by actual data. Moreover, if their claims are limited to claims about optogenetic synchrony (which I think they necessarily are), I do not see this as a meaningful advance.

While it may or may not be appropriate for the reviewer to ask us to curtail our thinking about the implications of the data or to censor a cautious statement to the effect that the results provide a framework for interpreting existing ideas, we removed the statement that we can use the data as a framework for interpreting, and we have removed all references to the existence of physiological synchrony except in the last paragraph. Regarding the meaningfulness of the advance, please see point #3.

5) The first part of the paper is meant to be a description of PC/CbN activity in mouse cerebellum during locomotion, which the authors argue, and I agree, is a necessary step in addressing their later questions. However, this analysis is incomplete and basic aspects of this characterization are lacking. For this part of the paper to represent an advance on its own, given the many similar published studies from other species, the authors would need to thoroughly analyze the speed dependence of the physiological responses they observe, for example. Given that they cannot speak to whether they were in a fore or hindlimb region, they would also need to at least consider changes in interlimb coordination and timing across speeds, to assess whether those could account for changes in neural responses, either across speeds in the same cells, or even possibly across cells, if the animals were locomoting at different speeds in different recordings. For example, in Figure 4, could changes in interlimb phasing across speeds account for these different categories? How common are the different categories of responses across speeds? Are these even distinct categories of responses? Similarly, in Figure 3, fifth paragraph of the Results, Figures 3E, F do not show that across speeds/stride durations 'firing rates tended to rise and fall at consistent phases of the stride cycle for both Pkj and CbN cells'. 3E, F are each one representative neuron, and there is no analysis of the relative phase modulation at different stance durations. The fact that there is still an average modulation when normalizing for stride duration does not speak to whether there is a consistent dependence of phase on stride duration. This is relevant, again, because fore/hind limb phasing also depends on speed.

The reviewer has misunderstood the goal of the manuscript. The first part of the

manuscript is not an effort to generate a complete, stand-alone description of locomotion, but is in service of the optogenetic experiments that follow. We tracked the hind paw because it was reliably detectable by our camera (more so than the fore paw) and we verified that optogenetic stimulation of the recording region led to hindlimb movements. That, and the constraints on location, were the means by which we limited the variance in the cells from which we recorded. We had neither interest in nor pretensions to studying interlimb phasing, which we did not record, and which is irrelevant to addressing the question of how do CbN cells fire during unperturbed locomotion and during different patterns of experimentally altered Purkinje cell activity that disrupt or fail to disrupt smooth locomotion, which is what we did study. There were not enough data to analyze each cell by speed because not all mice had enough strides at a range of speeds. Pilot analyses verified that the phase relationship between firing and the stride cells did not change with speed, and that eliminating especially long or especially short strides did not significantly alter the plots of stride-related modulation, justifying collapsing the data across stride durations. The reviewer is requesting not only a different analysis, but an entirely different study, which is beyond the scope or intentions of the experiments, data, or questions at hand.6) In the sixth paragraph of the Results and throughout the paper: The modulation index and the subsequent use of the word 'modulation' to refer both to changes in spike rate and timing is extremely confusing. In the rebuttal the authors try to explain that they are using modulation to refer to changes only in temporal patterns but the definition itself is spike-rate based. This needs to be fixed.

This comment reflects the misunderstanding mentioned above. We use “stride-related modulation” and “firing rate modulation” to refer to the rise and fall in firing rates on the time scale of the stride. This is defined in the text and reinforced in several places. Twice we referred to modulation of spike probability, which we have now removed. Once in the revision we used “modulation” with respect to changing firing in the LFP condition, which we have now removed. Elsewhere, we use the word modulation (52 times) only to refer to the changes in instantaneous firing rates on the time scale of the stride. See also comments about the time scales, response to point #1. We edited the text to verify that we used “modulation” or “stride-related modulation” to refer to the changes in firing rate on the time scale of the stride. We also added several repetitions of the time scale of the spike timing changes induced by trains for clarity.

[Editors' note: further revisions were requested prior to acceptance, as described below.]

Following this revision, the conclusions of the paper are now more in harmony with the data with respect to interpretation. However the reviewers had several comments that we would like you to consider and address so that the Senior editor can make a final decision on the manuscript. These are included below. Some are primarily for your information, but some will require adjustments in the text.1) Interpretationsa) The changes reflected in the rewriting of the first and the last sentences of the first paragraph of the Discussion, and the results are major improvements over their predecessors, which attempted to make larger claims about what was "required" during normal locomotion. These findings are now presented in a way that is adequately supported by the data.b) The changes in the Discussion (which now reads "In either case, these data provide evidence that cerebellar output can actively participate in locomotion") and elsewhere are similarly crucial.c) Synchrony: The authors have taken increased care in discussing physiological synchrony throughout the paper.d) Terminology: Clarifications of the use of terms such as spike rate, spike modulation, and spike timing/synchrony are a major improvement. The authors could go further in this regard, perhaps using "depth of modulation" more consistently, and avoiding the term "spike timing" if at all possible (because "timing" alone still could refer to time relative to the stride). These suggested changes may be helpful for eLife's broad readership.

We checked the manuscript for these phrases. There are five uses of “spike timing.” Two of them included the modifier “on a time scale of milliseconds.” We have added “on a millisecond time scale” (or similar phrase) as modifiers on the two other uses in the text. The final use is in the legend where we referred to “rates and timing” which has been changed to “rates and patterns.”

“Depth of modulation” is used only twice in the manuscript, both times explicitly to refer to the modulation index MI that is has a specific calculation associated with it. “Depth” is used on two other occasions to refer to the depth of the electrode but this usage seems clearly distinct.

2) Beyond interpretation, the major technical issue was to what extent synchronous vs. asynchronous responses had been demonstrated in response to trains vs. steps. The reviewers asked for a demonstration in awake animals that steps did not elicit synchrony because recordings in the anesthetized animal may not reflect the situation in awake animals. Without such a direct comparison, the conclusions need to be adjusted accordingly. Much of the toning down in the revised text helps here, but this caveat (anesthetized vs. awake LFP recordings, and the issue of the synchrony at the start of the steps) should be explicit in the Discussion.

We have added a paragraph to the Discussion, with explicit reference to what can and cannot be learned from the various approaches taken.

3) Proville et al. did show that stimulating Purkinje cells during whisking caused whisker deflections (see Figure 6C of their paper). They call this a change in setpoint, and argue that it did not interfere with ongoing whisking. However, the stimulation did in fact induce a movement. If one changes the position (setpoint) of whiskers, the mouse can keep whisking (as Proville et al. found). However, if one changes, for example, the position (setpoint) of a front limb during ongoing locomotion, it will necessarily "disrupt locomotion" at the level measured in the current study – because it will prevent that limb from touching down at the expected time (if the paw is in the air) or it will directly perturb the wheel motion (if the paw is on the surface when the stimulation comes). The consequent change in support pattern or wheel motion will cause compensatory changes in the movement of all limbs, which will resemble hind limb "slips" as measured in the current study. Until now, the authors seemed to be assuming that their stimulations were directly disrupting locomotion itself – whereas it is entirely possible that stimulation simply induces movement of a limb, and that locomotor patterns are altered as a result. For example, the same slips in the hindpaws and the same changes in CbN activity might occur if one physically halted the mouse's front limb while it was in the air during a step. The authors now acknowledge this possible alternative interpretation at several points in the manuscript.No response requested.4) In response to the concerns about the completeness of the description of neural responses during locomotion, the authors state in their rebuttal, "Pilot analyses verified that the phase relationship between firing and the stride cells did not change with speed, and that eliminating especially long or especially short strides did not significantly alter the plots of stride-related modulation, justifying collapsing the data across stride durations." This is quite useful information that we recommend should be incorporated in the Results section.

This information is already in the manuscript and is illustrated in Figures 3C and 3D. We added the sentence from the rebuttal (mildly edited for clarity) just to drive this point home further.

5) Introduction, paragraph two: "Consequently, to what extent and during which behaviors CbN cell activity may be assistive, predictive, reactive, or generative remains open to discussion." This sentence should be deleted as its placement gives a misleading impression of what the paper is about. The question in the third paragraph asks the same thing in a way that is more relevant for the paper, and the same point is made elsewhere, including in the Discussion.

The sentence has been deleted. However, the intent of the sentence was meant to organize possibilities, not to claim that we were going to close the discussion.

6) Paragraph beginning: "The present experiments on voluntary locomotion in intact mice demonstrate that transiently suppressing modulation of CbN cell firing by directly altering Purkinje cell activity is sufficient to disrupt smooth, well-practiced locomotion”. Because the authors highlight the effects of stimulation (as opposed to inhibition) of Purkinje cell activity, they could be more specific here and say "stimulation of" instead of "altering." At the end of this paragraph, "In either case, these data provide evidence that cerebellar output can actively participate in locomotion" is a major improvement, as it does not infer causality that has not been demonstrated. The authors could also say something like "and cerebellar output is tightly linked to locomotor performance," if they wanted to strengthen this statement.

The word “altering” has been changed to “stimulating.”

7) Subsection “Modulation of Purkinje and CbN cells during ad-lib locomotion”, passage "Here, we found that the pattern of modulation with respect to the stride across a given Purkinje or CbN cell did not depend strongly on stride duration; possibly the range of values assumed by variables such as pitch and roll was reduced by head fixation.": This is an interesting point, but it is also possible that this could be a species difference, perhaps resulting from speed-dependent interlimb phasing differences in rats vs mice.

We changed “possibly” to “one possibility is that” to make it clear that the proposal is not exclusive.

8) Second paragraph subsection “Cerebellar output and the persistence of well-trained movements”: Suggestion: "Such observations suggest that such modulation normally seen in cerebellar output neurons… may facilitate but may not be necessary for locomotion." Delete: "represents sensory feedback associated with movement, which", because the paper does not address this issue.

This sentence does not relate to our own data but is a summary of other people’s interpretations of their own studies. We altered “suggest” to “have led to the suggestion” so that it is clear that we are not making a claim based on our own data, but are referring to the literature.

9) In the same paragraph: Replace "may vary depending on the behavior and the conditions" with "is unclear".

We changed the sentence to say: “may vary with behavior and/or conditions.” (The phrase “is unclear” is uninformative and we prefer to avoid it.)

10) The x axis label for the bottom (histogram) panels of 6A, B should be seconds not milliseconds.

Thank you. This has been corrected.